# Convolutional Neural Operators for robust and accurate learning of PDEs

**Bogdan Raonić**[1,2]  **Roberto Molinaro**[1]  **Tim De Ryck**[1]  **Tobias Rohner**[1]

**Francesca Bartolucci**[3]  **Rima Alaifari**[1]  **Siddhartha Mishra**[1,2]  **Emmanuel de Bézenac**[1]

1 Seminar for Applied Mathematics, ETH, Zurich, Switzerland
2 ETH AI Center, Zurich, Switzerland
3 Delft University of Technology, Netherlands

## Abstract

Although very successfully used in conventional machine learning, convolution based neural network architectures – believed to be inconsistent in function space – have been largely ignored in the context of learning solution operators of PDEs. Here, we present novel adaptations for convolutional neural networks to demonstrate that they are indeed able to process functions as inputs and outputs. The resulting architecture, termed as convolutional neural operators (CNOs), is designed specifically to preserve its underlying continuous nature, even when implemented in a discretized form on a computer. We prove a universality theorem to show that CNOs can approximate operators arising in PDEs to desired accuracy. CNOs are tested on a novel suite of benchmarks, encompassing a diverse set of PDEs with possibly multi-scale solutions and are observed to significantly outperform baselines, paving the way for an alternative framework for robust and accurate operator learning.

## 1   Introduction.

Partial Differential Equations (PDEs) [13] are ubiquitous as mathematical models in the sciences and engineering. Solving a PDE amounts to (approximately) computing the so-called *solution operator* that maps function space inputs such as initial and boundary conditions, coefficients, source terms etc, to the PDE solution which also belongs to a suitable function space. Well-established numerical methods such as finite differences, finite elements, finite volumes and spectral methods (see [48]) have been very successfully used for many decades to approximate PDE solution operators. However, the prohibitive computational cost of these methods, particularly in high dimensions and for *many query* problems such as UQ, inverse problems, PDE-constrained control and optimization, necessitates the design of *fast, robust and accurate* surrogates. This provides the rationale for the use of *data-driven machine learning* methods for solving PDEs [21].

As *operators* are the objects of interest in solving PDEs, learning such operators from data, which is loosely termed as *operator learning*, has emerged as a dominant paradigm in recent years for the applications of machine learning to PDEs. A very partial list of architectures for operator learning include operator networks [9], DeepONets [39] and its variants [41, 7], PCA-net [5] , neural operators [25] such as graph neural operator [34], Multipole neural operator [35] and the very popular Fourier Neural Operator [33] and its variants [36, 45], VIDON [47], spectral neural operator [14], LOCA [23], NOMAD [52] and transformer based operator learning architectures [8].

37th Conference on Neural Information Processing Systems (NeurIPS 2023).

Despite the considerable success of the recently proposed operator learning architectures, several pressing issues remain to be addressed. These include, but are by no means restricted to, limited expressivity for some of these algorithms [28] to aliasing errors for others [14] to the very fundamental issue of possible *lack of consistency in function spaces* for many of them. As argued in a recent paper [2], a structure-preserving operator learning algorithm or *representation equivalent neural operator* has to respect some form of *continuous-discrete equivalence* (CDE) in order to learn the underlying operator, rather than just a discrete representation of it. Failure to respect such a CDE can lead to the so-called *aliasing errors* [2] and affect model performance at multiple discrete resolutions.

Despite many attempts, see [1, 16] and references therein, the absence of a suitable CDE, resulting in aliasing errors, has also plagued the naive use of convolutional neural networks (CNNs) in the context of operator learning, see [63, 33, 2] on how using CNNs for operator learning leads to results that heavily rely on the underlying grid resolution. This very limited use of Convolution (in physical space) based architectures for operator learning stands in complete contrast to the fact that CNNs [30] and their variants are widely used architectures for image classification and generation and in other contexts in machine learning [29, 37, 61]. Moreover, CNNs can be thought of as natural generalizations of the foundational finite difference methods for discretizing PDEs [17, 38]. Given their innate locality, computational and data efficiency, ability to process multi-scale inputs and outputs and the availability of a wide variety of successful CNN architectures in other fields, it could be very advantageous to bring CNN-based algorithms back into the reckoning for operator learning. This is precisely the central point of the current paper where we make the following contributions,

- We propose novel modifications to CNNs in order to enforce structure-preserving continuous-discrete equivalence and enable the genuine, alias-free, learning of operators. The resulting architecture, termed as *Convolutional Neural Operator* (CNO), is instantiated as a novel *operator* adaptation of the widely used U-Net architecture.

- In addition to showing that CNO is a *representation equivalent neural operator* in the sense of [2], we also prove a universality result to rigorously demonstrate that CNOs can approximate the operators, corresponding to a large class of PDEs, to desired accuracy.

- We test CNO on a *novel* set of benchmarks, that we term as *Representative PDE Benchmarks* (RPB), that span across a variety of PDEs ranging from linear elliptic and hyperbolic to nonlinear parabolic and hyperbolic PDEs, with possibly *multiscale solutions*. We find that CNO is either on-par or outperforms the tested baselines on all the benchmarks, both when testing in-distribution as well as in *out-of-distribution* testing.

Thus, we present a new CNN-based operator learning model, with desirable theoretical properties and excellent empirical performance, with the potential to be widely used for learning PDEs.

## 2   Convolutional Neural Operators.

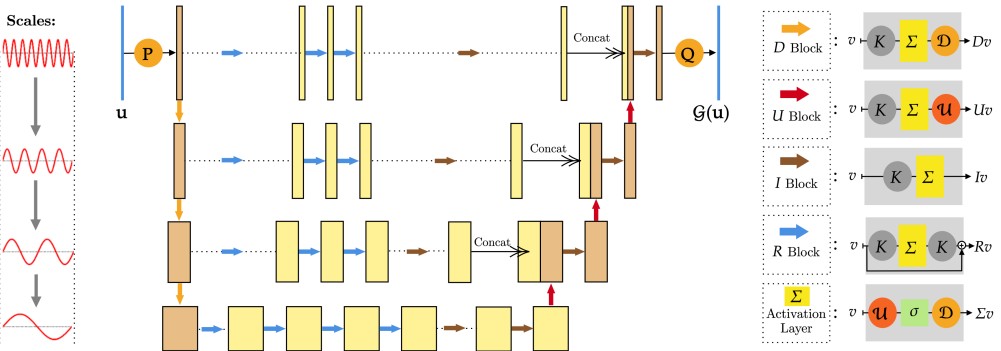

Figure 1: Schematic representation of CNO (2.3) as a modified U-Net with a sequence of layers (each identified with the relevant operators on the right, see Section 2) mapping between bandlimited functions. Rectangles represent multi-channel signals. Larger the height, larger is the resolution. Wider the rectangles, more channels are present.

**Setting.** For simplicity of the exposition, we will focus here on the two-dimensional case by specifying the underlying domain as $D = \mathbb{T}^2$, being the 2-d torus. Let $\mathcal{X} = H^r(D, \mathbb{R}^{d_{\mathcal{X}}}) \subset \mathcal{Z}$ and $\mathcal{Y} = H^s(D, \mathbb{R}^{d_{\mathcal{Y}}})$ be the underlying function spaces, where $H^{r,s}(D, \cdot)$ are Sobolev spaces of order $r$ and $s$. Without loss of generality, we set $r = s$ hereafter. Our aim would be to approximate *continuous operators* $\mathcal{G}^\dagger : \mathcal{X} \to \mathcal{Y}$ from data pairs $\left(u_i, \mathcal{G}^\dagger(u_i)\right)_{i=1}^M \in \mathcal{X} \times \mathcal{Y}$. We further assume that there exists a *modulus of continuity* for the operator i.e.,

$$\|\mathcal{G}^\dagger(u) - \mathcal{G}^\dagger(v)\|_{\mathcal{Y}} \leq \omega\left(\|u - v\|_{\mathcal{Z}}\right), \quad \forall u, v \in \mathcal{X}, \tag{2.1}$$

with $\omega : \mathbb{R}_+ \to \mathbb{R}_+$ being a monotonically increasing function with $\lim_{y \to 0} \omega(y) = 0$. The underlying operator $\mathcal{G}^\dagger$ can correspond to solution operators for PDEs (see Section 3 for the exact setting) but is more general than that and encompasses examples such as those arising in inverse problems, for instance in imaging [4].

**Bandlimited Approximation.** As argued in a recent paper [2], Sobolev spaces such as $H^r$ are, in a sense, *too large* to allow for any form of *continuous-discrete equivalence* (CDE), i.e., equivalence between the underlying operator and its discrete representations, which is necessary for robust operator learning. Consequently, one has to consider smaller subspaces of $H^r$ which allow for such CDEs. In this respect, we choose the space of *bandlimited functions* [57] defined by,

$$\mathcal{B}_w(D) = \{f \in L^2(D) : \mathrm{supp}\,\widehat{f} \subseteq [-w, w]^2\}, \tag{2.2}$$

for some $w > 0$ and with $\widehat{f}$ denoting the Fourier transform of $f$. It is straightforward to show using (2.1) (see **SM**A.1) that for any $\varepsilon > 0$, there exists a $w$, large enough depending on $r$, and a continuous operator $\mathcal{G}^* : \mathcal{B}_w(D) \to \mathcal{B}_w(D)$, such that $\left\|\mathcal{G}^\dagger - \mathcal{G}^*\right\| < \varepsilon$, with $\|\cdot\|$ denoting the corresponding operator norm. In other words, the underlying operator $\mathcal{G}^\dagger$ can be approximated to arbitrary accuracy by the operator $\mathcal{G}^*$ that maps between band-limited spaces. Consequently, as shown in **SM**A.2, one can readily define discrete versions of $\mathcal{G}^*$ using the underlying *sinc* basis for bandlimited functions and establish a continuous-discrete equivalance for it.

**Definition of CNO.** Given the above context, our goal will be to approximate the operator $\mathcal{G}^*$ in a *structure-preserving manner* i.e., as the underlying operator maps between spaces of bandlimited functions, we will construct our operator approximation architecture to also map bandlimited functions to bandlimited functions, thus respecting the continuous-discrete equivalence. To this end, we denote the operator $\mathcal{G} : \mathcal{B}_w(D) \to \mathcal{B}_w(D)$ as a *convolutional neural operator* (CNO) which we define as a compositional mapping between functions as

$$\mathcal{G} : u \mapsto P(u) = v_0 \mapsto v_1 \mapsto \ldots v_L \mapsto Q(v_L) = \overline{u}, \tag{2.3}$$

where

$$v_{l+1} = \mathcal{P}_l \circ \Sigma_l \circ \mathcal{K}_l(v_l), \quad 1 \leq \ell \leq L - 1. \tag{2.4}$$

From (2.3), we see that first, the input function $u \in \mathcal{B}_w(D)$ is lifted to the latent space of bandlimited functions through a *lifting layer*:

$$P : \left\{u \in \mathcal{B}_w(D, \mathbb{R}^{d_{\mathcal{X}}})\right\} \to \left\{v_0 \in \mathcal{B}_w(D, \mathbb{R}^{d_0})\right\}.$$

Here, $d_0 > d_{\mathcal{X}}$ is the number of channels in the lifted, latent space. The lifting operation is performed by a convolution operator which will be defined below.

Then, the lifted function is processed through the composition of a series of mappings between functions (layers), with each layer consisting of three elementary mappings, i.e., $\mathcal{P}_l$ is either the *upsampling* or *downsampling* operator, $\mathcal{K}_l$ is the convolution operator and $\Sigma_l$ is the activation operator. These elementary operators are defined below and are inspired by the modifications of CNNs for image generation in [22]. Finally, the last output function in the iterative procedure $v_L$ is projected to the output space with a *projection operator* $Q$, defined as

$$Q : \left\{v_L \in \mathcal{B}_w(D, \mathbb{R}^{d_L})\right\} \to \left\{\overline{u} \in \mathcal{B}_w(D, \mathbb{R}^{d_{\mathcal{Y}}})\right\}.$$

The projection operation is also performed by a convolution operator defined below.

**Convolution Operator.**    For simplicity of exposition, we will present the *single-channel* version of the convolution operator $\mathcal{K}_l$ here. See **SM** A.3 for the *multi-channel* version for this and other operators considered below. Convolution operations are performed with discrete kernels

$$K_w = \sum_{i,j=1}^{k} k_{ij} \cdot \delta_{z_{ij}}$$

defined on the $s \times s$ uniform grid on $D$ with grid size $\leq 1/2w$, in-order to satisfy the requirements of the Whittaker-Shannon-Kotelnikov sampling theorem [56], and $z_{ij}$ being the resulting grid points, $k \in \mathbb{N}$ being the kernel size and $\delta_x$ denoting the Dirac measure at point $x \in D$. The convolution operator for a *single-channel* $\mathcal{K}_w : \mathcal{B}_w(D) \to \mathcal{B}_w(D)$ is defined by

$$\mathcal{K}_w f(x) = (K_w \star f)(x) = \int_D K_w(x-y)f(y)dy = \sum_{i,j=1}^{k} k_{ij} f(x - z_{ij}), \quad \forall x \in D,$$

where the last identity arises from the fact that $f \in \mathcal{B}_w$. Thus, our convolution operator is directly parametrized in physical space, in contrast to the Fourier space parametrization of a convolution in the FNO architecture of [33]. Hence, our parametrization is of a *local* nature.

**Upsampling and Downsampling Operators.**    For some $\overline{w} > w$, we can *upsample* a function $f \in \mathcal{B}_w$ to the *higher band* $\mathcal{B}_{\overline{w}}$ by simply setting,

$$\mathcal{U}_{w,\overline{w}} : \mathcal{B}_w(D) \to \mathcal{B}_{\overline{w}}(D), \quad \mathcal{U}_{w,\overline{w}} f(x) = f(x), \quad \forall x \in D.$$

On the other hand, for some $\underline{w} < w$, we can *downsample* a function $f \in \mathcal{B}_w$ to the *lower band* $\mathcal{B}_{\underline{w}}$ by setting $\mathcal{D}_{w,\underline{w}} : \mathcal{B}_w(D) \to \mathcal{B}_{\underline{w}}(D)$, defined by

$$\mathcal{D}_{w,\underline{w}} f(x) = \left(\frac{\underline{w}}{w}\right)^2 (h_{\underline{w}} \star f)(x) = \left(\frac{\underline{w}}{w}\right)^2 \int_D h_{\underline{w}}(x-y)f(y)dy, \quad \forall x \in D,$$

where $\star$ is the convolution operation on functions defined above and $h_{\underline{w}}$ is the so-called *interpolation sinc filter*:

$$h_w(x_0, x_1) = \mathrm{sinc}(2wx_0) \cdot \mathrm{sinc}(2wx_1), \quad (x_0, x_1) \in \mathbb{R}^2. \tag{2.5}$$

**Activation Layer.**    Naively, one can apply the activation function pointwise to any function. However, it is well-known that such an application will no longer respect the band-limits of the underlying function space and generate *aliasing errors* [22, 14, 2]. In particular, nonlinear activations can generate features at arbitrarily high frequencies. As our aim is to respect the underlying CDE, we will modulate the application of the activation function so that the resulting outputs fall within desired band limits. To this end, we first upsample the input function $f \in \mathcal{B}_w$ to a higher bandlimit $\overline{w} > w$, then apply the activation and finally downsample the result back to the original bandlimit $w$ (See Figure 1). Implicitly assuming that $\overline{w}$ is large enough such that $\sigma(\mathcal{B}_w) \subset \mathcal{B}_{\overline{w}}$, we define the activation layer in (2.3) as,

$$\Sigma_{w,\overline{w}} : \mathcal{B}_w(D) \to \mathcal{B}_w(D), \quad \Sigma_{w,\overline{w}} f(x) = \mathcal{D}_{\overline{w},w}(\sigma \circ \mathcal{U}_{w,\widetilde{w}} f)(x), \quad \forall x \in D. \tag{2.6}$$

**Instantiation through an Operator U-Net architecture.**    The above ingredients are assembled together in the form of an Operator U-Net architecture that has bandlimited functions as inputs and outputs. In addition to the blocks that have been defined above, we also need additional ingredients, namely incorporate *skip connections* through *ResNet* blocks of the form, $\mathcal{R}_{w,\overline{w}} : \mathcal{B}_w(D, \mathbb{R}^d) \to \mathcal{B}_w(D, \mathbb{R}^d)$ such that

$$\mathcal{R}_{w,\overline{w}}(v) = v + \mathcal{K}_w \circ \Sigma_{w,\overline{w}} \circ \mathcal{K}_w(v), \quad \forall v \in \mathcal{B}_w(D, \mathbb{R}^d). \tag{2.7}$$

We also need the so-called *Invariant blocks* of the form, $\mathcal{I}_{w,\overline{w}} : \mathcal{B}_w(D, \mathbb{R}^d) \to \mathcal{B}_w(D, \mathbb{R}^d)$ such that

$$\mathcal{I}_{w,\overline{w}}(v) = \Sigma_{w,\overline{w}} \circ \mathcal{K}_w(v), \quad \forall v \in \mathcal{B}_w(D, \mathbb{R}^d). \tag{2.8}$$

Finally, all these ingredients are assembled together in a modified Operator U-Net architecture which is graphically depicted in Figure 1. As seen from this figure, the input function, say $u \in \mathcal{B}_w(D, \mathbb{R}^{d_x})$ is first lifted and then processed through a series of layers. Four types of blocks are used i.e.,

downsampling (D) block corresponding to using the downsampling operator $\mathcal{D}$ as the $\mathcal{P}$ in (2.4), upsampling (U) block corresponding to using the upsampling operator $\mathcal{U}$ as the $\mathcal{P}$ in (2.4), ResNet (R) block corresponding to (2.7) and Invariant (I) block corresponding to (2.8). Each block takes a band-limited function as input and returns another band-limited function (with the same band) as the output. Finally, U-Net style patching operators, which concatenate outputs for different layers as additional channels are also used. As these operations act only in the channel width and leave the spatial resolution unchanged, they conform to the underlying bandlimits. Thus, CNO takes a function input and passes it through a set of encoders, where the input is downsampled in space but expanded in channel width and then processed through a set of decoders, where the channel width is reduced but the space resolution is increased. At the same time, encoder and decoder layers (at the same spatial resolution or band limit) are connected through additional ResNet blocks. Thus, this architectural choice allows for transferring high frequency content via the skip connections, before filtering them out with the *sinc* filter as we go deeper into the encoder. Hence, the high frequency content is not just recreated with the activation function, but also modified through the intermediate networks. Consequently, we build a genuinely *multiscale operator learning architecture*.

**Continuous-Discrete Equivalence for CNO.** We have defined CNO (2.3) as an operator that maps bandlimited functions to bandlimited functions. In practice, like any computational algorithm, CNO has to be implemented in a discrete manner, with *discretized versions* of each of the above-defined elementary operations being specified in **SM** A.4. Given how each of the elementary blocks (convolution, up- and downsampling, activation, ResNets etc) are constructed, we prove the following proposition (in **SM** A.5):

**Proposition 2.1.** *Convolutional Neural Operator* $\mathcal{G} : \mathcal{B}_w(D, \mathbb{R}^{d_\mathcal{X}}) \to \mathcal{B}_w(D, \mathbb{R}^{d_\mathcal{Y}})$ (2.3) *is a* Representation equivalent neural operator *or* ReNO, *in the sense of [2], Definition 3.4.*

Further details about the notion of ReNOs is provided in **SM** A.4 and we refer the reader to [2], where this concept is presented in great detail and the representation equivalence of CNO is discussed. In particular, following [2], representation equivalence implies that CNO satisfies a form of *resolution invariance*, allowing it to be evaluated on multiple grid resolutions without aliasing errors.

## 3 Universal Approximation by CNOs.

We want to prove that a large class of operators, stemming from PDEs, can be approximated to desired accuracy by CNOs. To this end, we consider the following abstract PDE in the domain $D = \mathbb{T}^2$,

$$\mathcal{L}(u) = 0, \quad \mathcal{B}(u) = 0, \tag{3.1}$$

with $\mathcal{L}$ being a differential operator and $\mathcal{B}$ a boundary operator. We assume that the differential operator $\mathcal{L}$ only depends on the coordinate $x$ through a *coefficient* function $a \in H^r(D)$. The corresponding *solution* operator is denoted by $\mathcal{G}^\dagger : \mathcal{X}^* \subset H^r(D) \to H^r(D) : a \mapsto u$, with $u$ being the solution of the PDE (3.1). We assume that $\mathcal{G}^\dagger$ is continuous. Moreover, we also assume the following modulus of continuity,

$$\left\| \mathcal{G}^\dagger(a) - \mathcal{G}^\dagger(a') \right\|_{L^p(\mathbb{T}^2)} \leq \omega \left( \|a - a'\|_{H^\sigma(\mathbb{T}^2)} \right), \tag{3.2}$$

for some $p \in \{2, \infty\}$ and $0 \leq \sigma \leq r - 1$, and where $\omega : [0, \infty) \to [0, \infty)$ is a monotonously increasing function with $\lim_{y \to 0} \omega(y) = 0$. (3.2) is automatically satisfied if $\mathcal{X}^*$ is compact and $\mathcal{G}^\dagger$ is continuous. Under these assumptions, we have the following *universality theorem* for CNOs (2.3),

**Theorem 3.1.** *Let* $\sigma \in \mathbb{N}_0$ *and* $p \in \{2, \infty\}$ *as in (3.2),* $r > \max\{\sigma, 2/p\}$ *and* $B > 0$. *For any* $\varepsilon > 0$ *and any operator* $\mathcal{G}^\dagger$, *as defined above, there exists a CNO* $\mathcal{G}$ *such that for every* $a \in \mathcal{X}^*$ *with* $\|a\|_{H^r(D)} \leq B$ *it holds,*

$$\|\mathcal{G}^\dagger(a) - \mathcal{G}(a)\|_{L^p(D)} < \varepsilon. \tag{3.3}$$

In fact, we will prove a more general version of this theorem in **SM** B, where we also include additional source terms in the PDE (3.1).

## 4 Experiments.

**Training Details and Baselines.** We provide a detailed description of the implementation of CNO and the training (and test) protocol for CNO as well as all the baselines in **SM** C.1. To ensure a *level playing field* among all the tested models for each benchmark, we follow an *ensemble training procedure* by specifying a range for the underlying hyperparameters *for each model* and randomly selecting a subset of the hyperparameter space. For each such hyperparameter configuration, the corresponding models are trained on the benchmark and the configuration with smallest validation error is selected and the resulting test errors are reported, allowing us to identify and compare the *best performing* version of each model for every benchmark. We compare CNO with the following baselines: two very popular operator learning architectures, namely DeepONet (DON) [39] and FNO [33], a transformer based operator-learning architecture, i.e., Galerkin Transformer (GT) [8], feedforward neural network with with residual connections (FFNN) [18] and the very widely-used ResNet [18] and U-Net [50] architectures. [1]

Table 1: Relative median $L^1$ test errors, for both in- and out-of-distribution testing, for different benchmarks and models.

| | In/Out | FFNN | GT | UNet | ResNet | DON | FNO | CNO |
|---|---|---|---|---|---|---|---|---|
| **Poisson** | In | 5.74% | 2.77% | 0.71% | 0.43% | 12.92% | 4.98% | **0.21%** |
| **Equation** | Out | 5.35% | 2.84% | 1.27% | 1.10% | 9.15% | 7.05% | **0.27%** |
| **Wave** | In | 2.51% | 1,44% | 1.51% | 0.79% | 2.26% | 1.02% | **0.63%** |
| **Equation** | Out | 3.01% | 1.79% | 2.03% | 1.36% | 2.83% | 1.77% | **1.17%** |
| **Smooth** | In | 7.09% | 0.98% | 0.49% | 0.39% | 1.14% | 0.28% | **0.24%** |
| **Transport** | Out | 650.6% | 875.4% | 1.28% | 0.96% | 157.2% | 3.90% | **0.46%** |
| **Discontinuous** | In | 13.0% | 1.55% | 1.31% | **1.01%** | 5.78% | 1.15% | **1.01%** |
| **Transport** | Out | 257.3% | 22691.1% | 1.35% | 1.16% | 117.1% | 2.89% | **1.09%** |
| **Allen-Cahn** | In | 18.27% | 0.77% | 0.82% | 1.40% | 13.63% | **0.28%** | 0.54% |
| **Equation** | Out | 46.93% | 2.90% | 2.18% | 3.74% | 19.86% | **1.10%** | 2.23% |
| **Navier-Stokes** | In | 8.05% | 4.14% | 3.54% | 3.69% | 11.64% | 3.57% | **2.76%** |
| **Equations** | Out | 16.12% | 11.09% | 10.93% | 9.68% | 15.05% | 9.58% | **7.04%** |
| **Darcy** | In | 2.14% | 0.86% | 0.54% | 0.42% | 1.13% | 0.80% | **0.38%** |
| **Flow** | Out | 2.23% | 1.17% | 0.64% | 0.60% | 1.61% | 1.11% | **0.50%** |
| **Compressible** | In | 0.78% | 2.09% | 0.38% | 1.70% | 1.93% | 0.44% | **0.35%** |
| **Euler** | Out | 1.34% | 2.94% | 0.76% | 2.06% | 2.88% | 0.69% | **0.59%** |

**Representative PDE Benchmarks (RPB).** Given the lack of consensus on a *standard* set of benchmarks for machine learning of PDEs, we propose a *new suite of benchmarks* here. Our aims in this regard are to ensure i) sufficient diversity among the types of PDE considered, ii) access to training and test data is readily available for rapid prototyping and reproducibility and iii) *intrinsic computational complexity* of problem to make sure that it is worthwhile to design fast surrogates to classical PDE solvers for a particular problem. In other words, we will only consider PDEs where classical PDE solvers can *only* resolve the underlying operator on fine enough grids. To meet these requirements, we will not consider PDEs in one space dimension as traditional numerical methods are already quite fast for them. On the other hand, it is hard to obtain and store data for problems in three dimensions, due to computational expense of traditional methods. The *sweet spot* is achieved by considering PDEs in two space dimensions. We further restrict to Cartesian domains here as all models can be readily evaluated in this setting. In addition to including a diverse set of PDEs, we only consider problems with sufficiently *many spatial and temporal scales*. Otherwise, traditional numerical solvers can approximate the underlying PDE on very coarse grids and it is not worthwhile to design surrogates (see **SM** C.3.8 for a discussion in this context on a widely used Navier-Stokes benchmark). With these considerations in mind, we present the following subset of *Representative PDE Benchmarks* or **RPB**,

---

[1]The code can be found at `https://github.com/bogdanraonic3/ConvolutionalNeuralOperator`

**Poisson Equation.** This prototypical *linear elliptic PDE* is given by,

$$-\Delta u = f, \text{ in } D, \quad u|_{\partial D} = 0. \tag{4.1}$$

The solution operator $\mathcal{G}^{\dagger} : f \mapsto u$, maps the source term $f$ to the solution $u$. With source term,

$$f(x, y) = \frac{\pi}{K^2} \sum_{i,j=1}^{K} a_{ij} \cdot (i^2 + j^2)^{-r} \sin(\pi i x) \sin(\pi j y), \quad \forall (x, y) \in D, \tag{4.2}$$

with $r = -0.5$, the corresponding exact solution can be analytically computed (see SM C.3.1) and represents $K$- spatial scales. For training the models, we fix $K = 16$ in (4.2) and choose $a_{ij}$ to be i.i.d. uniformly distributed from $[-1, 1]$ (See **SM D** for a representation of the inputs and outputs of $\mathcal{G}^{\dagger}$). This *multiscale solution* needs fine enough grid size to be approximated accurately by finite element methods, fitting our complexity criterion for benchmarks. In addition to *in-distribution* testing , we also consider an *out-of-distribution* testing task by setting $K = 20$ in (4.2). This will enable us to evaluate the ability of the models to *generalize* to inputs (and outputs) with frequencies higher than those encountered during training.

**Wave Equation.** This prototypical *linear hyperbolic PDE* is given by

$$u_{tt} - c^2 \Delta u = 0, \text{ in } D \times (0, T), \quad u_0(x, y) = f(x, y), \tag{4.3}$$

with a constant propagation speed $c = 0.1$. The underlying operator $\mathcal{G}^{\dagger} : f \mapsto u(., T)$ maps the initial condition $f$ into the solution at the final time. If we consider initial conditions to be given by (4.2) with $r = 1$, then one can explicitly compute the exact solution (see **SM C.3.2**) to represent a *multiscale standing wave* with periodic pulsations (depending on $K$) in time. The training and *in-distribution* test samples are generated by setting $T = 5$, $K = 24$ and $a_{ij}$ to be i.i.d. uniformly distributed from $[-1, 1]$ (See **SM D** for input and output samples). For *out-of-distribution* testing, we change the exponent of decay of the modes in (4.2) to $r = 0.85$ and $K = 32$, in order to test the ability of the models to generalize to learn the effect of higher frequencies, than those present in the training data.

**Transport Equation.** The transport of scalar quantities of interest is modeled by PDE,

$$u_t + v \cdot \nabla u = 0, \quad u(t = 0) = f, \tag{4.4}$$

with a given velocity field and initial data $f$. The underlying operator $\mathcal{G}^{\dagger} : f \mapsto u(., T = 1)$ maps the initial condition $f$ into the solution at the final time. We set a constant velocity field $v = (v_x, v_y) = (0.2, 0.2)$ leading to solution $u(x, y, t) = f(x - v_x t, y - v_y t)$. Two different types of training data are considered, i.e., *smooth* initial data which takes the form of a radially symmetric Gaussian, with centers randomly and uniformly drawn from $(0.2, 0.4)^2$ and corresponding variance drawn uniformly from $(0.003, 0.009)$ and a *discontinuous* initial data in the form of the indicator function of radial disk with centers, uniformly drawn from $(0.2, 0.4)^2$ and radii uniformly drawn from $(0.1, 0.2)$ (See **SM C.3.3** for details and **SM D** for illustrations). For *out-of-distribution* testing in the smooth case, the centers of the Gaussian inputs are sampled uniformly from $(0.4, 0.6)^2$ and in the discontinuous case, the centers of the disk are drawn uniformly from $(0.4, 0.6)^2$, while keeping the variance and the radii, respectively, the same as that of *in-distribution testing*. This *out-of-distribution* task tests the model's ability to cope with input translation-equivariance.

**Allen-Cahn Equation.** It is a prototype for *nonlinear parabolic PDEs*,

$$u_t = \Delta u - \varepsilon^2 u(u^2 - 1), \tag{4.5}$$

with a reaction rate of $\varepsilon = 220$ and underlying operator $\mathcal{G}^{\dagger} : f \mapsto u(., T)$, mapping initial conditions $f$ to the solution $u$ at a final time $T = 0.0002$. The initial conditions for training and *in-distribution* testing are of the form (4.2), with $r = 1$ and $K = 24$ and coefficients $a_{ij}$ drawn uniformly from $[-1, 1]$. For *out-of-distribution* testing, we set $K = 16$ and randomly select the initial decay $r$, uniformly from the range $[0.85, 1.15]$ of the modes in (4.2), which allows us to test the ability of the model to generalize to different dynamics of the system. Both training and test data are generated by using a finite difference scheme [62] on a grid at $64^2$ resolution (see **SM D** for illustrations).

**Navier-Stokes Eqns.** These PDEs model the motion of incompressible fluids by,

$$u_t + (u \cdot \nabla)u + \nabla p = \nu \Delta u, \quad \text{div } u = 0, \tag{4.6}$$

in the torus $D = \mathbb{T}^2$ with periodic boundary conditions and viscosity $\nu = 4 \times 10^{-4}$, only applied to high-enough Fourier modes (those with amplitude $\geq 12$) to model fluid flow at *very high Reynolds-number*. The solution operator $\mathcal{G}^\dagger : f \mapsto u(., T)$, maps the initial conditions $f : D \to \mathbb{R}^2$ to the solution at final time $T = 1$. We consider initial conditions representing the well-known *thin shear layer* problem [3, 27] (See **SM** C.3.5 for details), where the shear layer evolves via vortex shedding to a complex distribution of vortices (see **SM** D for samples). The training and *in-distribution testing* samples are generated, with a spectral viscosity method [27], from an initial sinusoidal perturbation of the shear layer [27], with layer thickness $\rho = 0.1$ and 10 perturbation modes, each sampled uniformly from $[-1, 1]$. For *out-of-distribution* testing, the layer thickness is reduced to $\rho = 0.09$ and the layers are shifted up in the domain to test the ability of the models to *generalize* to a flow regime with an increased number and different locations of the shed vortices.

**Darcy flow.** The steady-state Darcy flow is described by the second order linear elliptic PDE,

$$-\nabla \cdot (a\nabla u) = f, \text{ in } D, \quad u|_{\partial D} = 0, \tag{4.7}$$

where $a$ is the diffusion coefficient and $f$ is the forcing term. We set the forcing term to $f = 1$. The solution operator is $\mathcal{G}^\dagger : a \mapsto u$, where the input is the diffusion coefficient $a \sim \psi\#\mu$, with $\mu$ being a Gaussian Process with zero mean and squared exponential kernel

$$k(x, y) = \sigma^2 \exp\left(\frac{|x - y|^2}{l^2}\right), \quad \sigma^2 = 0.1. \tag{4.8}$$

We chose the length scale $l = 0.1$ for the in-distribution testing and $l = 0.05$ for the out-of-distribution testing. The mapping $\psi : \mathbb{R} \to \mathbb{R}$ takes the value 12 on the positive part of the real line and 3 on the negative part. The push-forward measure is defined pointwise. The experimental setup is the same as the one presented in [33].

**Flow past airfoils.** We model this flow by the compressible Euler equations,

$$u_t + \text{div } F(u) = 0, \ u = [\rho, \rho v, E]^\perp, \ F = [\rho v, \rho v \otimes v + p\mathbf{I}, (E + p)]v]^\perp, \tag{4.9}$$

with density $\rho$, velocity $v$, pressure $p$ and total Energy $E$ related by an ideal gas equation of state. The airfoils we consider are described by perturbing the shape of a well-known RAE2822 airfoil [40] by Hicks-Henne Bump functions [42] (see **SM** C.3.7). Freestream boundary conditions are imposed and the solution operator maps the shape function onto the steady state density distribution (see **SM** D for samples) and training data are obtained with a compressible flow solver (NUWTUN) with shapes corresponding to 20 bump functions, with coefficients sampled uniformly from $[0, 1]$. *Out-of-distribution* testing is performed with 30 bump functions.

**Results.** The test errors, for both *in-distribution* and *out-of-distribution* testing for all the models on the **RPB** benchmarks are shown in Table 1. Starting with the *in-distribution* results, we see that among the baselines, FNO clearly outperforms both FFNN and DeepONet on all the **RPB** benchmarks as well as the Galerkin Transformer on all except the Poisson test case. On the other hand, the convolution-based U-Net and ResNet models are quite competitive vis-a-vis FNO, with comparable performances on most benchmarks, while outperforming FNO by a factor of $7 - 9$ for the Poisson test case. This already indicates that convolution-based architectures can perform very well. Moreover, we observe from Table 1 that CNO is the best performing architecture on every task except Allen-Cahn. It readily outperforms FNO, for instance by almost a factor of 20 on the Poisson test case but more moderately but significantly on other tasks. It also outperforms U-Net and ResNet on all tasks considered here, emerging as the best-performing model over these benchmarks.

**Out-of-Distribution Testing.** This trend is further reinforced when we consider *out-of-distribution* testing. CNO generalizes well to unseen data in a *zero-shot* mode, with test errors increasing by approximately a factor of 2, at most (with Allen-Cahn being an outlier) and still outperforms the baselines significantly in all cases other than Allen-Cahn, where FNO generalizes the best. FNO shows decent generalization for most problems but generalizes poorly on the transport problems. This can be attributed to its lack of translation equivariance, in contrast to U-Net, ResNet and CNO. Finally, the lack of translation invariance severely limits the out-of-distribution generalization performance of DeepONet, FFNN and Galerkin Transformer models on transport problems.

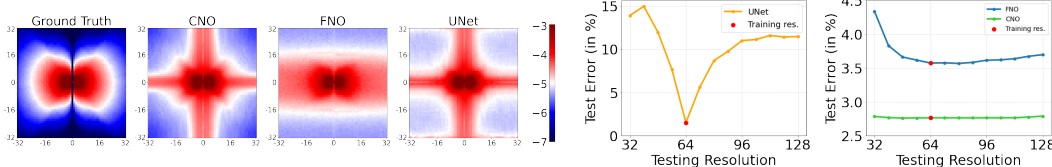

Figure 2: Thin Shear Layer Left: Averaged *logarithmic* amplitude spectra comparing Ground Truth, CNO, FNO and UNet. Right: Test error vs. Resolution for UNet, FNO and CNO.

**Resolution Invariance.** We select three of the best-performing models (U-Net, FNO and CNO) and highlight further differences between them for the Navier-Stokes test case (see also **SM** C). To this end, we start with Figure 2 (left), where we present the averaged (log) spectra for the ground truth (reference solution computed with the spectral viscosity method) and those computed with CNO, FNO and U-Net. We observe from this figure that i) the spectrum of the exact solution is very rich with representations of many frequencies, attesting to the *multi-scale* nature of the underlying problem and ii) there are significant differences in how CNO and FNO approximate the underlying solution in Fourier space. In particular, the decay in CNO's spectrum is more accurate. On the other hand, the FNO spectrum is amplified along the horizontal axis, possibly on account of *aliasing errors* that add incorrect frequency content. The U-Net spectra are similar to that of CNO but with high-frequency modes being amplified, which leads to a higher test error. Next, in Figure 2 (right), we compare CNO, FNO and U-Net vis-a-vis the metric of *how the test error varies across resolutions*, see **SM**C.4 for details, which is an important aspect for robust operator learning that been highlighted in [33, 25], see also [2] for a discussion with respect to representation equivalent neural operators or ReNOs. We find from Figure 2 (right) that for the Navier-Stokes benchmark, the FNO error is not *invariant* with respect to resolution, with an increase in error of up to $25\%$ on lower-resolutions as well as a more modest but noticeable increase of $10\%$ on resolutions, higher than the training resolution of $64^2$, implying that FNO is neither able to perform alias-error free super- or sub-resolution in this case, see also [2] for a detailed discussion on the resolution-invariance of FNO. Similarly, the increase of U-Net test error with respect to varying resolutions is even more pronounced, with a maximum increase of a factor of 3, indicating neither FNO nor U-Net are resolution (representation) equivalent in this case. In contrast, CNO error is invariant with respect to test resolution, verifying that it respects *continuous-discrete equivalence*. Further ablation studies for CNO are presented in **SM** C.5.

**Efficiency.** In order to further compare CNO with FNO, which is the most widely used neural operator these days, we illustrate not just the performance in terms of errors but also terms in computational efficiency. To this end, in Figure 3 (left), we plot the size (total number of parameters) vs validation error for the Navier-Stokes benchmark for all FNO and CNO models that were considered in the model selection procedure to observe that for the same model size, CNO models led to significantly smaller validation errors. This resulted in smaller errors for the same per-epoch training time (Figure 3 (center)) for CNO vis-a-vis FNO. As observed in Figure 3 (center), this also implies that one of the best-performing CNO models, with a significantly smaller error than the best-performing FNO model is also almost twice as fast to train, thus showcasing the computational efficiency of CNOs.

**Scaling laws.** A very important aspect of modern deep learning is to evaluate how the performance of models scales with respect to data (and model size). To investigate this, we focus on the Navier-Stokes benchmark and present test error vs. (log of) number of training samples for GT, FNO and CNO in Figure 3 (right) to observe that the errors decrease consistently with number of samples. To quantify the rates of decrease, we fit **power laws** of the form $E = (N_0/N)^{-r}$, with $E$ being the test error, $N$ number of samples and $N_0$ normalized to be the number of samples required to reach errors of $1\%$ to obtain that CNO attains a much faster convergence rate ($r = 0.37$) compared to FNO ($r = 0.28$) and GT ($r = 0.27$). This implies that CNO will require $N_0 \approx 14.3K$ samples to attain $1\%$ error which is 4-times less than FNO ($N_0 \approx 60.1K$) and almost 10-times less than GT ($N_0 \approx 133.2K$), highlighting the data efficiency of CNO. Finally, from Figure 3 (right), we also observe that a smaller CNO model (with $0.82$ M parameters) scales worse (with $r = 0.3$) compared to the best performing CNO model with $7.8M$ parameters, illustrating that model size could be a *bottleneck* for data scaling and larger models are required to scale with data.

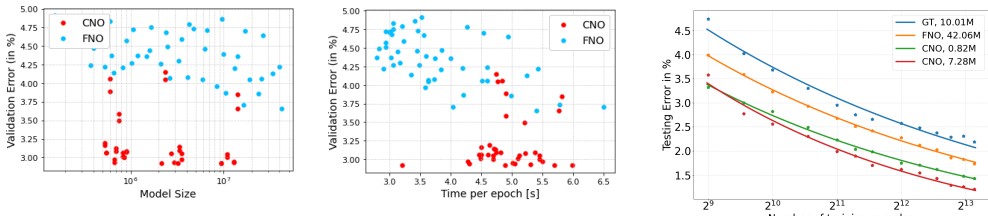

Figure 3: Navier-Stokes Benchmark: Test Error (Y-axis) vs. Model size (Left) and per-epoch training time (Center) for all the FNO and CNO models tested. The best-performing models are highlighted as is a small-scale yet efficient CNO model. Right: Log of Error vs. Log of Number of training samples for GT, FNO, CNO and a small-scale CNO model.

# 5 Discussion.

**Summary.** We propose CNO, a *novel* convolution-based architecture for learning operators. The basic design principle was to enforce a form of *continuous-discrete equivalence* in order to genuinely learn the underlying operators, rather than discrete representations of them. To this end, we modified the elementary operators of convolution, up-and downsampling and particularly nonlinear activations to realize CNO as a representation equivalent neural operator or ReNO in the sense of [2]. We also prove a universality theorem to show that CNO can approximate a large class of operators arising in PDEs to desired accuracy. A novel suite of experiments, termed as representative PDE benchmarks (**RPB**), encompassing a wide variety of PDEs, with multiple scales in the corresponding solutions, which are hard to resolve with traditional numerical methods, is also proposed and the model tested on them. We demonstrate that CNO outperforms the baselines, including FNO, significantly on most benchmarks. This also holds for the considered *out-of-distribution* testing tasks which ascertain the ability of the models to *generalize* to unseen data in a *zero-shot* manner.

**Comparison to Related Work.** We emphasize that our construction of CNO follows the theoretical prescription of recent paper [2] on enforcing structure preserving *continuous-discrete equivalence*. CNO is a *representation equivalent neural operator*, with respect to spaces of bandlimited functions, in the sense of [2]. Another motivating work for us is [22], see also [58], where the authors modify CNNs to eliminate (or reduce) aliasing errors in the context of image generation. We adapt the construction of [22] to our setting and deploy the resulting architecture in a very different context from that of [22], namely that of operator learning for PDEs rather than image generation. Moreover, we also instantiate CNO with a very different operator UNet architecture than that proposed in [22]. We would also like to mention related work on using CNNs for solving PDEs such as [1, 16] and emphasize that in contrast to CNO, they lack suitable notions of continuous-discrete equivalence. Finally comparing CNO to the widely used FNO model, we observe that unlike FNO which can fail to enforce CDE (see [14, 2] and Figure 2(right)), CNO preserves continuous-discrete equivalence. Moreover, the convolution operator in CNO is local in space, in contrast to convolution in Fourier space for FNO. The detailed empirical comparison presented here demonstrates CNO can outperform FNO and other baselines on many different metrics namely performance, computational efficiency, resolution invariance, out-of-distribution generalization as well as data scaling, paving the way for its widespread applications in science and engineering.

**Limitations and Future Work.** We have presented CNO for operators on an underlying two-dimensional Cartesian domain. The extension to three-space dimensions is conceptually straightforward but computationally demanding. Similarly, extending to non-Cartesian domains will require some form of transformation maps between domains, for instance reworking those suggested for FNO in [32, 55] can be readily considered. Adapting CNO to approximate trajectories (in time) of time-dependent PDEs, for instance by employing it in an *auto-regressive* manner, is another possible extension of this paper. At the level of theoretical results, we believe that the generic framework of [11] can be adapted to show that not only does CNO approximate a large class of PDEs universally, it does so without incurring any curse of dimensionality, as shown for DeepONets in [26] and FNOs in [24]. Finally, adapting and testing CNO for learning operators, beyond the forward solution operator of PDEs is also interesting. One such direction lies in efficiently approximating *PDE inverse problems*, for instance those considered in [44].

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
