**Supplementary Material for:**
Convolutional Neural Operators for Robust and Accurate Learning of PDEs.

# Table of Contents

# A    Technical Details for Section 2 of main text.

## A.1    Approximation of Operators mapping between Sobolev spaces by operators mapping between spaces of bandlimited functions.

We prove that one can approximate any continuous operator $\mathcal{G}^\dagger : \mathcal{X} \to \mathcal{Y}$ (as introduced in Section 2) of the main text by an operator mapping between spaces of bandlimited functions to arbitrary accuracy. We obtain this result by discarding the high-frequency components, e.g. higher than frequency $w$, of both the input and output of $\mathcal{G}^\dagger$. This can be performed by a Fourier projection $P_w$. For orthogonal Fourier projections and also trigonometric polynomial interpolation [19, 24] the following result on the accuracy of the projection holds,

**Lemma A.1.** *Given $\sigma, r \in \mathbb{N}_0$ with $r > d/2$ and $r \geq \sigma$, and $f \in C^r(\mathbb{T}^d)$ it holds for every $w \in \mathbb{N}$ that,*

$$\|f - P_w(f)\|_{H^\sigma(\mathbb{T}^d)} \leq C(r,d)w^{-(r-\sigma)} \|f\|_{H^r(\mathbb{T}^d)}, \tag{A.1}$$

*for a constant $C(r,d) > 0$ that only depends on $r$ and $d$.*

Using this result, we show that by discarding the high frequencies of the input and output of $\mathcal{G}^\dagger$ one can approximate $\mathcal{G}^\dagger$ to arbitrary accuracy by choosing an appropriate frequency cutoff.

**Lemma A.2.** *For any $\varepsilon, B > 0$ there exist $w \in \mathbb{N}$ such that $\left\|\mathcal{G}^\dagger(a) - P_w\mathcal{G}^\dagger(P_wa)\right\|_{L^2(D)} \leq \varepsilon$ for all $a \in H^r(D)$ with $\|a\|_{H^r(D)} \leq B$.*

*Proof.* We follow [24] and use Lemma A.1 repeatedly together with the stability of $\mathcal{G}^\dagger$ (2.1) to obtain,

$$
\begin{aligned}
\left\| \mathcal{G}^\dagger(a) - P_w \mathcal{G}^\dagger(P_w a) \right\|_{L^2} &\leq \left\| \mathcal{G}^\dagger(a) - P_w \mathcal{G}^\dagger(a) \right\|_{L^2} + \left\| P_w \mathcal{G}^\dagger(a) - P_w \mathcal{G}^\dagger(P_w a) \right\|_{L^2} \\
&\lesssim w^{-r} \left\| \mathcal{G}^\dagger(a) \right\|_{H^r} + \left\| \mathcal{G}^\dagger(a) - \mathcal{G}^\dagger(P_w a) \right\|_{L^2} \\
&\lesssim w^{-r} \|\mathcal{G}^\dagger\|_{op} \|a\|_{H^r} + \omega(\|a - P_w a\|_{H^\sigma}) \\
&\lesssim w^{-r} \|\mathcal{G}^\dagger\|_{op} \|a\|_{H^r} + \omega(C w^{-(r-\sigma)} \|a\|_{H^r}).
\end{aligned}
\tag{A.2}
$$

It follows immediately that for large enough $w$,

$$
\sup_{\|a\|_{H^r} \leq B} \left\| \mathcal{G}^\dagger(a) - P_w \mathcal{G}^\dagger(P_w a) \right\|_{L^2} \leq \varepsilon.
\tag{A.3}
$$

This proves the statement of the lemma. $\qquad\square$

Given that both $P_w a \in \mathcal{B}_w(D)$ and $P_w \mathcal{G}^\dagger(P_w a) \in \mathcal{B}_w(D)$, a consequence of the above lemma is the existence of an operator $\mathcal{G}^* : \mathcal{B}_w(D) \to \mathcal{B}_w(D) : a \mapsto P_w \mathcal{G}^\dagger(a)$ that can approximate $\mathcal{G}^\dagger$ arbitrarily well. It follows from the lemma and its proof that $\left\| \mathcal{G}^\dagger - \mathcal{G}^* \right\|_{op} \leq \varepsilon$, where the operators are considered as mappings from and to $\mathcal{B}_w(D) \cap H^r(D)$ equipped with the $H^r(D)$-norm.

## A.2 Continuous-Discrete Equivalence for Operator $\mathcal{G}^*$ from Section 2.1

For every $w > 0$, we denote by $\mathcal{B}_w(\mathbb{R}^2)$ the space of multivariate bandlimited functions

$$
\mathcal{B}_w(\mathbb{R}^2) = \{f \in L^2(\mathbb{R}^2) : \operatorname{supp} \widehat{f} \subseteq [-w, w]^2\},
$$

where $\widehat{f}$ denotes the Fourier transform on $L^1(\mathbb{R})$

$$
\widehat{f}(\xi) := \int_{\mathbb{R}} f(x) e^{-2\pi i x \xi}\, dx, \qquad \xi \in \mathbb{R},
$$

which extends to $L^2(\mathbb{R})$ by a classical density argument. The set $\Psi_w = \{\operatorname{sinc}(2w x_1 - m) \cdot \operatorname{sinc}(2w x_2 - n)\}_{m,n \in \mathbb{Z}}$ constitutes an orthonormal basis for $\mathcal{B}_w(\mathbb{R}^2)$. The bounded operator

$$
T_{\Psi_w} : \ell^2(\mathbb{Z}^2) \to \mathcal{B}_w(\mathbb{R}^2), \quad T_{\Psi_w}(c_{m,n}) = \sum_{m,n \in \mathbb{Z}} c_{m,n} \operatorname{sinc}(2w \cdot -m) \cdot \operatorname{sinc}(2w \cdot -n),
$$

which reconstructs a function from its basis coefficients, is called *synthesis operator*, and its adjoint

$$
T_{\Psi_w}^* : \mathcal{B}_w(\mathbb{R}^2) \to \ell^2(\mathbb{Z}^2), \quad T_{\Psi_w}^* f = \left\{ f\left( \frac{m}{2w}, \frac{n}{2w} \right) \right\}_{m,n \in \mathbb{Z}},
$$

which extract basis coefficients from an underlying function, is called *analysis operator*. Every bandlimited function can be uniquely and stably recovered from its sampled values $\left\{ f\left( \frac{m}{2w}, \frac{n}{2w} \right) \right\}_{m,n \in \mathbb{Z}}$ via the reconstruction formula

$$
f(x_1, x_2) = T_{\Psi_w} T_{\Psi_w}^* f(x_1, x_2) = \sum_{m,n \in \mathbb{Z}} f\left( \frac{m}{2w}, \frac{n}{2w} \right) \operatorname{sinc}(2w x_1 - m) \cdot \operatorname{sinc}(2w x_2 - n), \quad \text{(A.4)}
$$

and we say that there is a *continuous-discrete equivalence (CDE)* between $f$ and its samples $\left\{ f\left( \frac{m}{2w}, \frac{n}{2w} \right) \right\}_{m,n \in \mathbb{Z}}$. More in general, every bandlimited function $f \in \mathcal{B}_w(\mathbb{R}^2)$ can be uniquely and stably recovered from its values $\{f(mT, nT)\}_{m,n \in \mathbb{Z}}$ if the *sampling rate* or reciprocal of grid size, $1/T$ is greater or equal than the *Nyquist rate* $2w$. This simply follows from the fact that $\mathcal{B}_w(\mathbb{R}^2) \subset \mathcal{B}_{w'}(\mathbb{R}^2)$ for every $w' > w$. On the contrary, reconstructing $f \in \mathcal{B}_w$ at a sampling rate below the Nyquist rate, i.e. $1/T < 2w$, results in a non-zero value for the *aliasing error function*:

$$
\varepsilon(f) = f - T_{\Psi_{\frac{1}{2T}}} T_{\Psi_{\frac{1}{2T}}}^* f,
$$

and the associated *aliasing error* $\|\varepsilon\|_2$ (cfr. Definition 2.1 in [2]).

Let $\mathcal{G}^*$ be a (possibly) non-linear operator between band-limited spaces, i.e. $\mathcal{G}^* : \mathcal{B}_w(\mathbb{R}^2) \to \mathcal{B}_{w'}(\mathbb{R}^2)$, for some $w, w' > 0$. As argued in [2], the concepts of continuous-discrete equivalence

(CDE) and aliasing error can be adapted to the operator $\mathcal{G}^*$. The continuous operator $\mathcal{G}^*$ is uniquely determined by a map $\mathfrak{g}_{\Psi_w, \Psi_{w'}} : \ell(\mathbb{Z}^2) \to \ell^2(\mathbb{Z}^2)$ if the aliasing error operator

$$\varepsilon = \mathcal{G}^* - T_{\Psi_{w'}} \circ \mathfrak{g}_{\Psi_w, \Psi_{w'}} \circ T_{\Psi_w}^* \tag{A.5}$$

is identically zero, and we say that $\mathcal{G}^*$ and $\mathfrak{g}_{\Psi_w, \Psi_{w'}}$ satisfy a continuous-discrete equivalence (cfr. Definition 3.1 in [2]). Equivalently, the diagram

$$
\begin{array}{ccc}
\mathcal{B}_w & \xrightarrow{\mathcal{G}^*} & \mathcal{B}_{w'}, \\
\downarrow{\scriptstyle T_{\Psi_w}^*} & \uparrow{\scriptstyle T_{\Psi_{w'}}} & \\
\ell^2(\mathbb{Z}^2) & \xrightarrow{\mathfrak{g}_{\Psi_w, \Psi_{w'}}} & \ell^2(\mathbb{Z}^2)
\end{array}
$$

commutes, i.e. the black and the blue directed paths in the diagram lead to the same result. In this latter case, since $T_{\Psi_w}^* \circ T_{\Psi_w}$ is the identity operator from $\ell^2(\mathbb{Z}^2)$ onto itself, equation (A.5) forces the discretization $\mathfrak{g}_{\Psi_w, \Psi_{w'}}$ to be defined as

$$\mathfrak{g}_{\Psi_w, \Psi_{w'}} = T_{\Psi_{w'}}^* \circ \mathcal{G}^* \circ T_{\Psi_w}, \tag{A.6}$$

i.e. the diagram

$$
\begin{array}{ccc}
\mathcal{B}_w & \xrightarrow{\mathcal{G}^*} & \mathcal{B}_{w'} \\
\uparrow{\scriptstyle T_{\Psi_w}} & & \downarrow{\scriptstyle T_{\Psi_{w'}}^*} \\
\ell^2(\mathbb{Z}^2) & \xrightarrow{\mathfrak{g}_{\Psi_w, \Psi_{w'}}} & \ell^2(\mathbb{Z}^2)
\end{array}
$$

also commutes. In other words, once we fix the discrete representations associated to the input and output functions, there exists a unique way to define a discretization $\mathfrak{g}_{\Psi_w, \Psi_{w'}}$ that is consistent with the continuous operator $\mathcal{G}^*$ and this is given by (A.6). In practice, we may have access to different discrete representations of the input and output functions, e.g. point samples evaluated on different grids, which in the theory amounts to a change of reference systems in the function spaces. For instance, sampling a function $f \in \mathcal{B}_w$ on a finer grid $\left\{ \left( \frac{m}{2\overline{w}}, \frac{n}{2\overline{w}} \right) \right\}_{m,n \in \mathbb{Z}}$, $\overline{w} > w$, amounts to representing the function $f$ with respect to the system $\Psi_{\overline{w}} = \{ \operatorname{sinc}(2\overline{w}x_1 - m) \cdot \operatorname{sinc}(2\overline{w}x_2 - n) \}_{m,n \in \mathbb{Z}}$, which constitutes an orthonormal basis for $\mathcal{B}_{\overline{w}} \supset \mathcal{B}_w$. Then, one can define the associated CDE discretization $\mathfrak{g}_{\Psi_{\overline{w}}, \Psi_{\overline{w}'}}$ as in (A.6), and by equation (A.5), one readily obtains the change of basis formula

$$\mathfrak{g}_{\Psi_{\overline{w}}, \Psi_{\overline{w}'}} = T_{\Psi_{\overline{w}'}}^* \circ T_{\Psi_{w'}} \circ \mathfrak{g}_{\Psi_w, \Psi_{w'}} \circ T_{\Psi_w}^* \circ T_{\Psi_{\overline{w}}}, \tag{A.7}$$

see also Remark 3.5 in [2] for a more general *change of frame* formula. Finally, all the above concepts generalize to every pair of frame sequences $(\Psi, \Phi)$ that span respectively the input and output function spaces, and we refer to [2] for a complete exposition. Appendix A.2 can be adapted to bandlimited periodic functions, i.e. periodic functions with a finite number of non-zero Fourier coefficients, with the Dirichlet kernel as a counterpart of the sinc function, see [57, Section 5.5.2] for further details.

### A.3 Multi-channel versions of elementary operators for CNO (2.3)

In this section, we will define *multi-channel* versions of the elementary mappings which define CNO (2.3). Note that the single-channel versions were defined in the main text.

**Convolution Operator.** In the multi-channel settings, discrete kernels $K_w$ are defined on the $d_{in} \times d_{out} \times s^2$ uniform grids on $D$, where $d_{in}$ is the number of input channels and $d_{out}$ is the number of output channels. Formally, the kernels are defined as

$$K_{w,cl} = \sum_{i,j=1}^{k} k_{ij,cl} \cdot \delta_{z_{ij}}.$$

where $c$ is the channel index in the input space, while $l$ is the channel index in the output space. Each pair of channels defines corresponding single-channel convolution operation $\mathcal{K}_{w,cl} : \mathcal{B}_w(D) \to \mathcal{B}_w(D)$. For $a \in \mathcal{B}_w(D, \mathbb{R}^{d_{in}})$, the multi-channel convolution operation $\mathcal{K}_w$ is defined as

$$\left( \mathcal{K}_w a(x) \right)_l = \sum_{c=1}^{d_{in}} \mathcal{K}_{w,cl} \, a_c(x), \quad l = 1 \ldots d_{out}.$$

**Upsampling and Downsampling Operators.** To upsample a signal $a \in \mathcal{B}_w(D, \mathbb{R}^d)$ with $d$ channels from the bandlimit $w > 0$ to the bandlimit $\overline{w} > w$, one should apply the single-channel upsampling operator $\mathcal{U}_{w,\overline{w}}$ to each individual channel of the input signal, independently. Formally, for $a \in \mathcal{B}_w(D, \mathbb{R}^d)$, the multi channel upsampling $\mathcal{U}_{w,\overline{w}} : \mathcal{B}_w(D, \mathbb{R}^d) \to \mathcal{B}_{\overline{w}}(D, \mathbb{R}^d)$ is defined as

$$\left( \mathcal{U}_{w,\overline{w}} a(x) \right)_c = a_c(x), \quad \forall x \in D, \quad c = 1 \dots d.$$

The downsampling operator of a signal $a \in \mathcal{B}_w(D, \mathbb{R}^d)$ from the bandlimit $w > 0$ to the bandlimit $\underline{w} < w$ is defined in a similar manner (independent applications of the single-channel downsampling operators).

**Activation layer.** The multi-channel version of the activation layer, namely $\Sigma_{w,\overline{w}} : \mathcal{B}_w(D, \mathbb{R}^d) \to \mathcal{B}_w(D, \mathbb{R}^d)$, is realized by applying the single-channel activation layer to each of the $d$ channels, independently.

## A.4 Discrete operators for CNO

In this section, we will define the *discrete versions* of the elementary mappings in (2.3). Given a *discrete*, multi-channel signal $a_s \in \mathbb{R}^{s \times s \times d}$ on $s \times s \times d$ uniform grid, we will use the notation $a_s[i, j, c]$ to refer to the $(i, j)$-th coordinate of the $c$-th channel of the signal, where $i, j = 1 \dots s$ and $c = 1 \dots d$.

**Convolution operator.** Assume that instead of a continuous, single-channel signal $a \in \mathcal{B}_w(D)$, one has an access only to its sampled version $a_s \in \mathbb{R}^{s \times s}$ on $s \times s$ uniform grid on $D$. Assume that $a_s$ is to be convolved with a *discrete* kernel $K_w \in \mathbb{R}^{k \times k}$ with $k = 2\widehat{k} + 1$. Let $\widehat{a}_s \in \mathbb{R}^{s + 2\widehat{k} \times s + 2\widehat{k}}$ be an extended version of $a_s$ obtained by circular-padding or zero-padding of $a_s$. The discrete, single-channel convolution $\mathcal{K}_s : \mathbb{R}^{s \times s} \to \mathbb{R}^{s \times s}$ of the signal $a_s$ and the kernel $K_w$ is given by

$$\mathcal{K}_s(a_s) = (a_s \star K_w)[i, j] = \sum_{m,n=-\widehat{k}}^{\widehat{k}} K_w[m, n] \cdot \widehat{a}_s[i - m, j - n], \quad i, j = 1 \dots s,$$

where indices of $\widehat{a}_s$ outside the range $1 \dots s$ correspond to the padded samples. By performing the convolution in a described way, we ensure that the input and the output signals have the same spatial dimension $s \times s$.

Let $a_s \in \mathbb{R}^{s \times s \times d_{in}}$ be a discrete, multi-channel signal and $K_w \in \mathbb{R}^{k \times k \times d_{in} \times d_{out}}$ a discrete kernel with $k = 2\widehat{k} + 1$. The multi-channel convolution of $a_s$ and $K_w$ is defined by

$$(a_s \star K_w)[i, j, l] = \sum_{m,n=-\widehat{k}}^{\widehat{k}} \sum_{c=1}^{d_{in}} K_w[m, n, c, l] \cdot \widehat{a}_s[i - m, j - n, c], \quad i, j = 1 \dots s,$$

where $l$ corresponds to the index of the output channel and $c$ to the index of the input channel.

**Upsampling and Downsampling Operators.** In this section, we will define the discrete upsampling and downsampling operators. For $w > 0$, let $h_w$ be the interpolation *sinc* filter defined in 2.5. For a discrete, single-channel signal $a_s \in \mathbb{R}^{s \times s}$, let $(\widetilde{a}_s[n])_{n \in \mathbb{Z}}$ be its periodic extension into infinite length. In other words, $\widetilde{a}_s[n] = a_s[n \bmod s]$ for $n \in \mathbb{Z}$. The discrete upsampling $\mathcal{U}_{s,N} : \mathbb{R}^{s \times s} \to \mathbb{R}^{Ns \times Ns}$ by an *integer factor* $N \in \mathbb{N}$ of the signal $a_s \in \mathbb{R}^{s \times s}$ is done in *two* phases:

1. First step is to increase the number of samples of the signal $a_s$ from $s^2$ to $(Ns)^2$. One transforms the signal $a_s$ into the signal $a_{s,\uparrow Ns}$ obtained by separating each two signal samples of $a_s$ with $N - 1$ zero-valued samples. In other words, it holds that $a_{s,\uparrow Ns} \in \mathbb{R}^{Ns \times Ns}$ and

$$a_{s,\uparrow Ns}[i, j] = \mathbb{1}_S(i) \cdot \mathbb{1}_S(j) \cdot a_s[i \bmod s, j \bmod s], \quad i, j = 1 \dots Ns,$$

where $S = \{1, s + 1, \dots (N - 1)s + 1\}$ and $\mathbb{1}_S$ is the indicator function.

2. Second step is to convolve the periodic extension of $a_{s,\uparrow Ns}$ with the $h_{s/2}$ interpolation filter to eliminate high frequency components. The upsampled signal is formally obtained by

$$\mathcal{U}_{s,N}(a_s)[i,j] = \sum_{n,m\in\mathbb{Z}} \widetilde{a}_{s,\uparrow Ns}[n,m] \cdot h_{s/2}(is-ns, js-ms), \quad i,j = 1\ldots Ns.$$

The discrete downsampling $\mathcal{D}_{s,N} : \mathbb{R}^{s\times s} \to \mathbb{R}^{s/N \times s/N}$ by an *integer factor* $N \in \mathbb{N}$ of the signal $a_s \in \mathbb{R}^{s\times s}$ is also done in *two* phases (under the assumption that $s/N \in \mathbb{N}$):

1. First step is to convolve the periodic extension of $a_s$ with the $h_{s/(2N)}$ interpolation filter to eliminate high frequency content. Formally, the first step is defined by

$$a_{s,s/N}[i,j] = \sum_{n,m\in\mathbb{Z}} \widetilde{a}_s[n,m] \cdot h_{s/(2N)}(is-ns, js-ms), \quad i,j = 1\ldots s/N.$$

2. Second step is to decrease the sampling rate of $a_{s,N/s}$ by keeping *every* $N-th$ sample of the signal. The downsampled signal is formally defined by

$$\mathcal{D}_{s,N}(a_s)[i,j] = a_{s,N/s}[(i-1)s+1, (j-1)s+1], \quad i,j = 1\ldots s/N.$$

Multi-channel discrete upsampling and downsampling are performed by independent applications of the corresponding single-channel operators.

Since perfect filters $h_w$ have infinite impulse response and cause ringing artifacts around high-gradient points (e.g. discontinuities) due the Gibbs phenomenon, one usually uses *windowed-sinc* filters in the implementation. We will describe these filters later in the text (see C.1.4)

**Activation layer.** Given the definitions of the discrete operators, the discrete, single-channel activation layer is defined as

$$\Sigma_s : \mathbb{R}^{s\times s} \to \mathbb{R}^{s\times s}, \quad \Sigma_s(a_s) = \mathcal{D}_{s,N} \circ \sigma \circ \mathcal{U}_{s,N}(a_s),$$

where $\sigma : \mathbb{R} \to \mathbb{R}$ is an activation function applied point-wise and $N \in \mathbb{N}$ is a fixed constant. In our experiments, we noticed that $N = 2$ is sufficient for accurate predictions. The multi-channel activation layer is performed by independent applications of the single-channel activation layer.

## A.5 Proof of Proposition 2.1 of Main Text

We use the same notation as in Section 2 and Appendix A.2. The layers of a convolutional neural operator (2.3) are given by,

$$v_{l+1} = \mathcal{P}_l \circ \Sigma_l \circ \mathcal{K}_l(v_l), \quad 0 \le l \le L-1, \tag{A.8}$$

Hence, they consist of three elementary mappings between spaces of bandlimited functions, i.e., $\mathcal{K}_l$ is a convolution operator, $\Sigma_l$ is a non-linear operator whose definition depends on the choice of an activation function $\sigma : \mathbb{R} \to \mathbb{R}$, and $\mathcal{P}_l$ is a projection operator. We now show that CNO layers, whose discrete versions are outlined in the previous section, respect equation (A.6) and consequently CNOs are Representation equivalent Neural Operators (ReNOs) in the sense of [2, Definiton 3.4] and [2, Remark 3.5]. We recall that the convolutional operator appearing in (A.8) takes the form

$$\mathcal{K}_w f(x) = \sum_{m,n=-k}^{k} k_{m,n} f(x-z_{m,n}), \quad x \in \mathbb{R},$$

for some $w > 0$, where $k \in \mathbb{N}$, $k_{m,n} \in \mathbb{C}$ and $z_{m,n} = \left\{\left(\frac{m}{2w}, \frac{n}{2w}\right)\right\}_{m,n\in\mathbb{Z}}$. By definition, $\mathcal{K}_w$ is a well-defined operator from $\mathcal{B}_w(\mathbb{R}^2)$ into itself. Moreover, its discretized version is defined by the mapping

$$\left\{f\left(\frac{m}{2w}, \frac{n}{2w}\right)\right\}_{m,n\in\mathbb{Z}} \to \left\{\mathcal{K}_w f\left(\frac{m}{2w}, \frac{n}{2w}\right)\right\}_{m,n\in\mathbb{Z}} = \left\{\sum_{m',n'=-k}^{k} k_{m',n'} f(z_{m,n} - z_{m',n'})\right\}_{m,n\in\mathbb{Z}},$$

and thus results in the commutative diagram

$$
\begin{array}{ccc}
\mathcal{B}_w & \xrightarrow{\ \mathcal{K}_w\ } & \mathcal{B}_w \\
{\scriptstyle T_{\Psi_w}}\uparrow & & \downarrow{\scriptstyle T_{\Psi_w}^*} \\
\ell^2(\mathbb{Z}^2) & \longrightarrow & \ell^2(\mathbb{Z}^2)
\end{array}
$$

Equivalently, the discretized verion of $\mathcal{K}_w$ is defined via (A.6), which was to be shown. In order to define the activation layer $\Sigma_l$, we first assume that the activation function $\sigma\colon \mathbb{R}^2 \to \mathbb{R}^2$ is such that for every $f \in \mathcal{B}_w(\mathbb{R}^2)$

$$\sigma(f) \in \mathcal{B}_{\overline{w}}(\mathbb{R}^2), \tag{A.9}$$

for some $\overline{w} > w$. In fact, in Section 2 we assume that the pointwise activation can be approximated by an operator between bandlimited spaces and consequently (A.9) is satisfied up to negligible frequencies. Thus, the activation layer $\Sigma_{w,\overline{w}}\colon \mathcal{B}_w(\mathbb{R}^2) \to \mathcal{B}_w(\mathbb{R}^2)$ in (A.8) is defined by the composition

$$\Sigma_{w,\overline{w}} = P_{\mathcal{B}_w(\mathbb{R}^2)} \circ \sigma \circ P_{\mathcal{B}_{\overline{w}}(\mathbb{R}^2)}, \tag{A.10}$$

where $P_{\mathcal{B}_w(\mathbb{R}^2)}\colon \mathcal{B}_{\overline{w}}(\mathbb{R}^2) \to \mathcal{B}_w(\mathbb{R}^2)$ denotes the orthogonal projection onto $\mathcal{B}_w(\mathbb{R}^2)$ and $P_{\mathcal{B}_{\overline{w}}(\mathbb{R}^2)}\colon \mathcal{B}_w(\mathbb{R}^2) \to \mathcal{B}_{\overline{w}}(\mathbb{R}^2)$ denotes the natural embedding of $\mathcal{B}_w(\mathbb{R}^2)$ into $\mathcal{B}_{\overline{w}}(\mathbb{R}^2)$. The discretized version of each mapping in (A.10) is defined in order to guarantee a continuous-discrete equivalence (CDE) between the continuous and discrete levels. More precisely, $P_{\mathcal{B}_w(\mathbb{R}^2)}$ and $P_{\mathcal{B}_{\overline{w}}(\mathbb{R}^2)}$ are discretized via (A.6) as

$$\mathcal{D}_{\overline{w},w} = T_{\Psi_w}^* \circ P_{\mathcal{B}_w(\mathbb{R}^2)} \circ T_{\Psi_{\overline{w}}}, \qquad \mathcal{U}_{w,\overline{w}} = T_{\Psi_{\overline{w}}}^* \circ P_{\mathcal{B}_{\overline{w}}(\mathbb{R}^2)} \circ T_{\Psi_w},$$

which are respectively called downsampling and upsampling. Consequently, the discretized version of the activation layer is given by the composition

$$\mathcal{D}_{\overline{w},w} \circ \sigma \circ \mathcal{U}_{w,\overline{w}},$$

which yields the commutative diagram

$$
\begin{array}{ccccccc}
\mathcal{B}_w & \xleftarrow{P_{\mathcal{B}_{\overline{w}}(\mathbb{R}^2)}} & \mathcal{B}_{\overline{w}} & \xrightarrow{\ \sigma\ } & \mathcal{B}_{\overline{w}} & \xrightarrow{P_{\mathcal{B}_w(\mathbb{R}^2)}} & \mathcal{B}_w \\
{\scriptstyle T_{\Psi_w}}\uparrow & & \downarrow{\scriptstyle T_{\Psi_{\overline{w}}}^*} & & {\scriptstyle T_{\Psi_{\overline{w}}}}\uparrow & & \downarrow{\scriptstyle T_{\Psi_w}^*} \\
\ell^2(\mathbb{Z}^2) & \xrightarrow{\mathcal{U}_{w,\overline{w}}} & \ell^2(\mathbb{Z}^2) & \xrightarrow{\ \sigma\ } & \ell^2(\mathbb{Z}^2) & \xrightarrow{\mathcal{D}_{\overline{w},w}} & \ell^2(\mathbb{Z}^2)
\end{array}
$$

which we wanted to show. Finally, the activation layer might be followed by an additional projective operator, i.e., by a downsampling or an upsampling. Thus, this exact correspondence between its constituent continuous and discrete operators establishes CNO as an example of Representation equivalent neural operators or ReNOs in the sense of [2, Definiton 3.4] and [2, Remark 3.5], thus proving Proposition 2.1 of the main text. As in Appendix A.2, the above proofs can be readily adapted to bandlimited periodic functions, i.e. periodic functions with a finite number of non-zero Fourier coefficients.

# B  Proof of Theorem 3.1 of Main Text

We present the proof of a generalization of the universality result of Theorem 3.1. The theorem in the main text only holds when the differential operator $\mathcal{L}$ only depends on the coordinate $x$ through a coefficient function $a \in H^r(D)$. Although all benchmark PDEs in Section 4 satisfy this requirement, there are other important PDEs that do not, such as the standard elliptic PDE $\nabla \cdot (a\nabla u)) = f$. We therefore generalize this requirement in the following setting,

**Setting B.1.** *We set $D = \mathbb{T}^2$ and assume that the following is true,*

1. *$\mathcal{L}$ only depends on the coordinate $x$ through functions $a, f_1, \ldots, f_\ell \in H^r(\mathbb{T}^2)$.*

2. *The solution of the PDE characterized by $a$ and $f = (f_1, \ldots, f_\ell)$ is given by a continuous operator $\widetilde{\mathcal{G}} : \widetilde{\mathcal{X}} \subset (H^r(\mathbb{T}^2))^{\ell+1} \to H^r(\mathbb{T}^2) : (a, f) \mapsto u$ or $u(T)$, depending on the PDE. The operator of interest $\mathcal{G}^\dagger$ is a restriction of $\widetilde{\mathcal{G}}$ for fixed $f_1, \ldots, f_\ell$ i.e., $\mathcal{G}^\dagger : \mathcal{X}^* \subset H^r(\mathbb{T}^d) \to H^r(\mathbb{T}^d) : a \mapsto \widetilde{\mathcal{G}}(a, f_1, \ldots, f_\ell)$.*

3. *Similar to (2.1), it holds for all $(a, f), (a', f') \in \mathcal{X}^*$ it holds that*

$$\left\| \widetilde{\mathcal{G}}(a, f) - \widetilde{\mathcal{G}}(a', f') \right\|_{L^p(\mathbb{T}^2)} \leq \omega \left( \|a - a'\|_{H^\sigma(\mathbb{T}^2)} + \max_i \|f_i - f_i'\|_{H^\sigma(\mathbb{T}^2)} \right), \quad \text{(B.1)}$$

*for some $p \in \{2, \infty\}$ and $\sigma \in \mathbb{N}_0$ with $\sigma < r$. This is automatically satisfied if $\mathcal{X}^*$ is compact and $\widetilde{\mathcal{G}}$ is continuous [24].*

4. *It holds that the activation function $\sigma$ is at least $r$ times continuously differentiable and not a polynomial. (See Remark B.4 for a generalization.)*

In addition, we will use the following notation in the proof.

- For $J \in \mathbb{N}$ we define for every $j \in \{0, \ldots, J-1\}^2$ the grid $\mathbf{x}_j^J = (2\pi j_1/J, 2\pi j_2/J)$.
- We denote the Fourier basis by $\{\mathbf{e_k}\}_{\mathbf{k} \in \mathbb{Z}^2}$, following the notation of [24]. For $\mathbf{k} = (\mathbf{k}_1, \ldots, \mathbf{k}_d) \in \mathbb{Z}^d$, we let $\sigma(\mathbf{k})$ be the sign of the first non-zero component of $\mathbf{k}$ and we define

$$\mathbf{e_k} := C_{\mathbf{k}} \begin{cases} 1, & \sigma(\mathbf{k}) = 0, \\ \cos(\mathbf{k} \cdot \mathbf{x}), & \sigma(\mathbf{k}) = 1, \\ \sin(\mathbf{k} \cdot \mathbf{x}), & \sigma(\mathbf{k}) = -1, \end{cases} \quad \text{(B.2)}$$

where the factor $C_{\mathbf{k}} > 0$ ensures that $\mathbf{e_k}$ is properly normalized, i.e. that $\|\mathbf{e_k}\|_{L^2(\mathbb{T}^d)} = 1$.

- For $N \in \mathbb{N}$ let $P_N$ denote a trigonometric polynomial interpolation operator as in (B.16) in **SM** B.1.

Assuming Setting B.1 we can now prove the following theorem on the universality of CNOs. In the proof we will construct an operator $\mathcal{G} : H^r(\mathbb{T}^2) \to C(\mathbb{T}^2)$, mapping between function spaces, and we will therefore allow to apply the activation function to the continuous representation of the signal rather than an upsampled version. We then make the link to the discrete implementation of the CNO by considering an encoder $\mathcal{E}_K$ that maps the input function $a$ to the evaluation of $a$ on a grid, enhanced by some Fourier features [53] in case $\ell > 0$ in Setting B.1.

**Theorem B.2.** *Let $\sigma \in \mathbb{N}_0$ and $p \in \{2, \infty\}$ as in (B.1), $r > \max\{\sigma, 2/p\}$ and $B > 0$. For any $\varepsilon > 0$ and any operator $\mathcal{G}^\dagger$ satisfying Setting B.1, there exist $K, N \in \mathbb{N}_0$ and a CNO $\mathcal{G} : H^r(\mathbb{T}^2) \to C(\mathbb{T}^2)$ such that for every $a \in \mathcal{X}^*$ with $\|a\|_{H^r(\mathbb{T}^2)} \leq B$ it holds,*

$$\left\| \mathcal{G}^\dagger(a) - \mathcal{G}(a) \right\|_{L^p(\mathbb{T}^2)} < \varepsilon. \quad \text{(B.3)}$$

*The CNO is implemented through an encoder*

$$\mathcal{E}_K : H^r(\mathbb{T}^2) \to (\mathbb{R}^{N \times N})^{(K+1)^2} : a \mapsto (a(\mathbf{x}^N), (\cos(\mathbf{k} \cdot \mathbf{x}^N), \sin(\mathbf{k} \cdot \mathbf{x}^N))_{1 \leq \|\mathbf{k}\|_\infty \leq K}) \quad \text{(B.4)}$$

*and a single invariant block $\widehat{\Phi} : (\mathbb{R}^{N \times N})^{(K+1)^2} \to \mathbb{R}^{N \times N}$ such that $\mathcal{G}(a)(\mathbf{x}^N) = (\widehat{\Phi} \circ \mathcal{E}_K)(a)$. If $\ell = 0$ (see Setting B.1) then $K = 0$, meaning that no Fourier features are needed.*

*Proof.* Let $M, N \in \mathbb{N}$ with $N/M \in \mathbb{N}$. We will construct a CNO with input $a(\mathbf{x}^N)$ and the Fourier features $\mathbf{e}_k(\mathbf{x}^N)$ for $\mathbf{k} \in \mathcal{K} := \{-M/2, -M/2 + 1, \ldots, M/2\}^2 \setminus \{0, 0\}$, summarized in the tensor $(a(\mathbf{x}^N), \mathbf{e}^{M/2}(\mathbf{x}^N)) := (a(\mathbf{x}^N), (\mathbf{e_k}(\mathbf{x}^N))_{\mathbf{k} \in \mathcal{K}})$. In the proof, we will use the property that bandlimited functions can be represented by their function values on a fine enough grid. We will therefore first construct a continuous operator $\mathcal{G} : H^r(\mathbb{T}^2) \to C(\mathbb{T}^2)$ that is a good approximation of $\mathcal{G}^\dagger$. In the second step, we will then prove that $\mathcal{G}(a)(\mathbf{x}^N)$ indeed corresponds to a CNO.

**Step 1: construction of $\mathcal{G}$.** First, since $a, f \in H^r(\mathbb{T}^2)$ we can use Lemma A.1 and assumption (B.1) on the stability of $\widetilde{\mathcal{G}}$ to find that,

$$\left\| \widetilde{\mathcal{G}}(a, f) - \widetilde{\mathcal{G}}(P_M(a, f)) \right\|_{L^p(\mathbb{T}^2)} \leq \omega \left( C_{B,f} M^{-(r-\sigma)} \right). \quad \text{(B.5)}$$

Next, we define for any $J \in \mathbb{N}$ the set

$$\mathcal{A}_J = \{ \mathbf{y} \in (\mathbb{R}^{J \times J})^{(M+1)^2} \mid \exists a \in H^r(\mathbb{T}^2) : \mathbf{y} = (a(\mathbf{x}^J), \mathbf{e}^{M/2}(\mathbf{x}^J)) \text{ and } \|a\|_{H^r(\mathbb{T}^2)} \leq B \}, \quad \text{(B.6)}$$

and the map,

$$G : \mathcal{A}_M \subset (\mathbb{R}^{M \times M})^{(M+1)^2} \to \mathbb{R} : (a(\mathbf{x}^M), \mathbf{e}^{M/2}(\mathbf{x}^M)) \mapsto \widetilde{\mathcal{G}}(P_M(a, f))(\mathbf{x}_{0,0}). \tag{B.7}$$

The existence of the map $G$ can be justified as follows. Let $P_M$ denote a trigonometric polynomial interpolation operator as in (B.16) in **SM** B.1. By the Nyquist–Shannon sampling theorem and the Whittaker–Shannon interpolation formula there is a bijection between the discrete values $(a(\mathbf{x}^M), \mathbf{e}^{M/2}(\mathbf{x}^M))$ and $P_M a$ and $\mathbf{e}^{M/2}$, and therefore also $P_M a$ and $P_M f_i$ for all $1 \le i \le \ell$. Hence, the mapping $G$ is equivalent to the mapping $P_M(a, f) \mapsto P_M u$, and therefore well-defined. The continuity of $G$ follows from that of $\widetilde{\mathcal{G}}$. By the universal approximation theorem (Theorem B.6) there exists a shallow neural network $\Psi$ such that $|\Psi(\mathbf{y}) - G(\mathbf{y})| < \varepsilon$ for all $\mathbf{y} \in \mathcal{A}_M$. Note that $\Psi$ only provides an approximation in the point $\mathbf{x}_{0,0}$. We can expand $\Phi$ to the whole $\mathbb{T}^2$ by defining the operator $\Psi^*$ as follows,

$$\Psi^* : \mathcal{X}^* \to C(\mathbb{T}^2) : a \mapsto \left[ \mathbb{T}^2 \ni \mathbf{z} \mapsto \Psi \left( a(\mathbf{z} + \mathbf{x}^M), \mathbf{e}^{M/2}(\mathbf{z} + \mathbf{x}^M) \right) \right]. \tag{B.8}$$

For the intuition of the reader: the extension from $\Psi$ to $\Psi^*$ is similar to the extension from the local stencil of a finite difference scheme to its corresponding global approximation. As a result, $\Psi^*$ has the same accuracy as $\Psi$,

$$\left\| \mathcal{G}^\dagger(P_M a) - \Psi^*(P_M a) \right\|_{C^0(\mathbb{T}^2)} < \varepsilon. \tag{B.9}$$

We finalize our construction by projecting $\Psi^*(P_M a)$ on to the space of trigonometric polynomials. The accuracy of such a projection is given by Lemma A.1,

$$\|(P_N - \mathrm{Id})\Psi^*(a)\|_{L^p(\mathbb{T}^2)} \le \|(P_N - \mathrm{Id})\Psi^*(a)\|_{H^{1-2/p}(\mathbb{T}^2)} \le C N^{-(r-2/p)} \|\Psi^*(a)\|_{H^r(\mathbb{T}^2)}, \tag{B.10}$$

where we used that either $p = 2$ or $p = \infty$. It is important to note that $\|\Psi^*(a)\|_{H^r(\mathbb{T}^2)}$ is independent of $N$. We then define the operator $\mathcal{G}$ as,

$$\mathcal{G}(a)(\mathbf{z}) = (P_N \circ \Psi^*)(P_M a)(\mathbf{z}). \tag{B.11}$$

Finally, we can put all obtained estimates together to find,

$$
\begin{aligned}
&\left\| \mathcal{G}^\dagger(a) - \mathcal{G}(a) \right\|_{L^p(\mathbb{T}^2)} \\
&\le \left\| \mathcal{G}^\dagger(a) - \mathcal{G}^\dagger(P_M a) \right\|_{L^p(\mathbb{T}^2)} + \left\| \mathcal{G}^\dagger(P_M a) - \Psi^*(P_M a) \right\|_{C^0(\mathbb{T}^2)} \\
&\quad + \left\| \Psi^*(P_M a) - \mathcal{G}(a) \right\|_{L^p(\mathbb{T}^2)} \\
&\le \omega \left( C_{B,f} M^{-(r-\sigma)} \right) + \varepsilon + C_{B,M,\varepsilon} N^{-(r-2/p)}.
\end{aligned}
\tag{B.12}
$$

It then follows that one can make this upper bound arbitrarily small by choosing $\varepsilon$ sufficiently small and $M, N$ sufficiently large (in that order).

**Step 2: $\mathcal{G}$ corresponds to a CNO.** We will now show that the operator $\mathcal{G}$ is in agreement with our definition of a convolutional neural operator (CNO). To do so, we will use that trigonometric polynomials up to a certain degree can be exactly retrieved based on their values on a grid (see **SM** B.1 and [19]).

First of all, given that $N/M \in \mathbb{N}$ we find that the functions $P_M a$ and $\mathbf{e}^{M/2}$ can be exactly recovered from their discrete values on the grid $\mathbf{x}^N$. We therefore will look for a CNO with input $\mathbf{y} = (a(\mathbf{x}^N), \mathbf{e}^{M/2}(\mathbf{x}^N)) \in \mathcal{A}_N$ for which the continuous representation of the output agrees with $\mathcal{G}(a)$.

A crucial next observation is that $G$ is equivariant with respect to translations in space of the input (simultaneously across all channels). By [46, Theorem 1.1] there then exists a shallow CNN $\Phi$ such that $\pi \circ \Phi = \Psi$, where $\pi$ is the projection on the first coordinate $\pi : (\mathbb{R}^{N \times N})^{(M+1)^2} \to \mathbb{R}^{(M+1)^2} : X \mapsto (X_{1,1}^1, \ldots, X_{1,1}^\ell)$ (as in [46]). For simplicity we will assume that the CNN is of the form $\Phi(\mathbf{y}) = K_2 * \sigma(K_1 * \mathbf{y})$, i.e. that it only has one channel at every layer. The proof of the general case is completely analogous, but much heavier on notation.

We then lift the convolution filter $K_1 \in \mathbb{R}^{M \times M}$ to the grid $\mathbb{R}^{N \times N}$ by using a stride of $N/M$ and filling up the rest by zeroes. More rigorously, we consider the matrix $\widehat{K}_1 := K_1 \otimes E$ with

$E_{ij} = \delta_{i1}\delta_{j1}$. Similarly we define $\widehat{K}_2 := K_2 \otimes E$. We can then define a new CNN $\widehat{\Phi} : \mathcal{A}_N \to \mathbb{R}^{N \times N} : \mathbf{y} \mapsto \widehat{K}_2 \star \sigma(\widehat{K}_1 \star \mathbf{y})$. The output of $\widehat{\Phi}$ then consists of approximations of $\mathcal{G}^\dagger(a)$ at $(N/M)^2$ different $M \times M$ subgrids of $\mathbf{x}^N$, i.e. all possible translations of $\mathbf{x}^M$ within $\mathbf{x}^N$. More precisely, it holds that

$$\Psi^*(\mathbf{x}^N) = \widehat{\Phi}\left(P_M a(\mathbf{x}^N), \mathbf{e}^{M/2}(\mathbf{x}^N)\right) \in \mathbb{R}^{N \times N}. \tag{B.13}$$

Moreover, since the operator $P_N$ only uses the values of $\Psi^*$ on the grid $\mathbf{x}^N$ it follows that applying $P_N$ to the right-hand side of the above equation or applying an interpolation sinc filter with corresponding frequency gives the exact same result,

$$\mathcal{G}(a)(\mathbf{x}^N) = (P_N \circ \Psi^*)(P_M a)(\mathbf{x}^N) = h_N \star \widehat{\Phi}\left(P_M a(\mathbf{x}^N), \mathbf{e}^{M/2}(\mathbf{x}^N)\right). \tag{B.14}$$

The right-hand side exactly corresponds to our definition of a CNO, thereby concluding the universality proof. □

**Remark B.3** (Alternative proof). *We stress that it is crucial in the proof that $M$ can be chosen independently of $N$. A straightforward application of [46, Theorem 1.1] on the map $G$ with $N = M$ would not lead to an accurate CNO approximation as the $\|\Psi^*(a)\|_{H^r(\mathbb{T}^2)}$ will depend on $N = M$ such that $N^{-(r-2/p)} \|\Psi^*(a)\|_{H^r(\mathbb{T}^2)}$ might not convergence to zero. In addition, because of the used trick we obtain convolution kernels with a stride of $N/M$ and therefore a sparse kernel. An alternative strategy could be to replace the universal approximation (Theorem B.6) by an approximation theorem that provides explicit control on the network size and upper bounds on the weights such as those in [10]. Other than a much more complicated proof, one will also need to put stronger regularity conditions on $\mathcal{G}^\dagger$.*

**Remark B.4** (Polynomial and rational activations). *The CNO constructed in the above theorem is exactly equivariant. As suggested in [22], it can be sufficient in practice to break this perfect equivariance by applying the activation function $\sigma$ to an upsampled discrete version of the signal rather than the continuous representation of the signal. We do the same in our implementation of CNO. Note that if one would use polynomial activation functions one could still recover exact equivariance by choosing a high enough upsampling rate. In this case the universality of CNOs can be proven by replacing the universal approximation theorem for neural networks (Theorem B.6) by the Weierstrass approximation theorem. The rest of the proof of Theorem B.2 remains unchanged. Similarly, one could consider using Padé and rational approximants as activation functions [54, 43, 12, 6]. The computation of a rational activation function $\sigma(x) = p(x)/q(x)$ can then be approximated by iteratively minimizing $\|p(x)\sigma(x) - q(x)\|_2^2$, following the idea of [60]. Methods such as Newton-Raphson only involve multiplications and can therefore be completely applied in an alias-free way through proper upsampling before, and downsampling after each multiplication.*

**Remark B.5** (Physics-informed CNOs). *Physics-informed learning employs a PDE residual-based loss to circumvent the need for training data. Examples of such frameworks include physics-informed neural networks (PINNs) [49], physics-informed DeepONets [59] and physics-informed FNOs [36]. Using the continuous representation of the CNO output, one can use automatic differentation to created a physics-informed CNO loss. In order to use the tools provided in [11, Theorem 3.9] to obtain a bound on the approximation error for physics-informed CNOs, one needs to prove that the CNO error converges at a certain rate in terms of its size. Although Theorem B.2 does not provide such a rate, its proof does give a hint of which stronger assumptions are needed to obtain this result. The most notable ingredients include a stronger stability result (B.1) with a continuity modulus $\omega$ decreasing at least at a polynomial rate, and a stronger regularity assumption on $G$ (B.7), and hence $\mathcal{G}^\dagger$.*

## B.1 Auxiliary results

We list the auxiliary results that are used in the proof of Theorem B.2. First, we state a well-known version of the universal approximation theorem for feedforward neural networks [31]:

**Theorem B.6** ([31]). *Let $\sigma : \mathbb{R} \to \mathbb{R}$ be a function that is locally essentially bounded with the property that the closure of the set of points of discontinuity has zero Lebesgue measure. For $1 \le j \le n$, let $\alpha_j, \theta_j \in \mathbb{R}$ and $y_j \in \mathbb{R}^d$. Then finite sums of the form*

$$G(x) = \sum_{j=1}^{N} \alpha_j \sigma(y_j^T x + \theta_j), \quad x \in \mathbb{R}^d \tag{B.15}$$

*are dense in $C(\mathbb{R}^d)$ if and only if $\sigma$ is not an algebraic polynomial.*

Next, we demonstrate the equivalence of using the interpolation sinc filter (2.5) and trigonometric polynomial interpolation. If you sample a function $f \in C(\mathbb{T} = [0, 2\pi))$ with sampling frequency $\frac{2\pi}{N}$, the result obtained through trigonometric polynomial interpolation $P_N g$ is given by [19],

$$P_N f(x) = \begin{cases} \sum_{|n| \leq (N-1)/2} \frac{1}{N} \sum_{j=0}^{N-1} f(x_j) \exp(in(x - x_j)), & \text{for } n \text{ odd,} \\ \sum_{|n| \leq N/2} \frac{1}{Nc_n} \sum_{j=0}^{N-1} f(x_j) \exp(in(x - x_j)), & \text{for } n \text{ even,} \end{cases} \quad \text{(B.16)}$$

where $x_j = 2\pi j/N$, $c_n = 1$ for $|n| < N/2$ and $c_n = 2$ for $|n| = N/2$. We will prove that one obtains the exact same result by using an interpolation sinc filter with the same frequency on the periodic extension of $f$. We prove this result in the one-dimensional case for odd $N$. The result for even $N$ follows in an identical way, the result for the multi-dimensional case through tensorisation.

**Lemma B.7.** *For any $N \in 2\mathbb{N} + 1$ and $f \in C(\mathbb{T})$ it holds that,*

$$P_N f(x) = \frac{1}{N} \sum_{|n| \leq (N-1)/2} \sum_{j=0}^{N-1} f(x_j) \exp(in(x - x_j)) = \sum_{n \in \mathbb{Z}} f(x_n) \operatorname{sinc}\left(N \cdot \frac{x - x_n}{2\pi}\right) \quad \text{(B.17)}$$

*Proof.* As a first step, it follows from [19, Section 2.2.2] that

$$P_N f(x) = \frac{1}{N} \sum_{|n| \leq (N-1)/2} \sum_{j=0}^{N-1} f(x_j) \exp(in(x - x_j)) = \frac{1}{N} \sum_{n=0}^{N-1} f(x_n) \frac{\sin(N(x - x_n)/2)}{\sin((x - x_n)/2)}.$$

$$\text{(B.18)}$$

Then we use the result from [51], where we replace their $N$-periodic signal $x(t)$ by the function $f(x)$ through the transformation $t = Nx/2\pi$ and $x(t) = f(2\pi t/N)$. In their notation, but with the change that here we use the *normalized* sinc function ($\operatorname{sinc}(x) = \sin(\pi x)/\pi x$ for $x \neq 0$), [51] shows that

$$\sum_{n \in \mathbb{Z}} x(n) \operatorname{sinc}(t - n) = \frac{\sin(\pi t)}{N} \sum_{n=-L}^{M-1} x(n)(-1)^n \csc(\pi(t - n)/N) \quad \text{(B.19)}$$

with $L, M \in \mathbb{N}_0$ such that $L + M = N$. We will take $L = 0$ and $M = N$, and use that $\csc(z) = 1/\sin(z)$ and that $\cos(\pi n) = (-1)^n$ and $\sin(\pi n) = 0$ to obtain,

$$\sum_{n \in \mathbb{Z}} x(n) \operatorname{sinc}(t - n) = \frac{1}{N} \sum_{n=-L}^{M-1} x(n) \frac{\sin(\pi(t - n))}{\sin(\pi(t - n)/N)}, \quad \text{(B.20)}$$

which is equivalent to,

$$\sum_{n \in \mathbb{Z}} f(x_n) \operatorname{sinc}(N(x - x_n)/2\pi) = \frac{1}{N} \sum_{n=0}^{N-1} f(x_n) \frac{\sin(N(x - x_n)/2)}{\sin((x - x_n)/2)}. \quad \text{(B.21)}$$

Combining all obtained equalities proves the claim. $\qquad \square$

## C  Technical Details for Section 4 of Main Text

### C.1  Training and Implementation Details

We start with a succinct description of the baselines that were used in the main text.

#### C.1.1  Feed Forward Dense Neural Networks

Given an input $u \in \mathbb{R}^m$, a feedforward neural network (also termed as a multi-layer perceptron), transforms it to an output, through a layer of units (neurons) which compose of either affine-linear

maps between units (in successive layers) or scalar nonlinear activation functions within units [15], resulting in the representation,

$$\overline{u}_\theta(y) = C_L \circ \sigma \circ C_{L-1} \ldots \circ \sigma \circ C_2 \circ \sigma \circ C_1(u). \tag{C.1}$$

Here, $\circ$ refers to the composition of functions and $\sigma$ is a scalar (nonlinear) activation function. For any $1 \leq \ell \leq L$, we define

$$C_\ell z_\ell = W_\ell z_\ell + b_\ell, \text{ for } W_\ell \in \mathbb{R}^{d_{\ell+1} \times d_\ell}, z_\ell \in \mathbb{R}^{d_\ell}, b_\ell \in \mathbb{R}^{d_{\ell+1}}., \tag{C.2}$$

and denote,

$$\theta = \{W_\ell, b_\ell\}_{\ell=1}^L, \tag{C.3}$$

to be the concatenated set of (tunable) weights for the network. Thus in the terminology of machine learning, a feed forward neural network (C.1) consists of an input layer, an output layer, and $L$ hidden layers with $d_\ell$ neurons, $1 < \ell < L$. In all numerical experiments, we consider a uniform number of neurons across all the layer of the network $d_\ell = d_{\ell-1} = d$, $1 < \ell < L$. The first baseline model consists into a feed forward neural network with *residual blocks* which use *skip or shortcut connections* [18]. A residual block spanning $k$ layers is defined as follows,

$$r(z_\ell, z_{\ell-k}) = \sigma(W_\ell z_\ell + b_\ell) + z_{\ell-k}. \tag{C.4}$$

The residual network takes as input a sample function $u \in \mathcal{X}$, encoded at $m = s \times s$ *Cartesian grid* points $(x_1, \ldots, x_m)$, $\mathcal{E}(u) = (u(x_1), \ldots, u(x_m)) \in \mathbb{R}^m$, and outputs the output sample $\mathcal{G}(u) \in \mathcal{Y}$ encoded at the same set of points, $\mathcal{E}(\mathcal{G}(u)) = (\mathcal{G}(u)(x_1), \ldots, \mathcal{G}(u)(x_m)) \in \mathbb{R}^m$. In all the experiments, but the compressible Euler, $s = 64$. Instead, for the compressible Euler equation, the sampling rate is $s = 128$. The number of layers $L$, neurons $d$ are chosen though cross-validation, whereas the activation function $\sigma$ corresponds to a Leaky ReLU and the depth of the residual block $k$ is fixed and equal to 2.

### C.1.2 ResNet

For the ResNet baseline, we adopt a convolutional neural network architecture with additional skip connections, as described in [18] and . The architecture begins with an initial block composed of a convolutional layer with a $7 \times 7$ kernel, zero padding, and a ReLU activation function, all followed by batch normalization. The first layer generates an output with a channel count of $c$.

Subsequently, the output from the initial block undergoes downsampling via a second block. This second block consists of two sub-blocks, each mirroring the structure of the initial block but with a smaller $3 \times 3$ convolutional layer, a stride of 2, and padding of 1. The channel count doubles within each of these sub-blocks.

The downsampled output is then processed through a series of $N_{res}$ residual blocks, as defined in equation C.4. Each residual block consists of a convolution operation, batch normalization, and ReLU activation.

Finally, the signal is upsampled through a pair of blocks comprising transposed convolution, batch normalization, and ReLU activation.

The complete architecture is available at repository of the paper [20] f:`https://github.com/junyanz/pytorch-CycleGAN-and-pix2pix/blob/master/models/networks.py`.

### C.1.3 UNet

For the UNet baseline we use the model architecture proposed in [50]. However, we slightly modify the proposed architecture by varying the number of output channels $c$ of the first convolutional layer, which is chosen through cross validation. We ensure that the number of channels used in the subsequent layers align with the chosen value of $c$. Specifically, we respect the progressive increase or decrease in the number of channels as established in the original architecture across different layers.

### C.1.4 Convolutional Neural Operator

**Design of the filters.** As we noted before, perfect *sinc* interpolation filters $h_w$ have infinite impulse response and cause ringing artifacts around high-gradient points due the Gibbs phenomenon. In

practice, one uses *windowed-sinc* filters which serve as convenient approximations of $h_w$. They have finite impulse response and weakened ringing effect [57].

The *windowed-sinc* filters are constructed by multiplying the ideal filter $h_w$ by a corresponding *window function* of *finite* length. That is equivalent to convolving the filter with the window function in the frequency domain. To design the windowed filter, one can use standard Python libraries and their functions such as *scipy.signal.firwin*. By using this function, we are enabled to manually control the cutoff frequency $w_c$ and the half-width of the transition band $w_h$ of the designed filters. We design discrete filters with a prescribed compact support $N_{tap} \in \mathbb{N}$. In this case, we say that a designed filter has $N_{tap}$ *taps*. Implementation of the filters is borrowed from [22] (CUDA programming model).

We show several $1D$ designed filters in the Figure 4, where we set $w_c = s/(2 + \varepsilon)$, for $\varepsilon \ll 1$. We control the half-width of the filter $w_h = c_h \cdot s$ by controling the coefficient $c_h$. When $c_h$ is set to $0.5$, one would anticipate the design of an almost perfect *sinc* filter. However, the presence of undesirable oscillations in the frequency domain can be observed due to the finite impulse response of windowed filters, as depicted in Figure 4. That is why we set $c_h$ to be at least $0.6$. One can implement a 2D filter by first convolving a 1D filter with each row and then with each column.

The activation layer $\Sigma_{w,\overline{w}}$ plays a vital role in the CNO model. It is essential to closely examine the ratio $N_\sigma = \overline{w}/w$ as a significant parameter. To facilitate implementation of the CNO, we make the assumption that $N_\sigma \in \mathbb{N}$ and $N_\sigma \geq 2$. Throughout the entire architecture, we make the assumption that the value of $N_\sigma$ remains fixed. In our implementation of the CNO, it is worth noting that the value of $N_\sigma$ can also be a rational number if the sampling rate of an input signal requires it (e.g. if one wants to upsample a signal from the sampling rate 11 to the sampling rate 20).

We choose to fix the coefficient $w_c = s/2.0001$, so that the cutoff frequency is very close to the *Nyquist critical frequency*. It remains to choose the number of taps $N_{tap}$, the coefficient related to the half-width of the filter $c_h$ and the ratio related to the activation layer $N_\sigma$.

**Choice of parameters.** Throughout our experiments, we maintained a consistent configuration, setting $N_\sigma$ to 2, $c_h$ to 0.8, and $N_{tap}$ to 12. Prior to finalizing the filter parameters, we conducted experiments using various filters; however, no significant differences were observed. To further validate this assumption, we conducted the Navier-Stokes experiment using different filter designs. First, we selected the best-performing CNO model based on the criteria described in C.2 using filter parameters $N_\sigma = 2$, $c_h = 0.8$ and $N_{tap} = 12$. For this chosen model, we conducted training with identical model settings as outlined in 12, but with different values for coefficients $c_h$, $N_\sigma$ and $N_{tap}$.

In the first set of experiments, we set $N_{tap} = 12$ and vary $c_h$ and $N_\sigma$. Note that increasing the coefficient $N_\sigma$ leads to a significant increase in computational time. We show different test errors in the Table 2. Once the coefficient $c_h$ reaches a sufficiently high value (i.e. $c_h \geq 0.8$), we observe no significant difference in test errors. Additionally, we do not find a high correlation between the error and the coefficient $N_\sigma$. We set the $N_\sigma$ as low as possible, to a fixed value of $N_\sigma = 2$. Similarly, we set the coefficient $c_h$ to a fixed value of $0.8$.

In the second experiment, we fix $N_\sigma = 2$ and $c_h = 0.8$ and vary the number of taps $N_{tap}$. By increasing the number of taps, the computational time also increases. We show different test errors in the Table 3. Although there is an improvement of approximately $1.5\%$ in the test error when $N_{tap} = 20$ compared to when $N_{tap} = 12$, it comes at the cost of increased training time. Specifically, the training time per one epoch increases from $4.37s$ to $5.26s$, representing more than $20\%$ increase. Due to this significant increase in training time, but not very significant improvement in performance, we decide to fix the number of taps at $N_{tap} = 12$.

**Remark C.1.** *Given the above description, it is important to emphasize that, although in principle, the activation layer of CNO (2.3) needs to be exactly equivariant, i.e., $\sigma(\mathcal{B}_\omega) \subset \mathcal{B}_{\omega'}$ for the pair $(\omega, \omega')$, for the CNO architecture to be representation equivariant in the sense of [2], definition 3.4, see also section A.5, several approximations are used in practice that might be lead to this condition to hold only approximately. However, as the above results show, once the upsampling frequency is choosen high enough, this approximation of equivariance seems to suffice in practice, see also results in Section C.4. Neverthelesss, if exact equivariance is sought for, it can be realized through either polynominal or rational activation functions as suggested in Remark B.4 although this choice might be of little practical utility.*

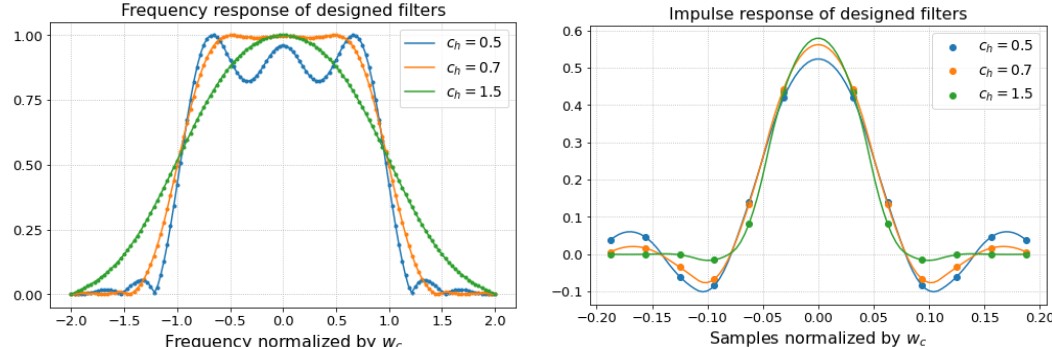

Figure 4: On the left: Frequency responses of different designed filters. On the right: Impulse responses of different designed filters. The sampling rate is $s = 128$, the cutoff frequency is $w_c = s/2.0001$, while the half-width of each filter is $w_h = c_h \cdot s$. Each filter has $N_{tap} = 12$ taps.

Table 2: CNO model. Navier-Stokes Equations. Relative median $L^1$-error computed over 128 in-distribution testing samples for different filter designs. The error of the model with original filter parameters $c_h = 0.8$, $N_\sigma = 2$ and $N_{tap} = 12$ is marked in blue.

|  | $\mathbf{c_h = 0.6}$ | $\mathbf{c_h = 0.8}$ | $\mathbf{c_h = 1.0}$ | $\mathbf{c_h = 1.5}$ | $\mathbf{c_h = 2.0}$ |
|---|---|---|---|---|---|
| $\mathbf{N_\sigma = 2}$ | 2.87% | **2.76%** | 2.77% | 2.91% | 2.86% |
| $\mathbf{N_\sigma = 3}$ | 2.93% | 2.86% | 2.86% | 2.87% | 2.97% |
| $\mathbf{N_\sigma = 4}$ | 2.80% | 2.89% | 2.88% | 2.87% | 2.89% |
| $\mathbf{N_\sigma = 5}$ | 2.93% | 2.84% | 2.88% | 2.98% | 2.89% |
| $\mathbf{N_\sigma = 6}$ | 3.02% | 2.86% | 2.88% | 2.99% | 2.82% |

In the simplest scenario, the architecture consists only of the lifting layer, number of (D) and (U) blocks, and the projection layer. In this simple scenario, once the input is lifted to higher dimensional space (in the channel width), one performs first $T$ iterations of (D) blocks. These $T$ iterations define the *encoder*, namely

$$v_{l+1} = \mathcal{D}_{s_l, s_{l+1}} \circ \Sigma_{s_l, s_{l+1}} \circ \mathcal{K}_{s_l}(v_l), \quad v_l \in \mathcal{B}_{s_l}(D, \mathbb{R}^{d_l}), \quad l = 0 \dots T - 1,$$

where $s_l = s/2^l$ is the current bandlimit and $d_l$ is the current number of channels. The next $T$ iterations are (U) blocks and are devoted to the *decoder*. Let $\widetilde{s}_l = s_{2T-l}$. The decoder is defined as

$$v_{l+1} = \mathcal{U}_{\widetilde{s}_l, \widetilde{s}_{l+1}} \circ \Sigma_{\widetilde{s}_l, \widetilde{s}_{l+1}} \circ \mathcal{K}_{\widetilde{s}_l}(v_l), \quad v_l \in \mathcal{B}_{\widetilde{s}_l}(D, \mathbb{R}^{d_l}), \quad l = T \dots 2T - 1.$$

The last output of the decoder is projected to the output space (in the channel width). In all the experiments, we use $d_l = d_e/2$ as the lifting dimension. In the encoder, the number of channels increases as per

$$d_e/2 \mapsto d_e \mapsto 2d_e \mapsto \cdots \mapsto 2^{T-1} d_e.$$

The number $d_e$ is a hyperparameter. In this simple case where *no* UNet style patching is present in the architecture, the number of channels in the decoder decreases as per

$$2^{T-1} d_e \mapsto 2^{T-2} d_e \mapsto \cdots \mapsto d_e$$

When the patching is present in the architecture (see Figure 1), number of channels in the decoder changes differently (as a certain number of transfered channels is concatenated).

**Operator UNet architecture.** We add 2 (I) block 2.8 before each upsampling block. One block is applied before patching the additional channels, while the other is applied after patching. Additionally, we add a few (R) blocks 2.7 between each level of the encoder and decoder. We denote the number of residual blocks in the bottleneck of the network as a hyperparameter $N_{res,b}$, while the number

Table 3: CNO model. Navier-Stokes Equations. Relative median $L^1$-error computed over 128 in-distribution testing samples for different number of taps $N_{tap}$. The error of the model with original filter parameters $c_h = 0.8$, $N_\sigma = 2$ and $N_{tap} = 12$ is marked in blue.

| | $\mathbf{N_{tap} = 12}$ | $\mathbf{N_{tap} = 16}$ | $\mathbf{N_{tap} = 20}$ | $\mathbf{N_{tap} = 24}$ |
|---|---|---|---|---|
| $\mathbf{N_\sigma = 2}$ & $\mathbf{c_h = 0.8}$ | **2.76%** | 2.72% | 2.70% | 2.75% |

of (R) blocks in the intermediate levels is denoted by $N_{res,i}$ (each level has the same number of (R) blocks). Throughout our training and testing, we fix the size of the convolution kernels to $k = 3$. Moreover, we apply *batch normalization* after each convolution operation, except in the lifting and the projection layers.

**Remark C.2.** *The objectives of cross-validation are $T$, $d_e$, $N_{res,b}$ and $N_{res,i}$.*

### C.1.5 Galerkin Transformer

The Galerkin Transformer (abbreviated as GT) as presented in [8] is a model founded on attention-based operator learning. Central to its design is a "softmax-free" attention mechanism. Structurally, GT is an encoder-decoder model, and it uses the Galerkin-type transformer at its architectural bottleneck.

The encoder's role is to convert the input into the latent feature domain. Its design comprises 4 convolutional layers, which incrementally downscale the input's dimensions while enlarging its channel dimensions. Further enhancing its function, the encoder incorporates positional encoding at a coarser level, which is then combined with the extracted features and forwarded to the bottleneck.

The features are flattened, paving the way for the application of scaled dot-product multi-head attention. Let us characterize the single-head Galerkin-type attention: Given an input embedding $y \in \mathbb{R}^{n \times d}$, and using trainable matrices $W_Q, Q_K, Q_V \in \mathbb{R}^{d \times d}$, we can determine the query, key, and value as $Q = yW_Q$, $K = yW_K$, and $V = yW_V$, respectively. The formal representation of the Galerkin-type single-head attention, denoted as Attn $: \mathbb{R}^{n \times d} \to \mathbb{R}^{n \times d}$ is

$$\text{Attn}(y) = y + Q(\widetilde{K}^T \widetilde{V})/n + g(y + Q(\widetilde{K}^T \widetilde{V})/n),$$

where $g$ is a 2-layer FFNN and $\widetilde{\cdot}$ is the layer normalization.

Lastly, the decoder is made up of a convolutional neural network that upsamples the output of the transformer to a desired dimension and several spectral convolutional layers. For an in-depth understanding of spectral convolutional layers, one can refer to [33].

Convolutional neural network in the encoder uses *relu* activation function, while the one in the decoder uses *silu* activation function. Spectral layers in the decoder use *silu* activation function.

The objectives of the cross-validation are:

- number of attention blocks : $n$
- number of heads in the attention: $h$
- latent dimension in the attention: $d$
- number of decoder layers: $L$
- latent dimension of the decoder: $d_v$
- number of Fourier modes of the decoder: $k_{max}$

### C.1.6 DeepONet

Let $x := (x_1, \ldots, x_m) \in D$ be a fixed set of *sensor points*. Given an input function $u \in \mathcal{X}$, we encode it by the point values $\mathcal{E}(u) = (u(x_1), \ldots, u(x_m)) \in \mathbb{R}^m$. DeepONet is formulated in terms of two neural networks [39]: (1) a *branch-net* $\beta$, which maps the point values $\mathcal{E}(u)$ to coefficients $\boldsymbol{\beta}(\mathcal{E}(u)) = (\beta_1(\mathcal{E}(u)), \ldots, \beta_p(\mathcal{E}(u)))$, resulting in a mapping

$$\beta : \mathbb{R}^m \to \mathbb{R}^p, \quad \mathcal{E}(\overline{u}) \mapsto (\beta_1(\mathcal{E}(\overline{u})), \ldots, \beta_p(\mathcal{E}(\overline{u}))). \tag{C.5}$$

and (2) a *trunk-net* $\tau(y) = (\tau_1(y), \dots, \tau_p(y))$, which is used to define a mapping

$$\tau : U \to \mathbb{R}^p, \quad y \mapsto (\tau_1(y), \dots, \tau_p(y)). \tag{C.6}$$

While the branch net provides the coefficients, the trunk net provides the "basis" functions in an expansion of the output function of the form

$$\mathcal{G}(u)(y) = \sum_{k=1}^{p} \beta_k(\overline{u})\tau_k(y), \quad \overline{u} \in \mathcal{X}, \ y \in U, \tag{C.7}$$

with $\beta_k(\overline{u}) = \beta_k(\mathcal{E}(\overline{u}))$. The resulting mapping $\mathcal{G} : \mathcal{X} \to \mathcal{Y}, u \mapsto \mathcal{G}$ is a *DeepONet*.

In the numerical experiments, for the trunk-net we use simple feed-forward neural networks. On the other hand the branch consists of a convolutional network of the following form:

$$\mathcal{G} : \mathcal{X} \to \mathcal{Y} : \quad \mathcal{G} = Q \circ Fl \circ R_{N_{res}} \circ \cdots \circ R_1 \circ D_M \circ I_M \circ \cdots \circ D_1 \circ I_1 \tag{C.8}$$

where $I$, $D$ and $R$ are the *invariant*, *downsampling* and *ResNet* blocks defined in 2, where the downsampling in $D$ and $\Sigma$ is instead performed by average pooling with kernel size 2. The parameter $r$ in the residual block is set to 1. The output is then flattened through $Fl$ and linearly transformed by $Q : \mathbb{R}^n \to \mathbb{R}^p$, with $n$ being the number of units after flattening. The convolution is performed with a kernel of size 3 and stride 1, whereas the number of channels across the layers is

$$32 \mapsto 64 \mapsto 128 \mapsto \cdots \mapsto 2^{M-1}32.$$

The activation function is chosen as Leaky ReLU. The number of layers $L$ and units $d$ of the trunk, the number of layers $M$ and residual blocks $N_{res}$ of the branch, and the number of bases $p$, are chosen through cross-validation.

### C.1.7 Fourier Neural Operator

A *Fourier neural operator* (FNO) $\mathcal{G}$ [33] is a composition

$$\mathcal{G} : \mathcal{X} \to \mathcal{Y} : \quad \mathcal{G} = Q \circ \mathcal{L}_T \circ \cdots \circ \mathcal{L}_1 \circ R. \tag{C.9}$$

It has a "lifting operator" $u(x) \mapsto R(u(x), x)$, where $R$ is represented by a linear function $R : \mathbb{R}^{d_u} \to \mathbb{R}^{d_v}$ where $d_u$ is the number of components of the input function and $d_v$ is the "lifting dimension". The operator $Q$ is a non-linear projection, instantiated by a shallow neural network with a single hidden layer, 128 neurons and $GeLU$ activation function, such that $v^{L+1}(x) \mapsto \mathcal{G}(u)(x) = Q\left(v^{L+1}(x)\right)$.

Each *hidden layer* $\mathcal{L}_\ell : v^\ell(x) \mapsto v^{\ell+1}(x)$ is of the form

$$v^{\ell+1}(x) = \sigma\left(W_\ell \cdot v^\ell(x) + \left(K_\ell v^\ell\right)(x)\right),$$

with $W_\ell \in \mathbb{R}^{d_v \times d_v}$ a trainable weight matrix (residual connection), $\sigma$ an activation function, corresponding to GeLU, and the *non-local Fourier layer*,

$$K_\ell v^\ell = \mathcal{F}_N^{-1}\left(P_\ell(k) \cdot \mathcal{F}_N v^\ell(k)\right),$$

where $\mathcal{F}_N v^\ell(k)$ denotes the (truncated)-Fourier coefficients of the discrete Fourier transform (DFT) of $v^\ell(x)$, computed based on the given $s$ grid values in each direction. Here, $P_\ell(k) \in \mathbb{C}^{d_v \times d_v}$ is a complex Fourier multiplication matrix indexed by $k \in \mathbb{Z}^d$, and $\mathcal{F}_N^{-1}$ denotes the inverse DFT.

The lifting dimension $d_v$, the number of Fourier layers $L$ and $k_{max}$, defined in 2, are objectives of cross-validation.

### C.2 Training Details

The training of the models, including the baselines (except GT), is performed with the ADAM optimizer, with a learning rate $\eta$ for 1000 epochs and minimizing the $L^1$-loss function. We also use a step learning rate scheduler and reduce the learning rate of each parameter group by a factor $\gamma$ every epoch. We train FFNN, UNet, and DeepONet in mini-batches of size 10 and FNO and CNO in

batches of 32. A weight decay of magnitude $w$ is used. All the parameters mentioned above ($\eta, \gamma, w$) are chosen through cross-validation.

The GT models are trained with ADAM optimizer, minimizing the weighted $L^2$-loss function (see [8] implementation for clarification). Number of epochs is 1000. A learning rate scheduler is set according to a *OneCycleLR* policy (max_lr $= 5 \cdot 10^{-4}$, div_fac $= 10^4$, pct_start $= 0.3$). The selection of max_lr relies on empirical observations.

At every epoch, the relative $L^1$ error is computed on the validation set, and the set of trainable parameters resulting in the lowest error during the entire process is saved for testing. Early stopping is used to interrupt the training if the best validation error does not improve after 50 epochs.

The cross-validation is performed by running a random search over a chosen range of hyperparameters values and selecting the configuration, realizing the lowest relative $L^1$ error on the validation set. Overall, 30 hyperparameters configurations are tested for the FFNN, UNet and DeepONet , 48 to 72 configurations for GT, 24 to 48 configurations for CNO and 36 to 72 configurations for FNO. The model size (minimum and maximum number of trainable parameters) covered in this search are reported in Table 5.

The results of the random search, i.e., the best-performing hyperparameter configurations for each model and each benchmark, are reported in tables 6, 10 and 7, 11 and 12.

**Different Initialization.** After selecting the models and computing the test median errors, we proceed to train the CNO, FNO, and UNet models again using the same settings but different initializations for the model parameters (by changing the random seeds). Each model is trained for each experiment a total of 10 times. We report the means and the standard deviations of the 10 different test median errors for each benchmark experiment in the Table 4. We observe from this table that CNO is very robust with respect to random initializations, with very low standard deviation to mean ratio for all the benchmarks in the **RPB** dataset.

Table 4: Means and standard deviations for the 10 relative median $L^1$ test errors, for both in-distribution testing, for the CNO, FNO and U-Net models. The format is *mean $\pm$ std*.

|  | **CNO** | **FNO** | **UNet** |
|---|---|---|---|
| **Poisson Equation** | $0.34 \pm 0.09\%$ | $4.88 \pm 0.18\%$ | $0.76 \pm 0.16\%$ |
| **Wave Equation** | $0.63 \pm 0.06\%$ | $1.08 \pm 0.07\%$ | $1.67 \pm 0.12\%$ |
| **Smooth Transport** | $0.27 \pm 0.04\%$ | $0.34 \pm 0.03\%$ | $0.79 \pm 0.21\%$ |
| **Discontinuous Transport** | $1.06 \pm 0.04\%$ | $1.18 \pm 0.03\%$ | $1.40 \pm 0.09\%$ |
| **Allen-Cahn** | $0.67 \pm 0.09\%$ | $0.28 \pm 0.03\%$ | $1.84 \pm 0.33\%$ |
| **Navier-Stokes** | $2.91 \pm 0.08\%$ | $3.68 \pm 0.10\%$ | $3.48 \pm 0.07\%$ |
| **Darcy Flow** | $0.42 \pm 0.02\%$ | $0.90 \pm 0.08\%$ | $0.65 \pm 0.10\%$ |
| **Compressible Euler** | $0.35 \pm 0.01\%$ | $0.45 \pm 0.01\%$ | $0.39 \pm 0.01\%$ |

## C.3 Details about the description and numerical results in each benchmark

This section provides details about all the experiments that are a part of the **RPB** benchmarks of the main text.

### C.3.1 Poisson Equation

In this experiment, we study Poisson equation 4.1 with the source term given by

$$f(x, y) = \frac{\pi}{K^2} \sum_{i,j}^{K} a_{ij} \cdot (i^2 + j^2)^r \sin(\pi i x) \sin(\pi j y), \quad \forall (x, y) \in D,$$

Table 5: Minimum (Top sub-row) and maximum (Bottom sub-row) number of trainable parameters among the random-search hyperparameters configurations for all the models in every problem reported in Table 1 in main text.

|  | FFNN | GT | ResNet | UNet | DON | FNO | CNO |
|---|---|---|---|---|---|---|---|
| **Poisson Equation** | 0.3M | 8.5M | 0.1M | 0.5M | 0.8M | 0.2M | 0.5M |
|  | 8.2M | 19.1M | 10.2M | 31.0M | 48.1M | 18.9M | 26.8M |
| **Wave Equation** | 0.3M | 8.5M | 0.1M | 0.5M | 0.8M | 0.2M | 1.5M |
|  | 6.0M | 19.1M | 10.2M | 31.0M | 48.1M | 7.9M | 23.6M |
| **Smooth Transport** | 0.3M | 8.5M | 0.1M | 0.5M | 0.7M | 0.2M | 0.5M |
|  | 8.2M | 19.1M | 10.2M | 7.7M | 49.2M | 23.6M | 18.9M |
| **Discontinuous Transport** | 0.3M | 8.5M | 0.1M | 0.5M | 0.7M | 0.2M | 0.5M |
|  | 5.5M | 19.1M | 10.2M | 7.7M | 49.2M | 23.6M | 18.9M |
| **Allen-Cahn** | 0.3M | 2.1M | 0.1M | 0.5M | 1.0M | 0.9M | 0.5M |
|  | 7.1M | 19.1M | 10.2M | 31.0M | 47.9M | 65.6M | 8.4M |
| **Navier-Stokes** | 0.3M | 2.1M | 0.1M | 0.5M | 1.0M | 0.2M | 0.5M |
|  | 7.1M | 19.1M | 10.2M | 31.0M | 47.9M | 65.6M | 14.1M |
| **Darcy Flow** | 0.3M | 2.1M | 0.1M | 0.5M | 1.0M | 0.2M | 0.5M |
|  | 7.1M | 19.1M | 10.2M | 31.0M | 47.9M | 23.6M | 8.4M |
| **Compressible Euler** | 1.1M | 2.1M | 0.1M | 0.5M | 0.8M | 0.2M | 1.5M |
|  | 18.6M | 19.1M | 10.2M | 31.0M | 49.4M | 23.6M | 31.7M |

Table 6: FFNN best-performing hyperparameters configuration for different benchmark problems.

|  | $\eta$ | $\gamma$ | $w$ | $L$ | $d$ | Trainable Params |
|---|---|---|---|---|---|---|
| **Poisson Equation** | 0.0005 | 0.98 | 1e-06 | 10 | 512 | 6.6M |
| **Wave Equation** | 0.001 | 0.98 | 1e-06 | 4 | 256 | 2.3M |
| **Continuous Translation** | 0.001 | 1.0 | 0.0 | 16 | 256 | 3.1M |
| **Discontinuous Translation** | 0.0005 | 1.0 | 0.0 | 6 | 512 | 5.5M |
| **Allen-Cahn** | 0.0005 | 0.98 | 0.0 | 8 | 512 | 6.0M |
| **Navier-Stokes** | 0.001 | 1.0 | 1e-06 | 16 | 256 | 3.1M |
| **Darcy Flow** | 0.0005 | 0.98 | 1e-06 | 16 | 256 | 3.1M |
| **Compressible Euler** | 0.0005 | 1.0 | 0.0 | 16 | 32 | 1.1M |

where $K = 16$, $r = 0.5$ and $a_{ij}$ are i.i.d. uniformly distributed from $[-1, 1]$. Given the source term above, the *exact* solution $u$ of the Poisson equation is given by

$$u(x,y) = \frac{1}{\pi K^2} \sum_{i,j}^{K} a_{ij} \cdot (i^2 + j^2)^{r-1} \sin(\pi i x) \sin(\pi j y), \quad \forall (x,y) \in D.$$

During the out-of-distribution testing, we augment the number of modes to $K = 20$ and evaluate the models' ability to generalize to inputs with frequencies higher than those encountered during training. We approximate the operator $\mathcal{G}^\dagger$, which maps $f$ to $u$. An illustration of the operator $\mathcal{G}^\dagger$ is given in the Figure 28. For training purposes, we generate 1024 samples and for testing, we generate 256 samples for both in-distribution and out-of-distribution testing, by sampling the exact solution $u$ at a resolution of $64 \times 64$ points on $D = [0, 1]^2$. We also create a validation set consisting of 128 samples for model selection. The training data is normalized to the interval $[0, 1]$. The testing data is normalized with the same normalization constants as the training data. In Figure 5, we show empirical test error distributions for UNet, FNO and CNO models (in-distribution in the left Figure

Table 7: UNet best-performing hyperparameters configuration for different benchmark problems.

| | $\eta$ | $\gamma$ | $w$ | $c$ | Trainable Params |
|---|---|---|---|---|---|
| **Poisson Equation** | 0.001 | 0.98 | 0.0 | 32 | 7.8M |
| **Wave Equation** | 0.001 | 1.0 | 1e-06 | 64 | 31.0M |
| **Continuous Translation** | 0.001 | 0.98 | 1e-06 | 16 | 1.9M |
| **Discontinuous Translation** | 0.001 | 0.98 | 1e-06 | 32 | 7.8M |
| **Allen-Cahn** | 0.0005 | 0.98 | 1e-06 | 64 | 31.0M |
| **Navier-Stokes** | 0.0005 | 0.98 | 1e-06 | 64 | 31.0M |
| **Darcy Flow** | 0.001 | 0.98 | 0.0 | 32 | 7.8M |
| **Compressible Euler** | 0.001 | 0.98 | 1e-06 | 32 | 7.8M |

Table 8: ResNet best-performing hyperparameters configuration for different benchmark problems.

| | $\eta$ | $\gamma$ | $w$ | $c$ | $N_{res}$ | Trainable Params |
|---|---|---|---|---|---|---|
| **Poisson Equation** | 0.001 | 0.98 | 0.0 | 8 | 8 | 0.2M |
| **Wave Equation** | 0.001 | 0.98 | 1e-06 | 16 | 8 | 0.6M |
| **Continuous Translation** | 0.001 | 0.98 | 1e-06 | 32 | 6 | 2.0M |
| **Discontinuous Translation** | 0.001 | 0.98 | 1e-06 | 32 | 4 | 1.4M |
| **Allen-Cahn** | 0.001 | 0.98 | 1e-06 | 32 | 4 | 1.4M |
| **Navier-Stokes** | 0.001 | 0.98 | 1e-06 | 64 | 8 | 10.2M |
| **Darcy Flow** | 0.001 | 0.98 | 0.0 | 8 | 8 | 0.2M |
| **Compressible Euler** | 0.001 | 1.0 | 0.0 | 32 | 8 | 2.6M |

Table 9: DeepONet best-performing hyperparameters configuration for different benchmark problems.

| | $\eta$ | $\gamma$ | $w$ | $p$ | $L$ | $d$ | $M$ | $N_{res}$ | Trainable Params |
|---|---|---|---|---|---|---|---|---|---|
| **Poisson Equation** | 0.001 | 0.98 | 0.0 | 500 | 8 | 128 | 4 | 4 | 5.2M |
| **Wave Equation** | 0.0005 | 0.98 | 0.0 | 100 | 4 | 512 | 4 | 4 | 4.2M |
| **Continuous Translation** | 0.0005 | 0.98 | 0.0 | 500 | 8 | 128 | 4 | 0 | 2.8M |
| **Discontinuous Translation** | 0.0005 | 0.98 | 0.0 | 100 | 8 | 512 | 4 | 4 | 5.3M |
| **Allen Cahn** | 0.0005 | 0.98 | 1e-06 | 50 | 8 | 512 | 4 | 4 | 5.0M |
| **Navier Stokes** | 0.0005 | 0.98 | 1e-06 | 100 | 8 | 512 | 6 | 2 | 30.3M |
| **Darcy Flow** | 0.0005 | 0.98 | 0.0 | 500 | 8 | 512 | 4 | 4 | 7.1M |
| **Compressible Euler** | 0.0005 | 0.98 | 1e-06 | 500 | 8 | 256 | 4 | 4 | 11.7M |

and out-of-distribution in the right Figure). We show a random in-distribution testing sample and an out-of-distribution testing sample, as well as predictions made by CNO, FNO and UNet in Figure 6.

Table 10: Galerkin Transformer best-performing hyperparameters configuration for different benchmark problems.

| | $n$ | $h$ | $d$ | $L$ | $d_v$ | $k_{max}$ | Trainable Params |
|---|---|---|---|---|---|---|---|
| **Poisson Equation** | 4 | 4 | 64 | 2 | 64 | 16 | 8.6M |
| **Wave Equation** | 4 | 4 | 64 | 2 | 64 | 16 | 8.6M |
| **Continuous Translation** | 4 | 2 | 128 | 3 | 64 | 16 | 17.4M |
| **Discontinuous Translation** | 8 | 2 | 128 | 4 | 64 | 16 | 17.4M |
| **Allen-Cahn** | 2 | 4 | 128 | 2 | 32 | 16 | 2.5M |
| **Navier-Stokes** | 2 | 2 | 256 | 2 | 64 | 16 | 10.0M |
| **Darcy Flow** | 4 | 2 | 256 | 2 | 32 | 16 | 4.4M |
| **Compressible Euler** | 2 | 4 | 64 | 4 | 64 | 16 | 16.9M |

Table 11: FNO best-performing hyperparameters configuration for different benchmark problems.

| | $\eta$ | $\gamma$ | $w$ | $pad$ | $k_{max}$ | $d_v$ | $L$ | Trainable Params |
|---|---|---|---|---|---|---|---|---|
| **Poisson Equation** | 0.001 | 0.98 | 1e-6 | 0 | 16 | 16 | 5 | 0.7M |
| **Wave Equation** | 0.001 | 0.98 | 1e-6 | 0 | 20 | 16 | 4 | 0.8M |
| **Smooth Transport** | 0.001 | 0.98 | 1e-6 | 4 | 20 | 32 | 5 | 4.1M |
| **Discontinuous Transport** | 0.001 | 0.98 | 1e-6 | 4 | 16 | 32 | 5 | 2.6M |
| **Allen-Cahn** | 0.001 | 0.98 | 1e-6 | 0 | 20 | 16 | 3 | 0.6M |
| **Navier-Stokes** | 0.001 | 0.98 | 1e-6 | 0 | 16 | 128 | 5 | 42.1M |
| **Darcy Flow** | 0.001 | 0.98 | 1e-6 | 0 | 24 | 16 | 2 | 0.6M |
| **Compressible Euler** | 0.001 | 0.98 | 1e-6 | 8 | 24 | 32 | 3 | 3.6M |

Table 12: CNO best-performing hyperparameters configuration for different benchmark problems.

| | $\eta$ | $\gamma$ | $w$ | $M$ | $d_e$ | $N_{res,b}$ | $N_{res,i}$ | Trainable Params |
|---|---|---|---|---|---|---|---|---|
| **Poisson Equation** | 0.001 | 0.98 | 1e-6 | 3 | 16 | 6 | 4 | 0.7M |
| **Wave Equation** | 0.001 | 0.98 | 1e-10 | 3 | 48 | 6 | 4 | 6.6M |
| **Smooth Transport** | 0.001 | 0.98 | 1e-6 | 3 | 32 | 6 | 2 | 2.8M |
| **Discontinuous Transport** | 0.001 | 0.98 | 1e-6 | 3 | 32 | 4 | 5 | 2.5M |
| **Allen-Cahn** | 0.001 | 0.98 | 1e-6 | 3 | 48 | 8 | 4 | 3.5M |
| **Navier-Stokes** | 0.001 | 0.98 | 1e-10 | 3 | 32 | 8 | 1 | 3.3M |
| **Darcy Flow** | 0.001 | 0.98 | 1e-6 | 3 | 48 | 4 | 4 | 5.3M |
| **Compressible Euler** | 0.001 | 0.98 | 1e-10 | 4 | 48 | 8 | 1 | 7.3M |

As was already evidenced in Table 1 of the main text, Figure 5 demonstrates that CNO is clearly the

best performing model here with U–Net a distant second. FNO performs very poorly on this problem, with test errors that are more than an order of magnitude higher than CNO. A closer perusal of Figure 6 reveals that FNO approximates the multiple scales in the exact solution very poorly and this is particularly striking for the out of distribution testing example shown in this figure. On the other hand, CNO approximates the multiple frequencies in the solution very accurately.

Finally, to further investigate the poor performance of FNO, as compared to CNO, for this problem, we present in Figure 7,the averaged logarithmic amplitude spectra, which compare the ground truth, CNO, FNO, and UNet models. We see from this spectrogram that i) the ground truth solution contains multiple scales, corresponding to a range of frequencies ii) the CNO model successfully captures the complete spectra with high accuracy, iii) FNO (and to some extent UNet) resolves the underlying spectrum with quite a lot of error, particularly in the high-frequency components, perhaps attributable to aliasing errors in this case.

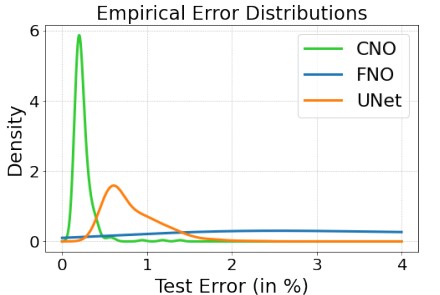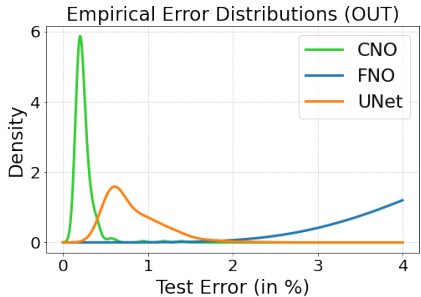

Figure 5: Poisson equation. Empirical test error distributions for UNet, FNO and CNO. Left: In-distribution testing. Right: Out-of-distribution testing.

### C.3.2 Wave Equation

In this experiment, we study Wave equation 4.3 with constant speed of propagation $c = 0.1$ and the initial condition given by 4.2 with $K = 24$ and $r = 1$. The exact solution at time $t > 0$ is given by

$$u(x, y, t) = \frac{\pi}{K^2} \sum_{i,j}^{K} a_{ij} \cdot (i^2 + j^2)^{-r} \sin(\pi i x) \sin(\pi j y) \cos\left(c \pi t \sqrt{i^2 + j^2}\right), \quad \forall (x, y) \in D.$$

The objective is to approximate the operator $\mathcal{G}^\dagger : f \mapsto u(\cdot, T = 5)$. An illustration of $\mathcal{G}^\dagger$ is given in Figure 29. During the out-of-distribution testing, we decrease the decay parameter to $r = 0.85$. This adjustment changes the ratio between the amplitudes of different modes, which alters the dynamics

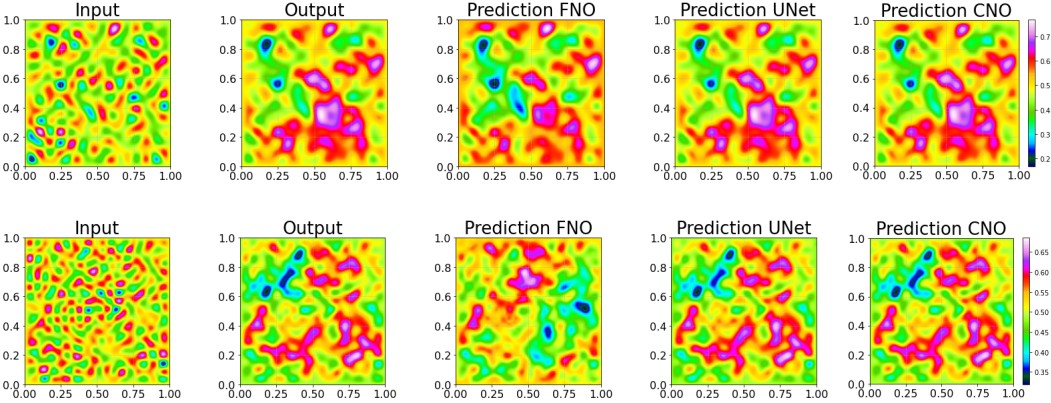

Figure 6: Poisson equation. Exact and predicted coefficients for an in-distribution (top row) and an out-of-distribution (bottom row) samples and for different models (columns). From left to right: input, output (ground truth), FNO, UNet and CNO.

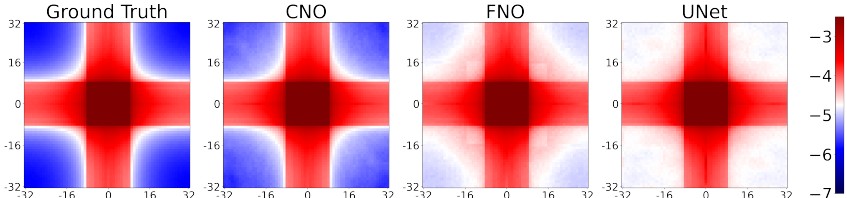

Figure 7: Poisson equation. Averaged logarithmic amplitude spectra comparing Ground Truth, CNO, FNO and UNet.

of the solution. For the training set, we generate a total of 512 samples. In addition, we generate 256 samples for both in-distribution and out-of-distribution testing, all by sampling the above exact solution at a resolution of $64 \times 64$. Furthermore, we create a validation set comprising 128 samples. The training data is normalized to the interval $[0, 1]$. The testing data is normalized with the same normalization constants as the training data.

In Figure 8, we present the empirical test error distributions for UNet, FNO and CNO models during in-distribution and out-of-distribution testing. We also show a random in-distribution testing sample and a random out-of-distribution testing sample, as well as predictions made by CNO, FNO and UNet in Figure 9. Both these figures demonstrate that CNO is the best performing model in this case, reinforcing the conclusion of Table 1 of the main text.

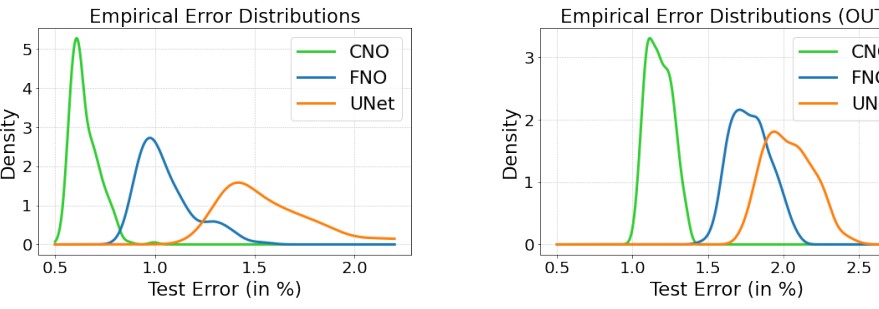

Figure 8: Wave equation. Empirical test error distributions for UNet, FNO and CNO. Left: In-distribution testing. Right: Out-of-distribution testing.

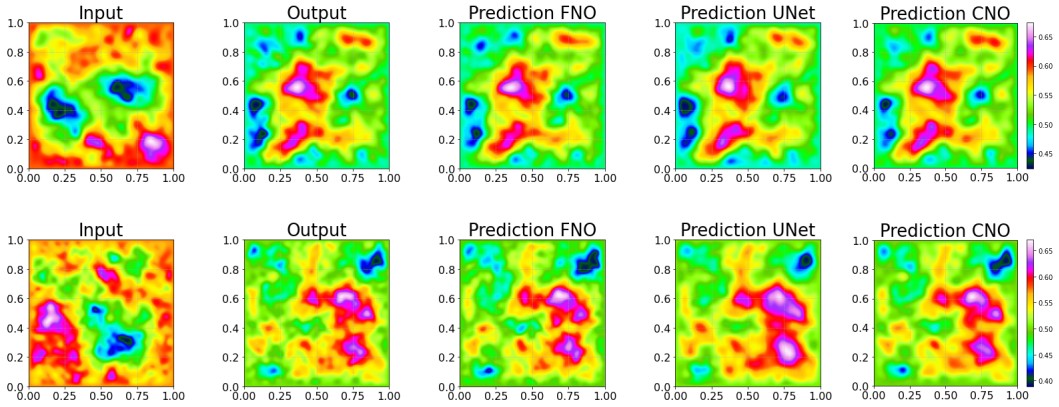

Figure 9: Wave equation. Exact and predicted coefficients for an in-distribution (top row) and an out-of-distribution (bottom row) samples and for different models (columns). From left to right: input, output (ground truth), CNO, FNO and UNet.

### C.3.3 Transport Equation

In this experiment, we study Transport equation 4.4. We fix the velocity field to $v = (v_x, v_y) = (0.2, -0.2)$ leading to solution $u(x, y, t) = f(x - v_x t, y - v_y t)$. We conduct two different experiments, i.e., Smooth Transport and Discontinuous Transport. In both cases, the goal is to approximate the operator $\mathcal{G}^\dagger : f \mapsto u(\cdot, T = 1)$. Moreover, in both cases, we generate 512 training samples, 256 validation samples and 256 in-distribution and out-of-distribution testing samples, all from the exact solution. Each sample is normalized to the interval $[0, 1]$.

**Smooth Transport.** In this case, the data takes form of of a radially symmetric Gaussian. The data is drawn from a Gaussian distribution with centers randomly and uniformly drawn from $(0.2, 0.4)^2$ and corresponding variance drawn uniformly from $(0.003, 0.009)$. Formally, the initial conditions are given by

$$f(\underline{x}) = \frac{1}{\sqrt{(2\pi)^2 \det(\Sigma)}} \exp\left(-\frac{1}{2}(\underline{x} - \mu)^T \Sigma^{-1}(\underline{x} - \mu)\right), \quad \underline{x} = (x, y), \quad \mu = (\mu_x, \mu_y),$$

where $\Sigma = \sigma I$ such that $\sigma \sim \mathcal{U}(0.003, 0.009)$ and $\mu_x, \mu_y \sim \mathcal{U}(0.2, 0.4)$. Here, $I$ is the identity matrix and $\mathcal{U}(\cdot)$ is the uniform distribution. Finally, each initial condition is normalized to $(0, 1)$.

For *out-of-distribution* testing, the centers of the Gaussian inputs are sampled uniformly from $(0.4, 0.6)^2$ (i.e. $\mu_x, \mu_y \sim \mathcal{U}(0.4, 0.6)$). The data is generated at $64 \times 64$ resolution. An illustration of the operator $\mathcal{G}^\dagger$ for the Smooth Transport experiment is shown in Figure 30. We show empirical test error distributions for UNet, FNO and CNO models (in-distribution and out-of-distribution testing) in Figure 10. We show a random in-distribution testing sample and an out-of-distribution testing sample, as well as predictions made by CNO, FNO and UNet in Figure 11. The figures reinforce the conclusions drawn from Table 1 i.e., CNO is slightly superior to UNet and FNO for in-distribution testing. However, there is a significant advantage for CNO over UNet on out-of-distribution testing. On the other hand, FNO generalizes poorly out-of-distribution, as clearly seen from the sample shown in Figure 11. Similarly, DeepONet and FFNN are even poorer in terms of their generalization abilities, justfitying the very high errors seen in Table 1. An example of this very poor generalization for DeepONet and FFNN can be seen in Figure 12.

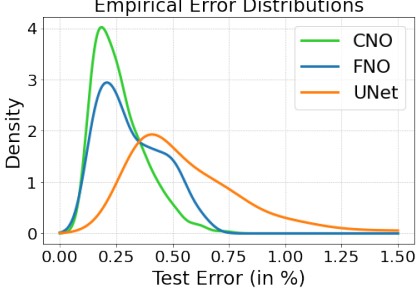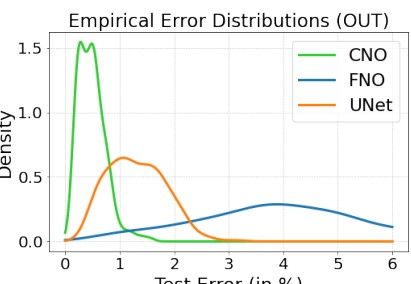

Figure 10: Smooth Transport. Empirical test error distributions for UNet, FNO and CNO. Left: In-distribution testing. Right: Out-of-distribution testing.

**Discontinuous Transport.** In this case, initial data in the form of the indicator function of radial disk with centers, uniformly drawn from $(0.2, 0.4)^2$ and radii uniformly drawn from $(0.1, 0.2)$. For *out-of-distribution* testing, the centers of the disk are drawn uniformly from $(0.4, 0.6)^2$. Formally, the initial conditions are given by

$$f(\underline{x}) = \mathbb{1}_{S_r(\mu)}(\underline{x}), \quad \underline{x} = (x, y), \quad \mu = (\mu_x, \mu_y),$$

where $r \sim \mathcal{U}(0.1, 0.2)$ and $\mu_x, \mu_y \sim \mathcal{U}(0.2, 0.4)$. Also, $\mathbb{1}.$ is an indicator function and $S_r(\mu)$ is the sphere of radius $r$ with the center $\mu$, defined by

$$S_r(\mu) = \{\underline{x} : ||\underline{x} - \mu||_2 \leq r\}.$$

Note that discontinuous data has infinite spectral content, so the aliasing error is always present when the data is sampled. For that reason, we first generate the samples at $128 \times 128$ resolution, to reduce

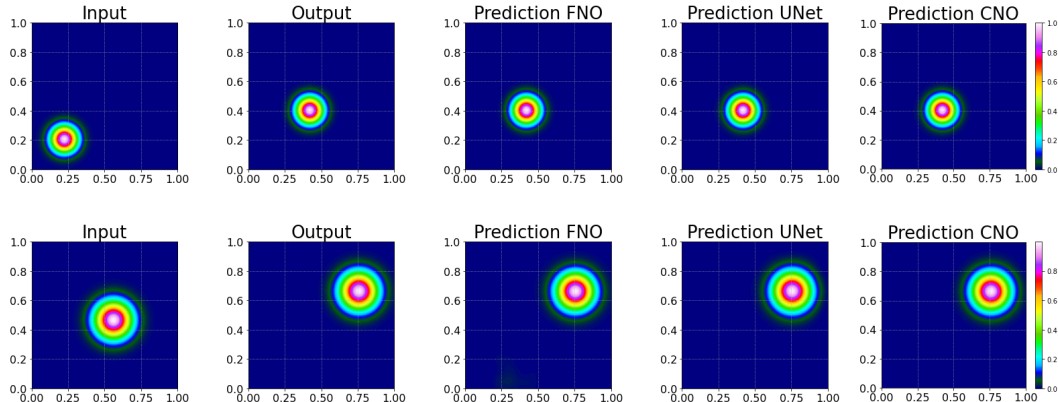

Figure 11: Smooth Transport. Exact and predicted coefficients for an in-distribution (top row) and an out-of-distribution (bottom row) samples and for different models (columns). From left to right: input, output (ground truth), CNO, FNO and UNet.

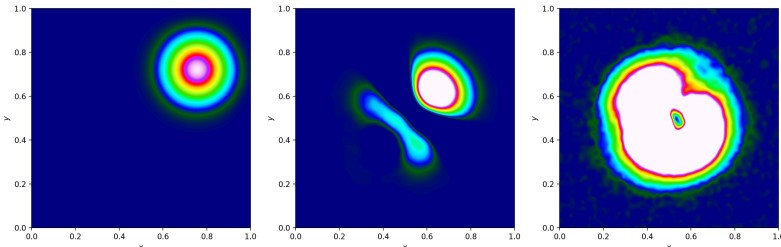

Figure 12: Smooth Transport. An out-of-distribution sample and predictions for DeepONet and FFNN. From left to right: output (ground truth), DeepONet and FFNN.

the aliasing error that emerges in data generation. We get our actual samples by downsampling the generated data in the frequency domain to the resolution $64 \times 64$. As the Gibbs phenomenon is strongly present when discontinuous data is downsampled in this way, we reduce the impact of this phenomenon by applying a Gaussian filter with a standard deviation $\sigma = 1.75$ to the generated samples, before downsampling them to the final resolution. An example of a random sample with horizontal cut plots is shown in the Figure 13.

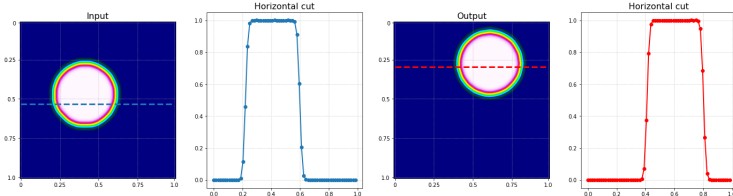

Figure 13: Discontinuous Transport. An example with horizontal cut plots of the disks.

We also plot empirical test error distributions for UNet, CNO and FNO models (in-distribution and out-of-distribution testing) in Figure 14. We plot a random in-distribution testing sample and an out-of-distribution testing sample, as well as predictions made by CNO, FNO and UNet in Figure 15. These figures clearly reinforce the conclusions from Table 1 that UNet performs as good as the CNO on out-of-distribution testing. On the other hand, FNO, DeepONet, FFNN and GT (in that order) generalize very poorly as they fail to be translation equivariant.

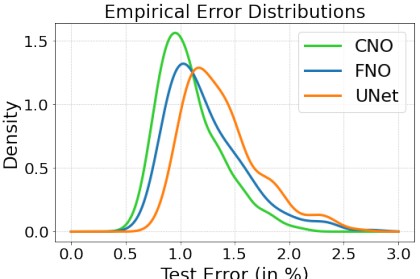 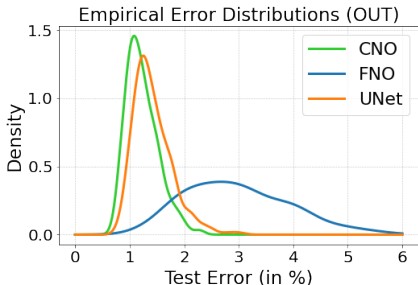

Figure 14: Discontinuous Transport. Empirical test error distributions for UNet, FNO and CNO. Left: In-distribution testing. Right: Out-of-distribution testing.

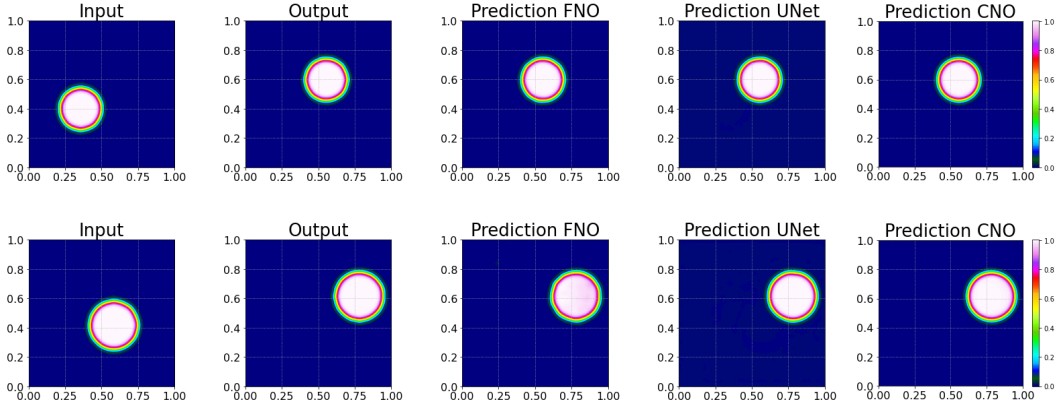

Figure 15: Discontinuous Transport. Exact and predicted coefficients for an in-distribution (top row) and an out-of-distribution (bottom row) samples and for different models (columns). From left to right: input, output (ground truth), CNO, FNO and UNet.

### C.3.4 Allen-Cahn Equation

In this experiment, we study Allen-Cahn equation 4.5 with fixed reaction rate $\varepsilon = 220$ and initial condition given by 4.2 with $K = 24$ and $r = 1$. The goal is to approximate the operator $\mathcal{G}^\dagger : f \mapsto u(\cdot, T = 0.0002)$ (see Figure 32 for illustrations).

As exact solutions are no longer available, we generate the training and test data using a standard finite difference discretization of the Allen-Cahn equation. We uniformly discretize space at the resolution $s^2 = 64 \times 64$ and set $\Delta x = 1/s$. As we are using an explicit method, we uniformly discretize the time domain with the time step $\Delta t \approx 5.47 \cdot 10^{-7}$ and set $N = \lfloor T/\Delta t \rfloor + 1$. We denote $U_{i,j}^n = u(i\Delta x, j\Delta x, n\Delta t)$ for $i, j = 0, 1, \ldots, s$ and $n = 0, 1, \ldots, N$. Additionally, we also add the zero-valued ghost cells at the boundaries. The Finite Difference scheme is given by

$$U_{i,j}^{n+1} = U_{i,j}^n + \frac{\Delta t}{\Delta x}\left(U_{i+1,j}^n + U_{i,j+1}^n + U_{i-1,j}^n U_{i,j-1}^n - 4U_{i,j}^n\right) - \Delta t \varepsilon^2 U_{i,j}^n \left(U_{i,j}^n \cdot U_{i,j}^n - 1\right),$$

for $i, j = 0, 1, \ldots, s$ and $n = 0, 1, \ldots, N$. With our choice of $\Delta t$, the CFL condition $\Delta t < \frac{(\Delta x)^2}{2\varepsilon}$ is satisfied. We generate 256 training samples, 128 validation samples and 128 in-distribution and out-of-distribution testing samples, all at the $64 \times 64$ resolution. The training data is normalized to the interval $[0, 1]$. The testing data is normalized with the same normalization constants as the training data. In Figure 16, we present the empirical test error distributions for UNet, FNO and CNO models during in-distribution and out-of-distribution testing. We also plot a random in-distribution testing sample and an out-of-distribution testing sample, as well as predictions made by CNO, FNO and UNet in Figure 17. Again, these figures reinforce the conclusions of Table 1 as FNO is marginally superior to CNO and UNet on in-distribution testing whereas UNet is the best model on out-of-distribution testing.

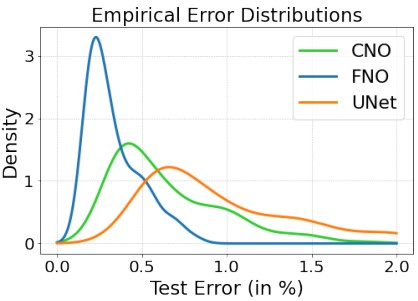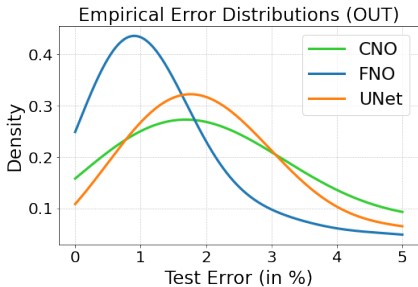

Figure 16: Allen-Cahn equation. Empirical test error distributions for UNet, FNO and CNO. Left: In-distribution testing. Right: Out-of-distribution testing.

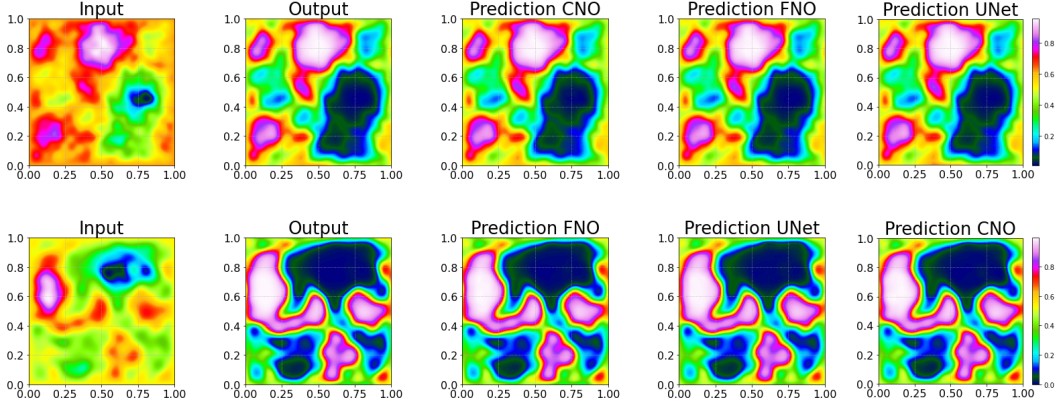

Figure 17: Allen-Cahn equation. Exact and predicted coefficients for an in-distribution (top row) and an out-of-distribution (bottom row) samples and for different models (columns). From left to right: input, output (ground truth), CNO, FNO and UNet.

### C.3.5 Navier-Stokes

In this experiment, we study a motion of an incompressible fluid with high Reynolds number. We study Navier-Stokes equations 4.6 in the torus $D = \mathbb{T}^2$ with periodic boundary conditions and, for stabilization, viscosity $\nu = 4 \times 10^{-4}$ only applied to high-enough Fourier modes. We take as initial conditions

$$u_0(x, y) = \begin{cases} \tanh\left(2\pi \frac{y - 0.25}{\rho}\right) & \text{for } y + \sigma_\delta(x) \leq \frac{1}{2} \\ \tanh\left(2\pi \frac{0.75 - y}{\rho}\right) & \text{otherwise} \end{cases} \tag{C.10}$$
$$v_0(x, y) = 0$$

where $\sigma_\delta : [0, 1] \to \mathbb{R}$ is a perturbation of the initial data given by

$$\sigma_\delta(x) = \delta \sum_{k=1}^{p} \alpha_k \sin(2\pi k x - \beta_k). \tag{C.11}$$

The random variables $\alpha_k$ and $\beta_k$ are i.i.d. uniformly distributed on $[0, 1]$ and $[0, 2\pi]$ respectively. The parameters $\delta$ and $p$ are chosen to be $\delta = 0.025$ and $p = 10$. For the smoothing parameter we choose $\rho = 0.1$. (see Figure 33 for illustrations). For the out-of-distribution experiments, we reduced $\rho$ to $\rho = 0.09$ and shifted the location of the shear layers towards the middle of the domain so that they were located at $y = 0.3$ and $y = 0.7$ instead of $y = 0.25$ and $y = 0.75$ like in the original initial condition.

Fix a mesh width $\Delta = \frac{1}{N}$ for some $N \in \mathbb{N}$. We consider the following discretization of the Navier-Stokes equations 4.6 in the Fourier domain

$$\begin{cases} \partial_t u^\Delta + \mathcal{P}_N(u^\Delta \cdot \nabla u^\Delta) + \nabla p^\Delta & = \varepsilon_N |\nabla|^{2s}(Q_N * u^\Delta) \\ \nabla \cdot u^\Delta & = 0 \\ u^\Delta|_{t=0} & = \mathcal{P}_N u_0 \end{cases} \quad \text{(C.12)}$$

where $\mathcal{P}_N$ is the spatial Fourier projection operator mapping a function $f(x,t)$ to its first $N$ Fourier modes: $\mathcal{P}_N = \sum_{|k|_\infty \leq N} \widehat{f}_k(t)e^{ik\cdot x}$. We additionally have the hyperviscosity parameter $s \geq 1$ which can be used to dampen the higher Fourier modes strongly, thus allowing for a larger part of the spectrum to be free of numerical dissipation. The artificial viscosity term we use for the stabilization of the solver consists of a resolution-dependent viscosity $\varepsilon_N$ and a Fourier multiplier $Q_N$ controlling the strength at which different Fourier modes are dampened. This allows us to not dampen the low frequency modes, while applying some diffusion to the problematic higher frequencies. The Fourier multiplier $Q_N$ is of the form

$$Q_N(\mathbf{x}) = \sum_{\mathbf{k} \in \mathbb{Z}^d, |\mathbf{k}| \leq N} \widehat{Q}_\mathbf{k} e^{i\mathbf{k}\cdot\mathbf{x}}. \quad \text{(C.13)}$$

In order to have convergence, the Fourier coefficients of $Q_N$ need to fulfill [27], [64], [65]

$$\widehat{Q}_k = 0 \text{ for } |k| \leq m_N, 1 - \left(\frac{m_N}{|k|}\right)^{\frac{2s-1}{\theta}} \leq \widehat{Q}_k \leq 1 \quad \text{(C.14)}$$

where we have introduced an additional parameter $\theta > 0$. The quantities $m_N$ and $\varepsilon_N$ are required to scale as

$$m_N \sim N^\theta, \varepsilon_N \sim \frac{1}{N^{2s-1}}, 0 < \theta < \frac{2s-1}{2s}. \quad \text{(C.15)}$$

For the experiment described here, we choose $s = 1, m_N = \sqrt{N}, \varepsilon_N = \frac{0.05}{N}$, and $N = 128$. This gives rise to the viscosity $\nu \approx 4 \cdot 10^{-4}$ mentioned above.

Applying the Fourier projection operator to the PDE C.12 causes the solutions to be bandlimited functions and therefore they only have finitely many nonzero basis function coefficients (at most $N$). By writing the above discretization in the Fourier basis, we transform the spatial derivatives into multiplications with the wave vectors $k$ and obtain

$$\partial_t \widehat{u}_k + ik^T \cdot \widehat{B}_k + ik\widehat{p}_k = -\nu|k|^2 \widehat{u}_k \quad \text{(C.16)}$$

where we have substituted $B = u \otimes u$. By requiring $\widehat{u}_k$ (and $\partial_t \widehat{u}_k$) to be divergence free, we can compute the pressure $\widehat{p}_k$ to be

$$\widehat{p}_k = -\frac{k^T \cdot \widehat{B}_k \cdot k}{|k|^2}. \quad \text{(C.17)}$$

Note that the pressure can be computed from local quantities only. This is in contrast to numerical methods solving the equations in physical space where the pressure is obtained as the solution to a Poisson equation. Finally, we can solve the incompressible Euler equations by computing

$$\partial_t \widehat{u}_k + \left(\text{Id} - \frac{kk^T}{|k|^2}\right) \cdot \widehat{b}_k = -\nu|k|^2 \widehat{u}_k \quad \text{(C.18)}$$

where $\widehat{b}_k = ik^T \cdot \widehat{B}_k$. Timestepping is done using a third-order strong stability preserving Runge-Kutta scheme (SSPRK3)

$$\begin{aligned} u^{(1)} &= u(t) + \Delta t \partial_t u(t) \\ u^{(2)} &= \frac{3}{4}u(t) + \frac{1}{4}u^{(1)} + \frac{1}{4}\Delta t \partial_t u^{(1)} \\ u(t + \Delta t) &= \frac{1}{3}u(t) + \frac{2}{3}u^{(2)} + \frac{2}{3}\Delta t \partial_t u^{(2)}. \end{aligned} \quad \text{(C.19)}$$

Note that through the construction of the pressure field, the numerical scheme is not exactly divergence-free. It merely preserves the divergence of the initial conditions $u_0$. We therefore implicitly project all the initial conditions onto divergence free vector fields. This operation is described by

the Leray projection $\mathbb{P} : L^2(\Omega) \to \{u \in L^2(\Omega) \mid \text{div } u = 0\}$ mapping $u \mapsto u - \nabla\Delta^{-1}(\text{div } u)$. In Fourier space, this can again be simplified to the local equation

$$\mathbb{P}\widehat{u}_k = \left(\text{Id} - \frac{kk^T}{|k|^2}\right) \cdot \widehat{u}_k. \tag{C.20}$$

For the training set, we generate a total of 750 samples. In addition, we generate 128 samples for validation set, in-distribution and out-of-distribution testing. To generate the training and test data, we simulate the Navier-Stokes equations with a spectral viscosity method on a $128 \times 128$ resolution and downsample the data to a $64 \times 64$ resolution. The goal is to learn the operator mapping the initial velocity to velocity at $T = 1$. The training data is normalized to the interval $[0, 1]$. The testing data is normalized with the same normalization constants as the training data. In Figure 18, we present the empirical test error distributions for UNet, FNO and CNO models during in-distribution and out-of-distribution testing. We also plot a random in-distribution testing sample and an out-of-distribution testing sample, as well as predictions made by CNO, FNO and UNet in Figure 19. These figures demonstrate that CNO is clearly the best performing model, for both in-distribution and out-of-distribution testing, outperforming UNet and FNO significantly. Moreover, given the highly multiscale nature of this problem (see Figure 2 of Main Text for spectrograms), it is not surprising that the errors with all the models are higher than in the other **RPB** benchmarks.

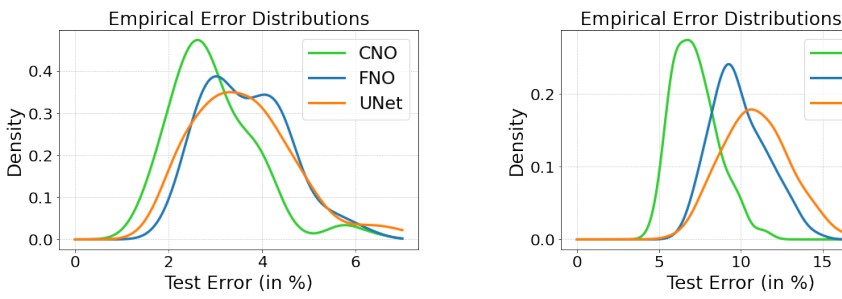

Figure 18: Navier-Stokes equations. Empirical test error distributions for UNet, FNO and CNO. Left: In-distribution testing. Right: Out-of-distribution testing.

### C.3.6 Darcy Flow

Steady-state Darcy flow is modeled by a PDE 4.7. The solution operator $\mathcal{G}^\dagger : a \mapsto u$ maps the diffusion coefficient $a$ (represented as a push forward of a Gaussian process) to the solution $u$. In-distribution and out-of-distribution samples differ in the length scales of the Gaussian process

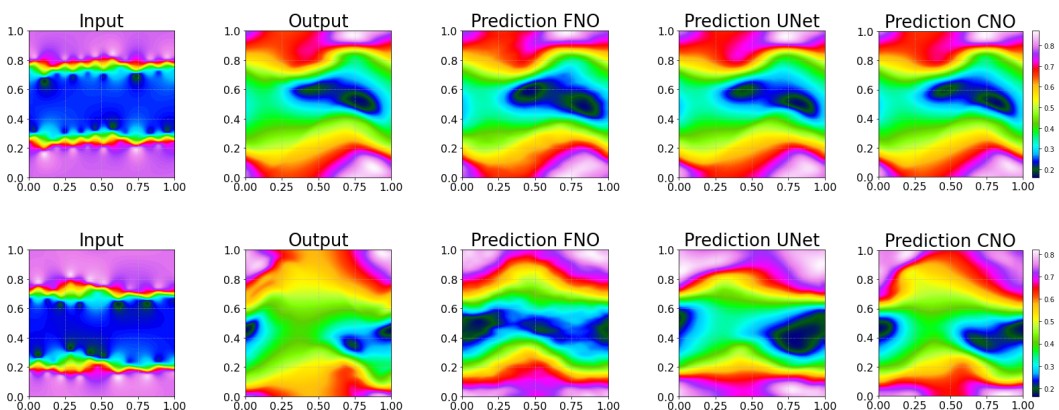

Figure 19: Navier-Stokes equations. Exact and predicted coefficients for an in-distribution (top row) and an out-of-distribution (bottom row) samples and for different models (columns). From left to right: input, output (ground truth), CNO, FNO and UNet.

in 4.8. We chose the length scale $l = 0.1$ for the in-distribution testing and $l = 0.05$ for the out-of-distribution testing. We generate 256 training samples. In addition, we generate 128 samples for validation set, in-distribution and out-of-distribution testing. The resolution of the data is $64 \times 64$.

In Figure 20, we show the empirical test error distributions for UNet, FNO and CNO models during in-distribution and out-of-distribution testing. We show an in-distribution and out-of-distributions predictions made by CNO, FNO and UNet in Figure 21. The CNO model is the best-performing model in this experiment in both in-distribution and out-of-distribution testing.

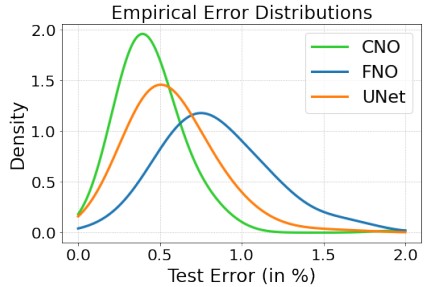 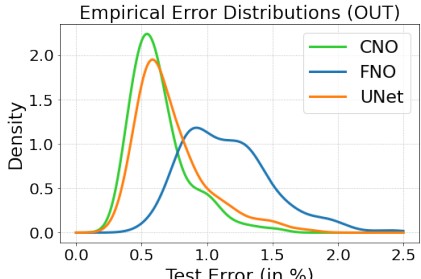

Figure 20: Darcy Flow. Empirical test error distributions for UNet, FNO and CNO. Left: In-distribution testing. Right: Out-of-distribution testing.

### C.3.7 Flow past airfoils

The flow past the airfoil is modeled by the two-dimensional compressible Euler equations

$$u_t + \text{div } F(u) = 0, \; u = [\rho, \rho v, E]^\perp, \; F = [\rho v, \rho v \otimes v + p\mathbf{I}, (E + p)]v]^\perp, \tag{C.21}$$

with density $\rho$, velocity $v$, pressure $p$ and total Energy $E$ related by the ideal gas equation of state:

$$E = \frac{1}{2}\rho|u|^2 + \frac{p}{\gamma - 1}, \tag{C.22}$$

where $\gamma = 1.4$. Additional important variables associated with the flow include the speed of sound $a = \sqrt{\frac{\gamma p}{\rho}}$ and the Mach number $M = \frac{|u|}{a}$.

We follow standard practice in aerodynamic shape optimization and consider a reference airfoil shape with upper and lower surface of the airfoil are located at $(x, y_{\text{ref}}^{\text{U}}(x/c))$ and $(x, y_{\text{ref}}^{\text{L}}(x/c))$ where $c$ is

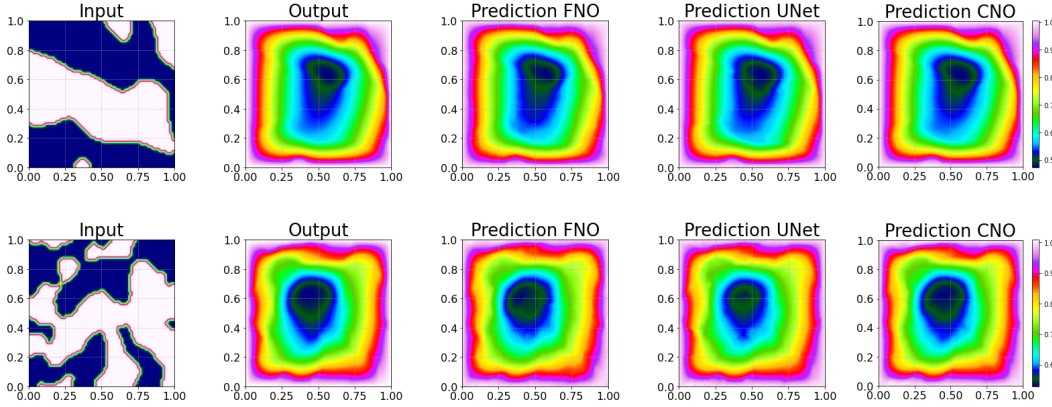

Figure 21: Darcy Flow. Exact and predicted coefficients for an in-distribution (top row) and an out-of-distribution (bottom row) samples and for different models (columns). From left to right: input, output (ground truth), FNO, UNet and CNO.

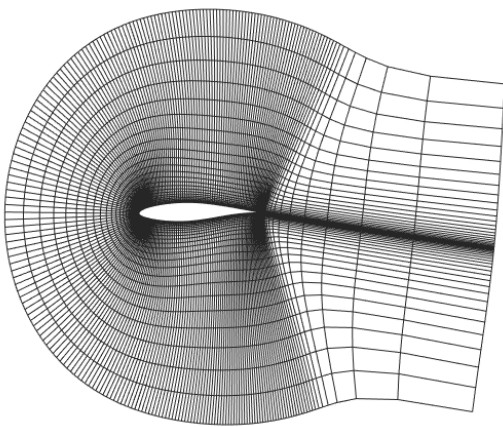

Figure 22: Elliptic mesh for the airfoil problem

the chord length and $y_{\text{ref}}^{\text{U}}$ and $y_{\text{ref}}^{\text{L}}$ corresponding to the well-known RAE2822 airfoil. The reference shape is then perturbed by *Hicks-Henne Bump functions* [40] :

$$y^{\text{L}}(\xi) = y_{\text{ref}}^{\text{L}}(\xi) + \sum_{i=1}^{10} a_i^{\text{L}} B_i(\xi), \quad y^{\text{U}}(\xi) = y_{\text{ref}}^{\text{U}}(\xi) + \sum_{i=1}^{10} a_i^{\text{U}} B_i(\xi),$$

$$B_i(\xi) = \sin^3(\pi \xi^{q_i}), \quad q_i = \frac{\ln 2}{\ln 14 - \ln i}, \quad \xi = \frac{x}{c},$$

$$a_i^{\text{L}} = 2(\psi_i - 0.5)(i+1) \times 10^{-3}, \quad a_i^{\text{U}} = 2(\psi_{i+10} - 0.5)(11-i) \times 10^{-3}, \quad i = 1, ..., 10$$

with $\psi \in [0,1]^d$.

We can now formally define the airfoil shape as $\mathcal{S} = \{(x,y) \in D : x \in [0,c], y^L \leq y \leq y^U\}$ and accordingly the shape function $f = \chi_{[\mathcal{S}]}(x,y)$, with $\chi$ being the *characteristic function*. The underlying operator of interest $\mathcal{G}^\dagger : f \mapsto \rho$ maps the shape function $f$ into the density of the flow at steady state of the compressible Euler equations.

The equations are solved with the solver NUWTUN on $243 \times 43$ elliptic mesh (Fig.22) given the following free-stream boundary conditions,

$$T^\infty = 1, \quad M^\infty = 0.729, \quad p^\infty = 1, \quad \alpha = 2.31^\circ.$$

The data is ultimately interpolated onto a Cartesian grid of dimensions $128 \times 128$ on the underlying domain $D = [-0.75, 1.75]^2$, and unit values are assigned to the density $\rho(x,y)$ for all $(x,y)$ in the set $\mathcal{S}$.

The shapes of the training data samples correspond to 20 bump functions, with coefficients $\psi$ sampled uniformly from $[0,1]^{20}$. Out-of-distribution testing is performed with 30 bump functions. During the training and evaluation processes, the difference between the learned solution and the ground truth is exclusively calculated for the points $(x,y)$ that do not belong to the airfoil shape $\mathcal{S}$.

We generate 750 samples for the training set and 128 samples for validation set, in-distribution testing set and out-of-distribution testing set. In this experiment, the data is *not* normalized. In Figure 23, we show the empirical test error distributions for UNet, FNO and CNO models during in-distribution and out-of-distribution testing. We also show a random in-distribution testing sample and an out-of-distribution testing sample, as well as predictions made by CNO, FNO and UNet in Figure 24. The latter figure clearly shows the superiority of CNO and UNet over FNO when it comes to out-of-distribution testing.

### C.3.8 On the Choice of the RPB benchmarks.

As noted in the main text, the rationale for the inclusion of benchmark experiments in the **RPB** dataset presented here is three-fold. First, we would like to span a variety of PDEs, ranging from linear elliptic (Poisson) to linear hyperbolic (wave, transport) to nonlinear parabolic (Allen-Cahn) to

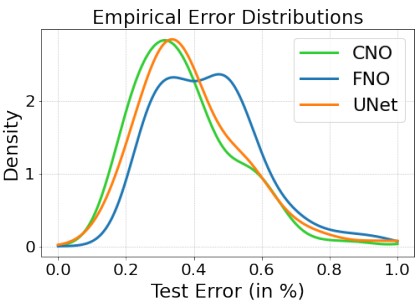 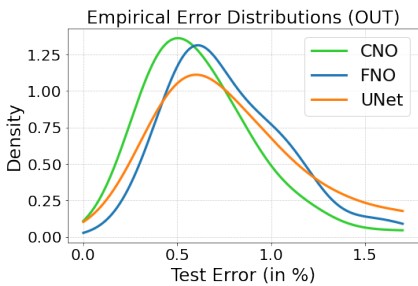

Figure 23: Airfoil experiment. Empirical test error distributions for UNet, FNO and CNO. Left: In-distribution testing. Right: Out-of-distribution testing.

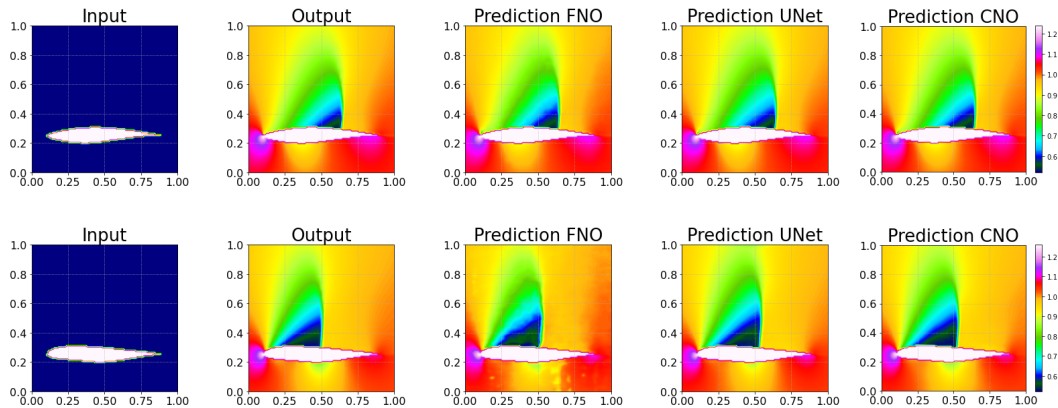

Figure 24: Airfoil experiment. Exact and predicted coefficients for an in-distribution (top row) and an out-of-distribution (bottom row) samples and for different models (columns). From left to right: input, output (ground truth), CNO, FNO and UNet.

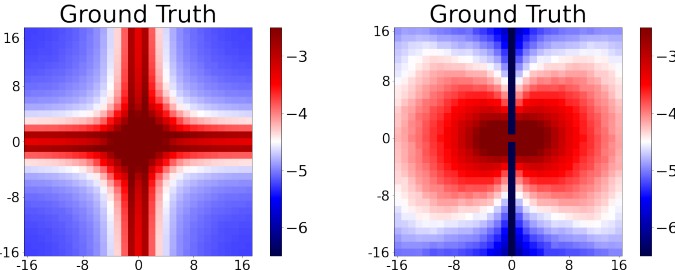

Figure 25: Comparison of the 32 central frequencies of averaged logarithmic amplitude spectra for the two Navier-Stokes experiments. Left: Old NS experiment. Right: Thin shear layer experiment.

nonlinear hyperbolic (Compressible Euler) to non-local advection-diffusion (Incompressible Navier-Stokes). Second, we would like the underlying data to be readily available for rapid prototyping and reproducibility. This limits the use of three-dimensional data-sets as data access can be cumbersome. This requirement also leads us to prioritize problems with available analytical solutions. Finally, the selected benchmarks should be *sufficiently computationally complex* such that traditional numerical methods for approximating them are expensive and there is a potential pay-off for the design of efficient machine learning based surrogates. This criterion rules out one-dimensional (in space) problems as traditional numerical methods are very fast in this case on modern computers and there is little reason to discard them for ML surrogates. Even among two-dimensional problems, one has to be careful in selecting appropriate benchmarks to ensure that they entails sufficient computational complexity.

We illustrate this issue by comparing and contrasting two possible benchmarks. First, we consider a *Navier-Stokes* data-set, considered in [33] and widely used in the recent literature on machine learning for PDEs. In this problem, the incompressible Navier-Stokes equations (4.6) are recast in the so-called *velocity-vorticity* formulation by considering the vorticity $\omega = \nabla \times u$ of the fluid. In two space dimensions, the following evolution equation for the vorticity can be readily derived from (4.6),

$$\omega_t + (u \cdot \nabla)\omega = \nu \Delta \omega, \quad \omega(0, \cdot) = \omega_0. \tag{C.23}$$

We consider the above evolution of the vorticity with periodic boundary conditions. The underlying solution operator maps the initial vorticity $\omega_0$ to the vorticity $\omega(\cdot, T)$ at a final time $T$. Following [33], we choose the initial conditions $\omega_0 \sim \mu$ where $\mu = \mathcal{N}(0, 7^{\frac{3}{2}}(-\Delta + 49\text{Id})^{-2.5})$ and extend (C.23) with a forcing term $f(x) = 0.1(\sin(2\pi(x_1 + x_2)) + \cos(2\pi(x_1 + x_2))$. Furthermore, the viscosity is chosen to be $\nu = 10^{-3}$. To generate the training and test data, we use a spectral method such as the one suggested in [27] and references therein. A rough estimate on the computational complexity of this problem can already be formed by observing Figure 25 (Left) where we present the averaged logarithmic amplitude spectra corresponding to the ground truth output (vorticity at time $T = 30$ as considered in [33]). We clearly see from this figure that only very few frequency modes (2-3) in each direction have relatively high amplitude and the spectrum decays quite fast for higher frequencies. Thus, this problem could be potentially approximated to high accuracy on fairly coarse grids.

To provide a quantitative elaboration of the above argument, we write $u_i^{N_f} = \mathcal{P}_{N_f}(u_i)$ where $u_i$ is the solution corresponding to the $i$-th drawn initial conditions and $\mathcal{P}_N$ is the spatial Fourier projection operator mapping a function $f(x, t)$ to its first $N$ Fourier modes: $\mathcal{P}_N = \sum_{|k|_\infty \leq N} \widehat{f}_k(t) e^{ik \cdot x}$. For each sample $u_i$ we compute the relative $L^1$error against the downsampled solution $u_i^{N_f}$. This provides us with an estimate how many Fourier modes need to be accurately approximated in order to achieve reasonable errors. The supremum and median of the errors over 128 samples, at time $T = 30$, are plotted in Figure 26. One can observe from this figure that even after $t = 30$ time units, only a maximum of 20 Fourier modes (in each direction) are needed to approximate the solution with an error of approximately $1\%$. Hence, a standard numerical method would only need to simulate it on a grid of $20 \times 20$ points will suffice in order to achieve the same error. Consequently, the time requirements for solving the problem on very coarse mesh with traditional spectral or finite difference methods are in the range of $10^{-3}$ seconds or lower. In contrast, we tested both FNO and CNO on this dataset to obtain test errors of $1.15\%$ and $0.96\%$, respectively. Moreover, the inference time for both FNO and CNO in this case are of the order of $10^{-4}$ secs on a NVIDIA qaudro t2000 GPU. Thus to achieve similar test errors, FNO and CNO are atmost only one order of magnitude faster than a traditional numerical method. Given the training time and data generation overheads, it is clear that there is very little payoff on using such a relatively simple two-dimensional problem as a benchmark for ML surrogates for PDEs.

On the other hand, we perform exactly the same analysis for the *thin shear layer* problem for the incompressible Navier-Stokes equation that is described in the main text. First, from Figure 25 (Right), we see that the ground truth output (horizontal velocity at time $T = 1$) has much more of a multiscale structure than in the previous experiment (compare with Figure 25 (Left)) with at least non-trivial frequencies upto 32 modes, suggesting that it is much more challenging to approximate it numerically. This is indeed verified from Figure 26 (Right) where we present the averaged (over 128 samples) $L^1$-error for the velocity as a function of the number of modes to observe that almost 100 Fourier modes are needed to get an $L^1$-error of $2\%$. This corresponds to a $100 \times 100$ spatial grid and even a state-of-the-art GPU implementation of the spectral viscosity method of [27] would require $10^{-1}$ seconds of run time. When compared to a CNO inference time of $10^{-4}$ secs for an error

of approximately 3%, we see that the ML surrogate (CNO) provides *three orders of magnitude* or more of speedup in this case, making its deployment worthwhile. Thus, we have demonstrated the rationale for the choice of this benchmark, rather than the Navier-Stokes benchmark of [33], in our proposed **RPB** dataset.

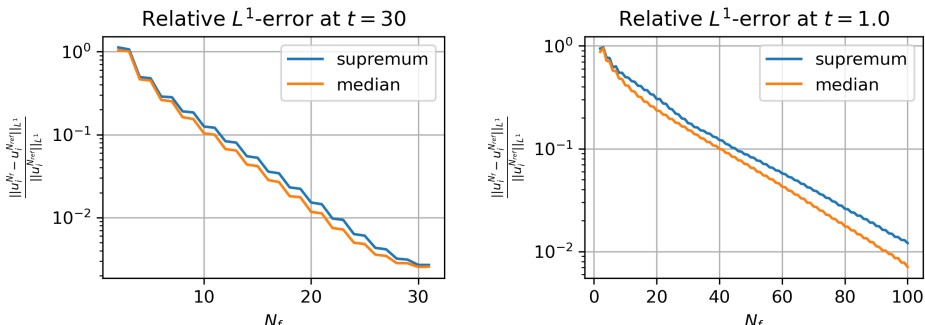

Figure 26: Relative $L^1$-error of the vorticity experiment when restricting the solution to $N_f$ Fourier modes.

## C.4 Testing at Different Resolutions.

We have emphasized repeatedly that CNO upholds the principle of continuous-discrete equivalence (CDE), which implies that there is an equivalence between the underlying operator and its discrete representations. As a reminder, the CNO models are operators denoted as $\mathcal{G}^* : \mathcal{B}_w(D) \to \mathcal{B}_w(D)$ and are designed to ensure that the continuous representations of functions align with their discrete samples on a uniform grid. This holds true when the sampling rate $s$ of the grid is sufficiently high, specifically $s \geq 2w$. It is important to note that the implemented CNO models are specified on a predefined *computational grid* with a sampling rate of $s \geq 2w$. Hence following [2] Remark 3.7, the input functions must be *compatible* with this grid. If the input function is not compatible with the computational grid, one needs transform it to an appropriate representation. Once the model is applied, the output is transformed back to the original representation (see Remark 3.5 of [2] for a formal explanation) and also Formula (A.7) for a precise description of these transformations.

Hence to apply an implemented CNO model to a continuous function $f \in \mathcal{B}_{w'}(D)$, it is necessary to employ a discrete representation of the function on a computational grid with a sampling rate of $s$. Essentially, it means that one needs to sample $f$ on that grid. If the band limit $w'$ exceeds half the sampling rate $s/2$, it is crucial to first filter out frequencies above $s/2$ to prevent aliasing effects, which involves applying a downsampling filter. Once the function's representation and the computational grid are compatible with each other, the model can be applied.

To apply an implemented CNO model to a discrete representation $f_{s'} \in \mathbb{R}^{s' \times s'}$, it is necessary to follow (A.7) and transform $f_{s'}$ into a compatible representation $f_s \in \mathbb{R}^{s \times s}$. If $s' \leq s$, the signal needs to be upsampled to the sampling rate $s$ by using an appropriate upsampling filter. However, if $s' > s$, it is necessary to filter out frequencies above $s/2$ that are present in the signal. One should downsample the signal to the sampling rate $s$ by applying an appropriate downsampling filter.

As highlighted in the main text, an important characteristic of an operator learning model is to maintain a relatively consistent test error when evaluated on various resolutions or discretizations. To assess this aspect, we evaluate the performance of UNet, FNO, and CNO models on different resolutions for Navier-Stokes equations. The original data is generated at a resolution of $128 \times 128$. To obtain data at any lower resolution $s < 128$, we downsample the original data to the desired resolution. The models that we use to make predictions are the ones that we trained on $64 \times 64$ resolution. The configurations of all the models are reported in C.2.

We apply the afore-described strategy to practically realize Formula (A.7) and apply CNO to different resolutions. In contrast, we follow the approach outlined in [33] to evaluate the FNO or UNet models at different resolutions by applying the underlying model directly to the original, unresized input.

We show the variations of the test errors across resolutions for the Navier-Stokes benchmark in Figure 2, right. The CNO model demonstrates the highest stability when it comes to resolution changes and is (approximately) invariant to resolution, unlike the other two models which exhibit notable fluctuations at different resolutions. Specifically, the UNet model displays a strong reliance on the training resolution, whereas the FNO model exhibits a slightly less pronounced dependence. This example show that the CNO model respects continuous-discrete equivalence, while the other two models are not resolution (or representation) equivalent.

## C.5    Ablation Studies.

We conduct two ablation studies focusing on two key aspects of CNO. Firstly, we examine the impact of modified operations, assessing how they affect the overall performance. Secondly, we investigate the influence of ResNets that connect the Encoder and Decoder components within the Operator UNet architecture (refer to Figure 1). These studies aim to provide valuable insights into the effects of these key elements.

In our first ablation study, we aim to evaluate the effects of modifying operations, including upsampling operators, downsampling operators, and activation layers, on performance and training time. It is worth reiterating that the modified operations enable *continuous-discrete equivalence* (CDE). Specifically, we replace the upsampling operator in the Operator UNet architecture with a discrete, nearest neighbor upsampling method, while the downsampling operator is substituted with average pooling. Additionally, we replace the activation layer with a simple pointwise application of the activation function. As a result, the model takes on a structure resembling a regular UNet architecture, but with the inclusion of additional ResNets that establish connections between the Encoder and the Decoder components. We will refer to this model as *CNO w/o Filters*.

The second ablation study focuses on evaluating the influence of additional ResNets that connect the Encoder and the Decoder components on both the overall performance and training time. In this study, we remove these ResNets while retaining the UNet-like concatenations between corresponding levels of the Encoder and the Decoder. It is important to note that the ResNet between the deepest levels of the Encoder and the Decoder is preserved within the model. This ablation model respects the *continuous-discrete equivalence* (CDE).

**Performance.**    We train two ablation models for every benchmark experiment that we studied in the main text. In order to maintain consistency, we use the same hyperparameter configurations for the ablation models as those of the best-performing CNO models (refer to Table 12 for the specific values). We report the in-distribution and out-of-distribution test errors in the Table 13.

Among the 16 tests conducted, the original CNO model outperforms the others in 12 of them. In other 4 cases, the testing errors are close to the testing errors of the best performing models. In almost all of the tests conducted, the first ablation model exhibits inferior performance compared to the original CNO model. This observation indicates that the aliasing errors resulting from regular CNN operations like average pooling, nearest neighbor upsampling and a regular application of the activation function have an impact on the test error. Furthermore, it is important to note that the first ablation study does not adhere to the continuous-discrete equivalence (CDE) property, resulting in the model's resolution dependence, similar to the UNet model (see Figure 2 and Section C.4).

In one out-of-distribution tests, the second ablation model demonstrates slightly superior performance compared to the original CNO model. It is worth noting that in many cases, the original CNO model exhibits significantly better performance than the second ablation model. This disparity in performance ranges from less than 10% in the Compressible Euler and Darcy Flow benchmarks to almost 300% in the Poisson Equation benchmark. While it is true that the second ablation model maintains the continuous-discrete equivalence (CDE) property, we observe that the inclusion of ResNets is vital for achieving good performance and decent generalization.

**Training time.**    Since the first ablation model does not utilize any interpolation filters, it is reasonable to anticipate that it will have a faster training time than the original CNO model.

Specifically, it trains approximately 1.75 times faster for the Poisson Equation, while for the Darcy Flow, it trains around 1.35 times faster. For the Navier-Stokes Equations, the training is 1.25 times faster. For the Wave Equation, Continuous Transport and Allen-Cahn Equation, it trains approximately

1.1 times faster. Finally, for the Discontinuous Transport, they need approximately equal amount of time to train.

The second ablation model, which excludes the middle ResNets from the architecture, is also expected to have faster training process than the original CNO model. Specifically, it trains approximately 1.75 times faster for the Poisson Equation. For the Navier-Stokes Equations and Darcy Flow, the model trains 1.25 times faster. The training for the Wave Equation, Discontinuous Transport and Compressible Euler is 1.1 to 1.15 times faster. Other benchmarks need similar amount of time to train.

Table 13: Relative median $L^1$ test errors, for both in- and out-of-distribution testing, for the CNO models and two ablation models.

|  | In/Out | CNO | CNO w/o Filters | CNO w/o ResNets |
|---|---|---|---|---|
| **Poisson Equation** | In | **0.21%** | 0.93% | 0.85% |
|  | Out | **0.27%** | 1.65% | 0.82% |
| **Wave Equation** | In | 0.63% | **0.59%** | 1.64% |
|  | Out | 1.17% | **1.12%** | 1.64% |
| **Smooth Transport** | In | **0.24%** | 0.31% | 0.31% |
|  | Out | **0.46%** | **0.46%** | 0.76% |
| **Discontinuous Transport** | In | **1.03%** | 1.21% | 1.17% |
|  | Out | **1.18%** | 1.32% | 1.60% |
| **Allen-Cahn** | In | **0.54%** | 0.69% | 0.71% |
|  | Out | 2.23% | **2.16%** | 2.21% |
| **Navier-Stokes** | In | **2.76%** | 3.20% | 3.00% |
|  | Out | 7.04% | 9.60% | **5.85%** |
| **Darcy** | In | **0.38%** | 0.47% | 0.41% |
|  | Out | **0.50%** | 0.65% | 0.58% |
| **Compressible Euler** | In | **0.35%** | 0.38% | 0.37% |
|  | Out | **0.59%** | 0.62% | **0.59%** |

### C.6   Error vs. number of training samples.

Once again, we revisit the best-performing CNO and FNO model architectures for the Poisson equation and Wave equation, as reported in C.2. This time, we focus on varying the number of training samples and retraining the selected CNO and FNO models accordingly. Consequently, we generate a plot that illustrates the in-distribution test error as we change the cardinality of the training set, as shown in Figure 27 (left for Poisson equation, right for Wave equation). In the case of Poisson equation, the CNO model outperforms by far the FNO model in all the data regimes.In the case of Wave equation, we notice that the FNO performs better than the CNO in low data regime, with the opposite behaviour in the large data regime. Moreover, CNO shows an approximately error decay rate of $0.5$ with respect to the number of training samples.

## D   Depiction of the Datasets.

In the following figures, we illustrate the different PDE forward problems considered in the main text.

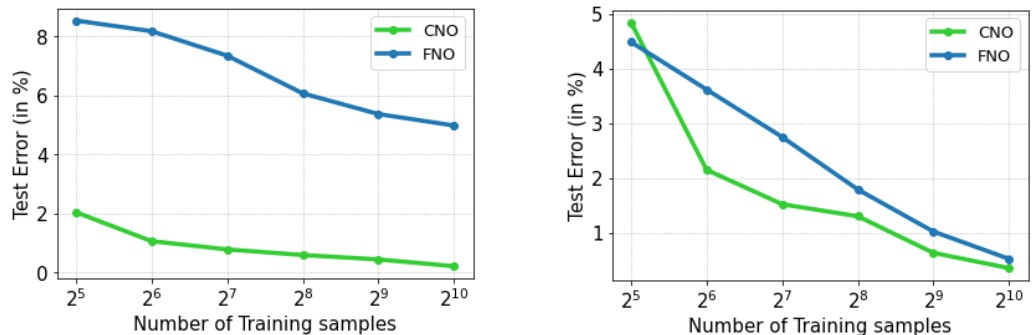

Figure 27: In-distribution testing errors for different cardinalities of the training set for FNO and CNO. Left: Poisson equation. Right: Wave equation.

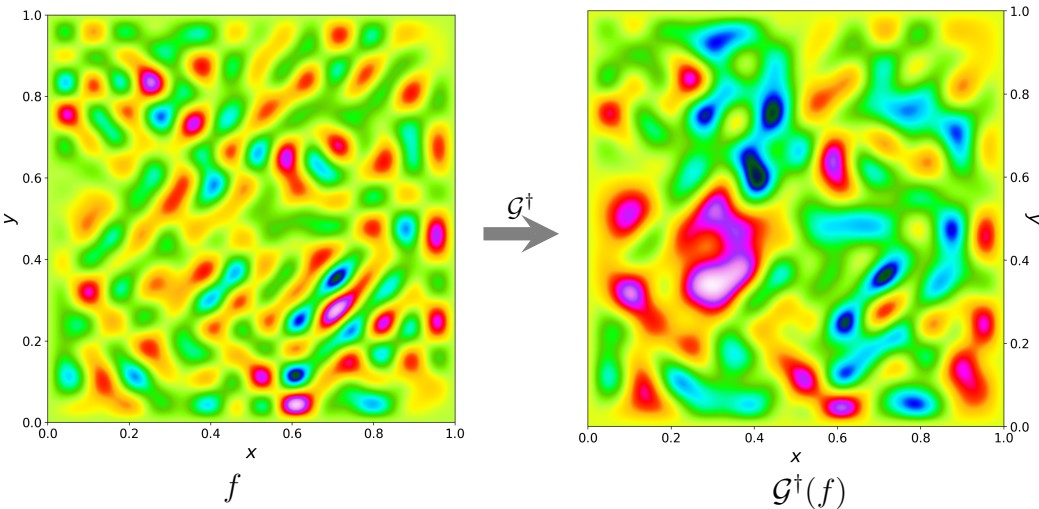

Figure 28: Illustration of input (left) and output (right) samples for the Poisson Equation.

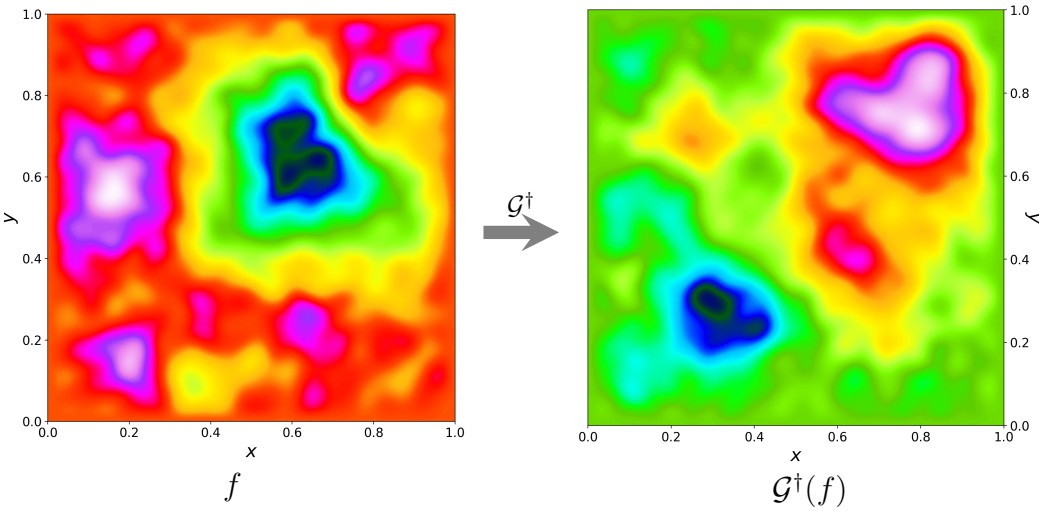

Figure 29: Illustration of input (left) and output (right) samples for the Wave Equation.

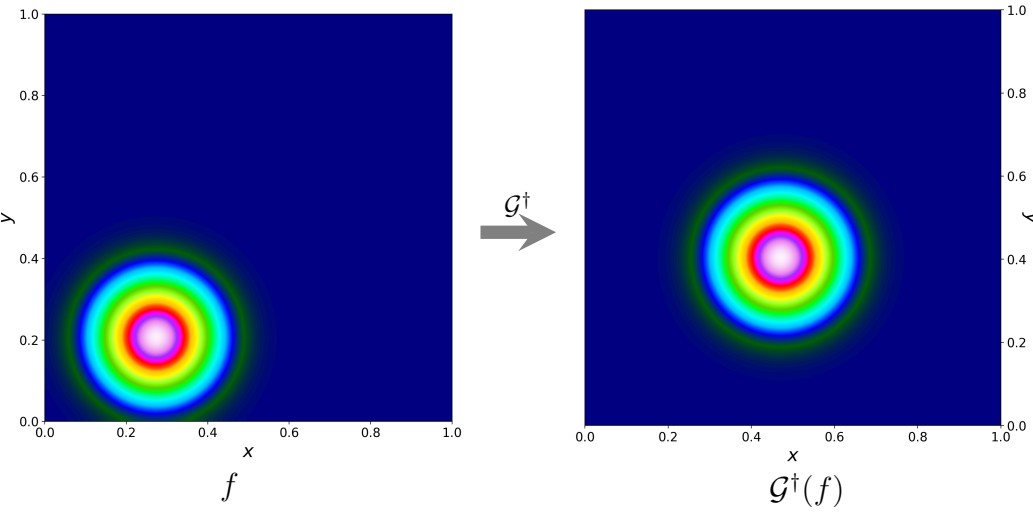

Figure 30: Illustration of input (left) and output (right) samples for the Continuous Transport.

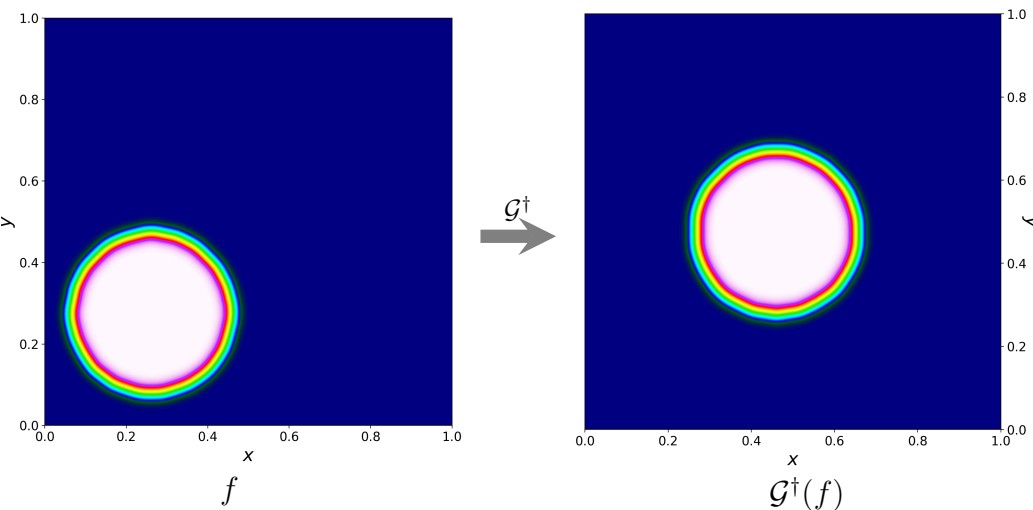

Figure 31: Illustration of input (left) and output (right) samples for the discontinous transport problem.

# E   Additional References

[64] E. Tadmor. Convergence of spectral methods for nonlinear conservation laws. SIAM Journal on Numerical Analysis, 26(1):30–44, 1989.474

[65] E. Tadmor. Burgers' Equation with Vanishing Hyper-Viscosity. Communications in Mathematical Sciences, 2(2):317 – 324, 2004

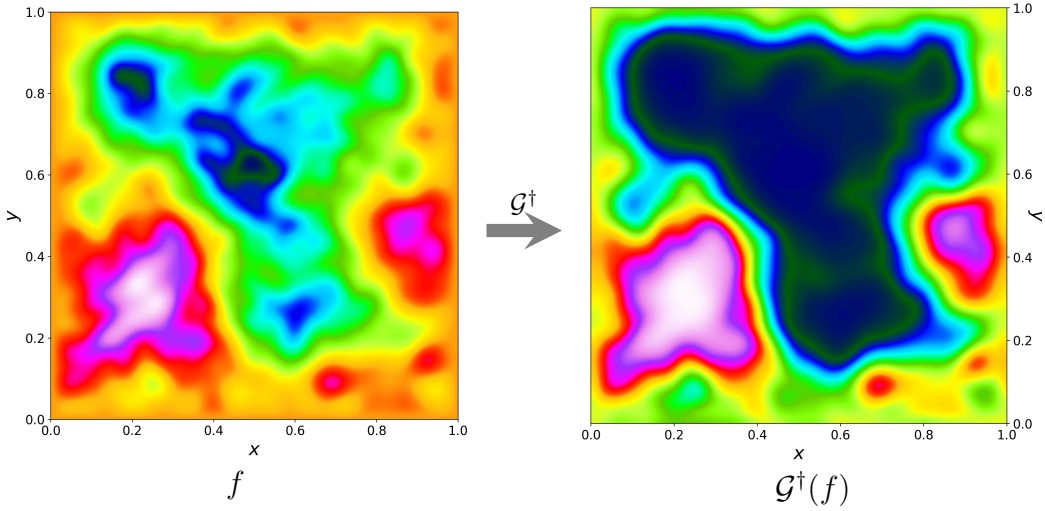

Figure 32: Illustration of input (left) and output (right) samples for the Allen-Cahn equation.

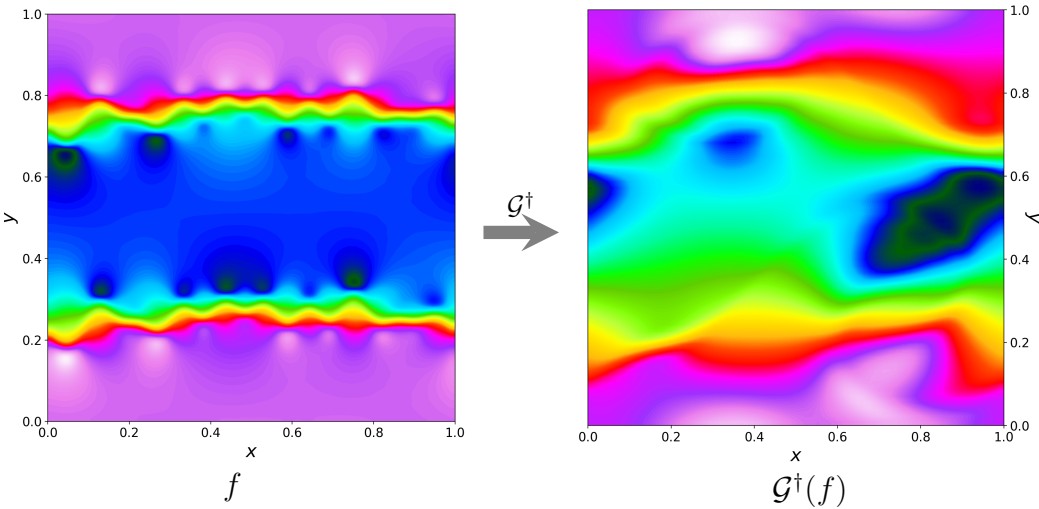

Figure 33: Illustration of input (left) and output (right) samples for the Navier-Stokes equations.

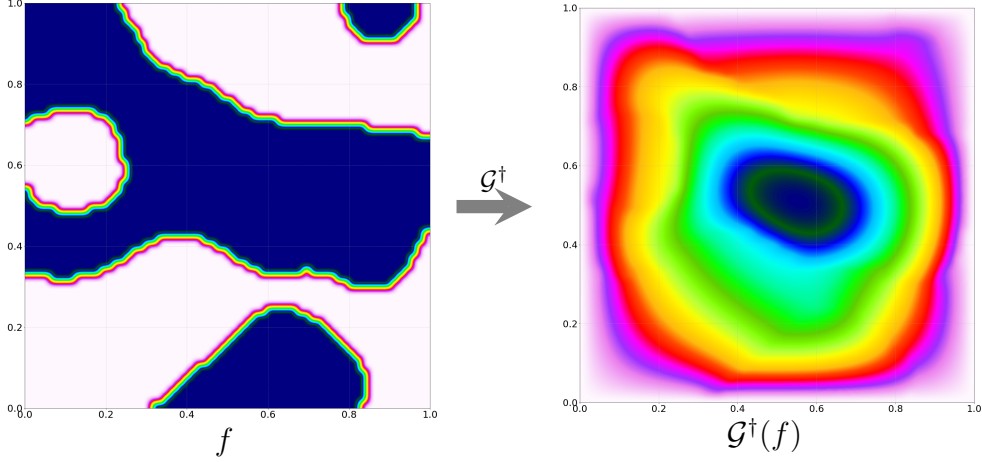

Figure 34: Illustration of input (left) and output (right) samples for the Darcy flow.

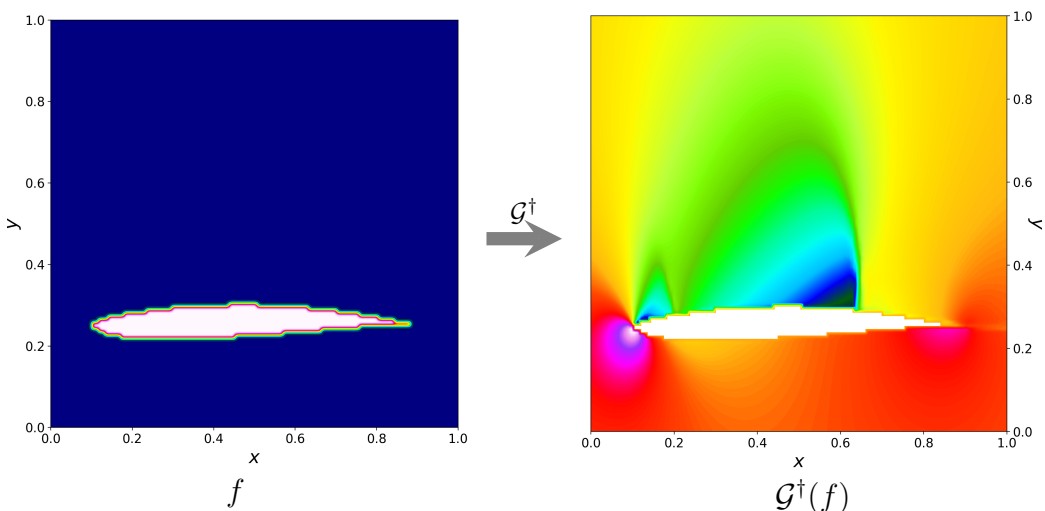

Figure 35: Illustration of input (left) and output (right) samples for the compressible Euler equations.