# OpenReview forum: "Convolutional Neural Operators for robust and accurate learning of PDEs"
_NeurIPS.cc/2023/Conference — NeurIPS 2023 poster_

### Official Review · Reviewer_uN3N · 2023-07-05

**Soundness:** 3 good
**Presentation:** 3 good
**Contribution:** 3 good
**Rating:** 7
**Confidence:** 4

**Summary:**

The authors propose a UNet-like architecture for learning solution operators of PDEs. The architecture is assembled from building blocks analogous to the classical UNet architecture, but individual components respect a continuous-discrete equivalence. This makes the proposed architecture a representation equivalent neural operator in the sense of a recent paper (Bartolucci et al., 2023). The authors show that under some continuity assumptions the proposed method can learn the solution operators to a variety of PDEs. They also compare the in- and out-of-distribution performance of their method to a variety of baselines for a suite of different types of PDEs.


**Strengths:**

- There is a large body of work on methods and architectures for operator learning which this paper builds up on. Having architectures that respect the continuous-discrete equivalence seems very significant to me.
- The presentation is clear and the paper overall well written.
- The proposed CNO architecture improves the overall performance for a comprehensive suite of problems and appropriate baselines.
- Many evaluations and ablation studies are included in the appendix.


**Weaknesses:**

- As discussed in the related work section, the building blocks that make up the CNO architecture have been introduced in previous papers (Karras et al. 2021, Alias-free generative adversial networks). This slightly weakens the novelty of the paper.
- Many definitions are adopted from a relatively recent paper (Bartolucci et al. 2023), which to my knowledge is only available as a preprint at the moment.

- The modifications to the UNet architecture introduce additional up-sampling and down-sampling layers, which increase the training time significantly compared to other methods. This needs to be kept in mind. Also, the implementation of CNOs is much more complicated than of e.g. FNOs due to the windowed-sinc filters.

- I found the evaluation across resolutions somewhat unclear and partially misleading. The appendix goes into more detail here, but the results in the main paper show a single NS case that resulted from a downsampled solution (hence no high-frequency details exist in the targets). In this case the CNO seems to show a constant error, which is dubious for meaningful real world cases where higher resolutions naturally would exhibit structures that aren't resolved with lower resolutions. I think it will be important for future versions to clarify this, and replace figure 2 with one of the other two cases from the appendix.


**Questions:**

- The practical implementation of the interpolation filters uses windowed-sinc. In theory, doesn't this break the continuous-discrete equivalence?

- Regarding "operator" networks, I was wondering why the authors didn't compare to a fully convolutional ResNet (along the lines of Solver-in-the-loop Um'21 or the accelerated CFD from Kochkov'21); that architecture should more naturally extend to different resolutions than a Unet. Can the authors comment on this omission?

- Can the authors provide details on the performance, especially how much slower the CNO is compared to the regular Unet?


**Limitations:**

Limitations are only discussed as a future work discussion. I don't think this is sufficient. E.g., a clear discussion of the performance impact is completely missing as far as I can tell.

---

> ### Author Rebuttal · Authors · 2023-08-06
>
> We start by thanking the reviewer for your appreciation of the merits of our paper and your welcome suggestions to improve it. We address your detailed concerns below.
> 1. Regarding the reviewer's point about novelty vis-a-vis Karras et al 2021, while acknowledging that we have used building blocks, particularly the dealiasing trick from there, we would also like to highlight the novelty of our contributions with respect to this paper. They are 1) unlike in Karras et al, we had to rewrite their discrete operators in a continuous setting in order to realize CNO as a neural operator 2) Karras et al considered a very different setting (image generation) and a significant part of the novelty of our paper lies in realizing these ideas in the setting of approximating PDE operators where these ideas have not been previously proposed 3) we instantiate CNO with a different UNet type model than the construction of Karras et al 4) We prove that CNO is a ReNO and can approximate any
> continuous operator mapping between Sobolev spaces and 5) provide considerable amount of empirical testing of CNO on many PDE benchmarks. Some of these points are already mentioned in lines 312-318 of the paper and this differentiation from Karras et al will be sharpened and strengthened in the main text of a CRV, if accepted.
> 2. Indeed, the key definition of ReNO is adopted from Bartolucci et al. Given the 9-page limit, we did not have the space to include this definition in the main paper. But with an additional page in the CRV, if accepted, we would include the definition and make the paper as self-contained as possible.
> 3. The reviewer's point about the computational efficiency of CNO, vis-a-vis other models is certainly valid. To address your question, we have included a figure (Figure 1) in the uploaded pdf with this rebuttal where we we showcase the comparative efficiency of CNO vis-a-vis FNO for the Navier-Stokes benchmark (see detailed description in SM C3.5) in Figure 1 in the uploaded pdf for this rebuttal. In this figure, we plot the validation error (Y-axis) for all the CNO and FNO models we have tested (details in SM C.2) with respect to  Model size (Left panel) and per-epoch training time (Right panel). We see from this figure that 1) for fixed model size: CNO is clearly more accurate than FNO 2) For fixed training time: CNO is also clearly more accurate than FNO 3) Instead of selecting the CNO with the smallest error (as in the paper), if we select a more efficient CNO model (highlighted with dark Red in Figure 1), we observe that this model has almost two orders of magnitude less parameters (model size) than the best performing FNO while also being more than 50% faster (2.8 secs vs 4.4 secs) in terms of training time and at the same time, with a significantly smaller validation error (3.5%) compared to FNO (4%). This figure demonstrates that CNO is not only more accurate but clearly more efficient than FNO for this problem. Needless to say, we will include a detailed efficiency study along these lines for each of the benchmarks and also compared with other models in the CRV, if accepted and thank the reviewer for asking us to explore this very important issue of computational efficiency that we believe further strengthens our message.
> 4. The ground truth for the Navier-Stokes experiment is computed at a resolution of $128^2$. As the reviewer alludes to, the CNO discretization invariance results in Figure 2 can be explained by the fact that almost all the frequency information in the problem is already contained at the training resolution of $64^2$) (See spectrogram for Ground Truth Fig.2 Left) and there is very little high-frequency content in finer discretizations. On the other hand, and as spotted by the reviewer, for the Poisson problem, Spectogram (SM Fig. 6) and discretization dependence (SM fig 24) show that missing high-frequency information can cause small test errors as the test resolution is increased towards $128^2$, corroborating our explanation about the reviewer's question on Figure 2. We agree with the reviewer's contention that SM Figure 24 is a better visual representation of this issue and we will include it (and SM Figure 6) for further context on this issue in a CRV, if accepted.
> 5. Indeed windowed sinc is an approximation to the ideal sinc filter and can induce some aliasing error. This issue has been mentioned and discussed in some detail in SM C.1.3 and the effects of windowing are quantified there.
> 6. We thank the reviewer for their excellent suggestion of testing out fully-convolutional ResNets as baselines. As the solver-in-loop and Kochkov et al ResNets are in-house, we use the widely used CycleGAN implementation of convolutional ResNets (link in Table 2 caption). Given the time constraints for the rebuttal, we could only perform the tests with some benchmarks and the results for both in- and out-of-distribution testing are presented in Table 2 of the uploaded pdf. They show that ResNets are a competitive baseline on some problems but do not change the main conclusions of Table 1 of our paper. We will include convolutional ResNets as an additional baseline in the CRV, if accepted.
> 7. As the reviewer points out, adding additional filtering operations will increase the cost of CNO. This issue has already been addressed in an ablation study shown in SM C.5 where the CNO w/o filters model is similar to an UNet. We report there (see lines 1280-1285) that removing filters is between 1.5-2 times faster than including them and UNet has a similar training time. However, we introduce very large aliasing errors by doing so (Fig 2 and SM fig 24). Hence, we argue that the additional cost is justified to retain CDE and a fairer comparison in efficiency is wrt FNO (see pt 3 for this comparison).
> 8. We will further discuss the computational efficiency of CNO (see pts 3 and 7) in a CRV, if accepted.
>
> We sincerely hope to have addressed your concerns to your satisfaction.

---

### Official Review · Reviewer_ZTbu · 2023-07-06

**Soundness:** 4 excellent
**Presentation:** 3 good
**Contribution:** 4 excellent
**Rating:** 8
**Confidence:** 4

**Summary:**

In this paper, the authors propose a new operator learning framework called Convolution Neural Operator which although works on discrete space, satisfies the property of Continuous-Discrete Equivalence (CDE) property. They define CNO over a UNet architecture and provide some universal approximation proofs. Further, they come up with a Benchmark of representative PDEs and compare all the existing methods and present superior performance of CNO over other methods.

**Strengths:**

+ Novel idea of CNOs
+ Theory provided for soundness
+ Representative Benchmark Dataset and making it public in zenodo is also good.
+ Very neat and clean code.

**Weaknesses:**

- Some notations are messed up or confusing to follow. e.g. r in line 82.
- While the performance of these models is compared... the computation requirements like memory or training time are not compared. I think if we are talking about benchmarking, it is appropriate to give the complete story including the computational requirements.
- Some ablation studies on whether using CNOs reduces the architecture/compute requirements compared to just using a CNN or FNO is missing.

**Questions:**

See above in weaknesses*

**Limitations:**

Yes. The authors have accurately identified the limitations. There are no negative societal impact for their work.

---

> ### Author Rebuttal · Authors · 2023-08-06
>
> We start by thanking the reviewer for your appreciation of the merits of our paper and your welcome suggestions to improve it. We address your detailed concerns below.
>
> 1. We apologize about the possible lack of clarity in notation for $r$ on line 82 -- it refers to the smoothness index $r$ of the Sobolev space $H^r$, introduced in line 67 where the underlying operator maps from this Sobolev space. We will make this $r$ more explicit in a CRV, if accepted and also comb through the paper for any further notational ambiguities.
>
> 2. Our focus in the empirical part of the paper was on accuracy of the tested architectures and during model selection, we chose the hyperparameters for the model with the lowest validation error. We agree fully with the reviewer's point that efficiency, particularly in terms of model size and training time, is also a very important aspect and following your excellent suggestion, we showcase the comparative efficiency of CNO vis-a-vis FNO for the Navier-Stokes benchmark (see detailed description in SM C3.5) in Figure 1 in the uploaded pdf for this rebuttal. In this figure, we plot the validation error (Y-axis) for all the CNO and FNO models we have tested (details in SM C.2) with respect to  Model size (Left panel) and per-epoch training time (Right panel). We see from this figure that 1) for fixed model size: CNO is clearly more accurate than FNO 2) For fixed training time: CNO is also clearly more accurate than FNO 3) Instead of selecting the CNO with the smallest error (as in the paper), if we select a more efficient CNO model (highlighted with dark Red in Figure 1), we observe that this model has almost two orders of magnitude less parameters (model size) than the best performing FNO while also being more than 50% faster (2.8 secs vs 4.4 secs) in terms of training time and at the same time, with a significantly smaller validation error (3.5%) compared to FNO (4%). This figure demonstrates that CNO is not only more accurate but clearly more efficient than FNO for this problem. Needless to say, we will include a detailed efficiency study along these lines for each of the benchmarks and also compared with other models in the CRV, if accepted and thank the reviewer for asking us to explore this very important issue of computational efficiency that we believe further strengthens our message.
>
> 3. The ablation study that the reviewer refers to has been addressed in some detail in point 2 of our rebuttal to your comments. In figure 1 of the uploaded rebuttal pdf, we have only compared with FNO due to time constraints as well as the fact that FNO is acknowledged as a state of the art neural operator. We will also compare with other models, include CNNs (UNets) and present the results in a CRV, if accepted. We would also like to refer the reviewer to SM C.5. where we have performed a detailed ablation study for different building blocks of CNO in order to investigate which building blocks are crucial to its functioning.
>
> We sincerely hope to have addressed  your concerns, particularly about the computational efficiency of CNOs and would thank the reviewer again for pointing out these concerns.

---

> > ### Comment · Reviewer_ZTbu · 2023-08-14
> >
> > I have read the comments from the authors and am satisfied with the paper.  I have increased my rating to a strong accept. Cheers!

---

> > > ### Author Response · Authors · 2023-08-14
> > > **Thanking the Reviewer**
> > >
> > > We thank the reviewer for their positive comments and for increasing our score.

---

### Official Review · Reviewer_MYhB · 2023-07-06

**Soundness:** 3 good
**Presentation:** 2 fair
**Contribution:** 3 good
**Rating:** 5
**Confidence:** 4

**Summary:**

The paper under review proposes a CNN architecture as a neural operator for solving PDEs via neural networks.  The goal is to preserve the underlying continuous structure while being implemented with discrete convolutional architecture, based on U-Net.  The idea is to consider the space bandlimited functions in Sobolev spaces.  Defining a convolution layer involves a discrete convolution, upsampling/downsampling that requires sinc interpolation to avoid aliasing, and an activation layer that upsamples, activates and then downsamples to maintain the original band-limitation.  The authors prove in supplementary material that their approach can approximate the solution to a PDE with arbitrary accuracy with the CNN architecture based on their neural operator.  Experiments are done on solving several PDEs, with experiments favorable to the authors' method against state of the art for most PDEs considered.

**Strengths:**

- The architecture based on CNNs potentially simplifies e.g., FNO operators where operations are done in the Fourier domain without the use of down-sampling - which can potentially be more expensive than the authors' proposed method.
- Rigorous mathematical treatment of the interplay between continuous and discrete operators.
- Experiments show higher accuracy compared to SOA in PDE solving with NN.

**Weaknesses:**

- No evaluation of efficiency; is there a speed advantage of the proposed method (e.g., compared to FNO)?
- Evaluation of parameter sizes is in supplementary and there isn't a clear trend; I would have thought this method would require fewer parameters than e.g., FNO; but in many cases this isn't so.
- I would think efficiency in the sense above is a key differentiator compared to SOA, but this isn't the case at least looking at supplementary.
- The authors make use of sinc interpolation, which is the ideal interpolator for sequences of infinite length.  In practice, you are operating on finite length data, for which the sinc interpolater is not the right one.  How applicable is the theory presented to the case of finite length data?
- Prop 2.1 talks about a representation equivalent operator and the definition is cited in a reference.  Please specify the definition in the paper.
- Theory is poorly presented in the paper; everything is in supplementary.  At least the key ideas should be in the main paper.
- In general many of the key parts that should be in the paper are in supplementary.
- I find the authors' proposed idea to set a standard for benchmarking misguided.   There is a laundry list of PDEs to approximate in the benchmark, however, I doubt any one architecture would be good for all PDEs.  Each PDE has particular properties that one would want to aim to preserve (e.g., conservation laws in some PDEs) that may not be relevant to others.  So the implication that one architecture should do well on all PDEs seems mis-guided.  I do not think this should be an accepted standard benchmarking.

**Questions:**

- Bandlimited: many solutions to PDEs will have shocks, discontinuities, etc.  How relevant is this assumption?

**Limitations:**

Yes, discussed.

---

> ### Author Rebuttal · Authors · 2023-08-06
>
> We start by thanking the reviewer for your appreciation of the merits of our paper and your welcome suggestions to improve it. We address your detailed concerns below.
> 1. Our focus in the empirical part of the paper was on accuracy and during model selection, we chose the hyperparameters for the model with the lowest validation error. We agree fully with the reviewer that efficiency is also a very important aspect and following your excellent suggestion, we showcase the comparative efficiency of CNO vis-a-vis FNO for the Navier-Stokes benchmark in Figure 1 in the uploaded pdf for this rebuttal. In this figure, we plot the validation error (Y-axis) for all the CNO and FNO models we have tested (details in SM C.2) with respect to  Model size (Left panel) and per-epoch training time (Right panel). We see from this figure that 1) for fixed model size: CNO is clearly more accurate than FNO 2) For fixed training time: CNO is also clearly more accurate than FNO 3) Instead of selecting the CNO with the smallest error (as in the paper), if we select a more efficient CNO model (highlighted with dark Red in Figure 1), we observe that this model has almost two orders of magnitude less parameters than the best performing FNO while also being more than 50% faster (2.8 secs vs 4.4 secs) and at the same time, with a significantly smaller validation error (3.5%) compared to FNO (4%). This figure demonstrates that CNO is not only more accurate but clearly more efficient than FNO for this problem. Needless to say, we will include a detailed efficiency study along these lines for each of the benchmarks and other models in the CRV, if accepted and thank the reviewer for asking us to explore this very important issue that we believe further strengthens our message.
> 2. The reviewer raises an important point about infinite length sinc filters. We have already discussed this issue in some detail in SM C.1.3 about the use of windowed sinc filters, where we have also shown the effect of choosing different window lengths.
> 3. Given the 9-page limit for the main paper, we had to make some difficult choices about what to include in the main paper and what to include in the SM. In particular, the definition of Representation equivalent neural operator (ReNO) from Bartolucci et. al. is quite technical and we had not included it. We had tried to motivate the heuristics behind this definition in lines 137-141 but we agree with the reviewer's point and will include the definition of ReNO in the CRV, if accepted. This will also enable us to make the paper more self-contained.
> 4. Our objective was to motivate the theory in the main paper while presenting the very detailed and technical arguments of the proofs in the SM, particularly SM A.1, A.2, A.5 and B. The fact that the motivations were very terse is partly due to the 9-page limit. A CRV, if accepted, allows us to add an additional page and we will focus on providing more motivation about the theory in this additional space. In particular, we will add more details about the definition of ReNO and how and under what conditions CNO is a ReNO and what happens when these conditions are satisfied only approximately. We will also provide an outline of the proof of theorem 3.1, mentioning the main steps while postponing the details to the SM. We again thank the reviewer for highlighting this issue and acknowledge that it is difficult to balance the main text/SM delineation within a strict page limit. We will also use the additional space to discuss computational efficiency (see Point 1).
> 5. We apologize for possible misunderstanding in our write-up about benchmarks (line 177-192), Clearly, we agree with the reviewer that PDEs are too vast in scope to allow a single set of benchmarks. This was not our aim. Rather, our starting point was to propose a set of criteria for choosing possible benchmarks and these are outlined in lines 179-182. We believe that the reviewer will agree that these are necessary criteria for good benchmarks while we acknowledge that these are not sufficient and other issues such as conservation, symmetries etc should also be considered. We also use this criteria to critically examine some existing benchmarks (see SM C3.7). We can include a further discussion on these criteria in a CRV. Moreover, we have not claimed that one model will do well on all chosen problems -- in fact, Table 1 already shows a diversity of best-performing models and we will make that more explicit in a CRV, if accepted.
> 6. Consider a PDE with discontinuities such as linear advection Eqn with discontinuous initial data. The underlying solution operator is readily approximated (in $L^2$) to desired accuracy by mollifying the initial data and this approximate operator, which maps between smooth functions, can be approximated further by an operator mapping between band-limited functions (SM A.1). Similarly, consider inviscid Burgers' Eqn with sine-wave initial data -- the solution operator will contain shocks but we can approximate it to desired accuracy (in $L^1$) by the solution operator of viscous Burgers' (with small viscosity) which maps between smooth functions and can, in turn, be approximated by operators mapping between band limited functions (SM A.1). Again, the underlying operator can be approximated to desired accuracy by an operator mapping between bandlimited spaces. This consideration is also corroborated by the section on numerical experiments where we consider a 2D advection equation with moving discontinuities (SM C3.3) as well as a compressible Euler equations with shocks (SM C3.6). In both cases (see Table 1), CNO which maps between band limited functions, performs very well for both in-distribution as well as for out-of-distribution testing, supplementing our arguments above.
>
> We sincerely hope to have addressed  your concerns, particularly about the computational efficiency of CNOs and would kindly request the reviewer to update their assessment accordingly.

---

> > ### Author Response · Authors · 2023-08-19
> > **Requesting the reviewer for feedback**
> >
> > Due to imminent closure of the discussion period, We kindly request the reviewer to provide us with your valuable feedback on our rebuttal and we are at your disposal to answer any further questions in this regard

---

> > ### Comment · Reviewer_MYhB · 2023-08-20
> > **response to rebuttal**
> >
> > 1. Thanks for the additional plots about efficiency, which shows an advantage in the authors' method.
> > 2. I was referring to what you can say theoretically about the finite length case, not empirical.
> >
> > While a good contribution, I thought the presentation could be greatly improved.

---

> > > ### Author Response · Authors · 2023-08-21
> > > **Reply to the reviewer's comments.**
> > >
> > > We thank the reviewer for your comments and answer your follow-up questions below:
> > >
> > > 1. We thank the reviewer for clarifying that they wanted a theoretical statement about finite-length case, rather than an empirical one and we apologize for misinterpreting your question here. We can also make a very clear theoretical statement about this issue. It goes as follows: replacing the infinite length with the finite length (filter windowing) may, in general, lead to aliasing errors, as defined in Eqn (3.1) of Bartolucci et. al. Arxiv:2305.19913v1. However, it is relatively straightforward to show a) that this error can be made as small as possible by considering sufficient filter width and b) for input and output signals of given regularity, for instance Sobolev regularity as is common with the inputs/outputs of PDEs, one can explicitly bound this aliasing error in-terms of the Sobolev regularity for any windowing length of the filter. In particular, the decay of this error will be exponentially fast in the Sobolev exponent $r$ of our problem formulation. These results are totally consistent with our empirical results, shown in the SM, where the errors are very small even for short window lengths. Finally, finite-lengths do not affect the universal approximation theorem 3.1 as the resulting aliasing errors can be made as small as desired in the above sense. We will certainly include a discussion on this issue in the SM, with mathematically precise results as outlined above, of a CRV, if accepted.
> > >
> > > 2. We have already laid out concrete steps to improve the presentation, following the reviewer's suggestions, in our original rebuttal.
> > >
> > > We sincerely hope that we have addressed both the reviewer's original empirical concerns and current theoretical question to your satisfaction. If so, we kindly request the reviewer to upgrade their assessment accordingly.

---

### Official Review · Reviewer_KkQX · 2023-07-14

**Soundness:** 3 good
**Presentation:** 4 excellent
**Contribution:** 3 good
**Rating:** 6
**Confidence:** 4

**Summary:**

The paper re-introduces convolution neural networks to the operator learning setting. It uses interpolation including up-sampling and downsampling layer to make sure the convolution layer is well-defined in the function space. The paper also discuss the space of band-limited functions and the trade off between continuous-discrete equivalence and representation power of the infinite dimension function space. The paper proves an approximation theorem for the CNO model and compared it across many partial differential equations.

**Strengths:**

The convolution neural operator (CNO) takes the advantages of the efficient of the conventional CNNs and UNets methods, meanwhile it also satisfies the resolution-invariant and representation-equivalent properties. The work shows an approximation theorem that the CNO models can approximate any continuous solution operator, and the numerical experiments show it has a comparative performance compared to other machine learning methods such as  UNet, DeepONet, and FNO.

**Weaknesses:**

I have a few questions and concerns, mainly in the trade-off between the band-limited functions and Lp/Hp space.
1. The paper claim the CNO models can only learn band-limited functions, however, the target PDE system are intrinsically infinitely dimensional system (in Lp/Hp). It means, given a fixed number of parameters (size of the model), as the number of training sampling and the qualify(resolution) of training dataset increase, the truncation error will dominate and the bandlimited class of operator will underperform the band-unlimited models.
2. The band-limited operator is, by definition, linear-reconstruction of the chosen basis, meaning it can only learn the coefficient but unable to learn the basis. The linear-reconstruction are less efficient compared to non-linear reconstruction model, as discussed in [1].
3. The experiments (Table 1) are designed in a manner that CNO dominates all other methods. It shows the potential of CNO, but the results might be biased. It's certainly reasonable that the band-limited CNO outperform band-unlimited model on super-resolution tasks. However, for in-distribution problems, if the resolution is fixed and the dataset is sufficient, then Unet should be equivalent to CNO, right? And for these non-smooth problems, the band-unlimited could be more expressive than CNO. These cases might not be sufficiently considered in the experiments design. It is great to introduce these new experiments, but it would be better to include as least one of the previous standard benchmark, for example, the Burgers or Darcy equation dataset from [2].
4. While the perspective is new and interesting, it's hard to say the proposed model is very novel. The architecture is highly similar to the previous UNet model + StyleGAN3 de-aliasing trick.

[1] Lanthaler, Samuel, et al. "Nonlinear reconstruction for operator learning of pdes with discontinuities." arXiv preprint arXiv:2210.01074 (2022).

[2] Li, Zongyi, et al. "Fourier neural operator for parametric partial differential equations." arXiv preprint arXiv:2010.08895 (2020).

**Questions:**

As discussed in page 4, CNO uses upsampling and downsampling to reduce the aliasing error, but it's also claimed that the activation layer will introduce the aliasing error. This is overcome by "Implicitly assuming that $\bar{w}$ is large enough". Can the author give some examples of the activation such that this is satisfied? Speaking of frequencies expansion, for ReLU or leaky ReLU, $\sigma(x)$  is non-smooth. Even if $\sigma$ is a k-frequency function, $\sigma \circ \sigma$ can be $k^2$-frequency, right? So after several layers it easily goes unbounded.

Figure 2 is also a bit surprising. Can CNO achieve constant super-resolution error curve for problem like Navier-Stokes? The FNO error also seems a bit too high. How many modes are used in FNO? If FNO is designed with a reasonably small number of modes and no padding (padding may cause error in resolution), FNO may also get flatten curve.

**Limitations:**

The author discussed some limitations, but it's also good to discuss the trade-off between using band-limited functions and Lp/Hp space. I think both side have their advantages and disadvantages.

---

> ### Author Rebuttal · Authors · 2023-08-05
>
> We start by thanking the reviewer for your comments which we address below.
>
> 1. Regarding the reviewer's question about "trade-off between $L^p-H^p$ functions and bandlimited functions" and concern that "CNO only learns bandlimited functions", we start by pointing out that in Theorem 3.1 of our paper, we prove an universal approximation theorem by showing that CNOs can approximate any continuous operator (with a modulus of continuity), mapping between two ($H^p$) Sobolev spaces, to desired accuracy. Most PDE solution operators satisfy this assumption. As was shown in SM A.1, our approximation is performed in two steps: first, using the modulus of continuity of the underlying operator, it can be approximated to desired accuracy by an operator mapping between band-limited spaces  and then this operator is in turn, approximated by CNOs. Thus, the "truncation error" can be made arbitrarily small by choosing the right bands. In practice, the bands are given by the resolutions of the underlying input and output function, which also determine the associated truncation error. It is unclear to us why the size of the training dataset matters in this context, in fact, in SM Figure 25, we show empirically how the test error of CNO decreases with increasing sample size. Also, we would like to emphasize that "band-unlimited" models are subject to discretization errors as the data is only available at finite resolutions, which automatically determines the size of their discretization error. Given this very valid question raised by the reviewer, we will discuss this issue further in a CRV, if accepted.
> 2. We would like to clarify that the results of [1] were specific to the case of scalar 1-D transport of a single discontinuity and a related 1-D Burgers example where they claimed that FNO is better than DeepONet as the approximation error of a linear reconstruction operator can be bounded below by the rate of decay of eigenvalues that may converge slowly for some transport equations. This does not imply that methods with a linear reconstruction cannot be efficient for a very large class of PDEs which have a regularizing effect resulting in fast spectral decay. Moreover, the result of [1] only pertains to size efficiency vis a vis approximation error. Even if a model is size efficient, the overall error will be dominated by the finite resolution of given data. Another pertinent source of error is the aliasing error (as discussed in our paper) and a nonlinear reconstruction might lead to aliasing errors which can affect the test error whereas, CNO is based on reducing this important source of error. Thus, a priori, it is not possible to rule out that a method with a fixed basis is necessarily less accurate than a nonlinear reconstruction method. To further investigate this question,  we have extensively tested our method on transport problem with discontinuities as well as on a compressible Euler problem with shocks and contacts and in both cases (see Table 1), despite the presence of discontinuities, CNO was comparable to FNO in-distribution and significantly better out-of-distribution due to translation invariance, demonstrating that method like CNO can also be very competitive in approximating transport-dominated problems. We will include a discussion on this question in the CRV.
> 3. Regarding reviewer's concern about the choice of benchmarks, we would like to state that the benchmarks were explicitly chosen, not to favor CNO. On the contrary, they include problems with discontinuities (Transport, Euler), sharp gradients (Allen-Cahn) and problems with explicit solutions in Fourier space (Poisson, Wave) which we argue favor FNO a priori. However, the results clearly show that even on this dataset, CNO is competitive with FNO (and many times is performing better) (Table 1). Similarly we had already presented the Navier-Stokes vorticity benchmark from [2] in SM C.3.7, providing a clear rationale on why did not include this benchmark in the main paper. Even here (Line 1177), CNO is marginally better than FNO. Following the reviewer's excellent suggestion, we will also include the Darcy benchmark from [2] in the CRV. The results shown in uploaded pdf (Table 1) again show that CNO performs very well on this benchmark too. Finally, apart from the filtering operators, CNO is also different from the traditional U-Net architecture of Ref. [49] in using ResNets connecting encoder and decoder as well as in using a different pooling operation.
> 4. We differentiate our work from StyleGAN paper in lines 314-318 in related work section and the difference with U-Nets are described above.
> 5. A detailed discussion on activation functions and aliasing is provided in SM Remark B.4 where polynomial and rational activation functions are given as examples of exactly alias-free activations. Moreover, the choice of $\bar{\omega}$ is also extensively discussed is remark SM Section C1.3 and Remark C.1, see Table 2 for a comparative study on how the choice of $\bar{\omega}$ affects performance.
> 6. The CNO discretization invariance results in Figure 2 can be explained by the fact that almost all the frequency information in the problem is already contained at the training resolution of $64^2$) (See spectrogram for Ground Truth Fig.2 Left) and there is very little high-frequency content in finer discretizations. On the other hand for the Poisson problem, Spectogram (SM Fig. 6) and discretization dependence (SM fig 24) show that missing high-frequency information can cause small test errors as the test resolution is increased towards $128^2$, corroborating our explanation about the reviewer's question on Figure 2. Regarding the discretization dependence of FNO, we agree with the reviewer that padding might be a factor in causing in some aliasing error and will investigate the effect of padding on FNO errors in a CRV.
>
> We hope to have addressed all the reviewer's concerns and kindly request you to update your assessment accordingly

---

> > ### Author Response · Authors · 2023-08-19
> > **Requesting the reviewer for feedback**
> >
> > Due to the imminent closure of the discussion period, We kindly request the reviewer to provide us with their valuable feedback on our rebuttal and we are at their disposal to answer any further questions in this regard

---

### Author Rebuttal · Authors · 2023-08-06

At the outset, we would like to thank all four reviewers for their thorough and patient reading of our article. Their criticism and constructive suggestions will enable us to improve the quality of our article. If our paper is accepted, we will incorporate all the changes that we outline below in a camera-ready version ( CRV) of our article. As allowed by the conference, we are uploading a one page pdf that contains figures and tables on numerical experiments which support our arguments below. With this context, We proceed to answer the points raised by each of the reviewers individually, below.

Yours sincerely,

Authors of "Convolutional neural operators for robust and accurate learning of PDEs".

---

### Decision · Program_Chairs · 2023-09-21

**Decision:**

Accept (poster)

**Comment:**

Overall the reviewers all appreciated the work and the use of convolution based neural operators may be of interest to the community. There were some good questions by reviewers and the authors should add these discussions and promised changes to the paper.